# Optimal Discriminators for GANs with Higher-order Gradient Regularizers

## Abstract

We consider the problem of optimizing the discriminator in generative adversarial networks (GANs) subject to higher-order gradient regularization. We show analytically, via the least-squares (LSGAN) and Wasserstein (WGAN) GAN variants, that the discriminator optimization problem is one of high-dimensional interpolation. The optimal discriminator, derived using variational calculus, turns out to be the solution to a partial differential equation involving the iterated Laplacian or the polyharmonic operator. The solution is implementable in closed-form via polyharmonic radial basis function (RBF) interpolation. In view of the polyharmonic connection, we refer to the corresponding GANs as Poly-LSGAN and Poly-WGAN. As a proof of concept, the analysis is supported by experimental validation on multivariate Gaussians. While the closed-form RBF does not scale favorably with the dimensionality of data for image-space generation, we employ the Poly-WGAN discriminator to transform the latent space distribution of the data to match a Gaussian in a Wasserstein autoencoder (WAE). The closed-form discriminator, motivated by the polyharmonic RBF, results in up to 20% improvement in terms of Fréchet and kernel inception distances over comparable baselines that employ trainable or kernel-based discriminators. The experiments are carried out on standard image datasets such as MNIST, CIFAR-10, CelebA, and LSUN-Churches. The training time in Poly-WGAN is comparable to those of kernel-based methods, while being about two orders faster than GANs with a trainable discriminator.

## 1 Introduction

Generative adversarial networks (GANs) (Goodfellow et al., 2014) constitute a two players game between a generator $G$ and a discriminator $D$. The generator $G$ accepts high-dimensional Gaussian noise as input and learns a transformation (by means of a network), whose output follows the distribution $p_g$. The generator is tasked with learning $p_d$, the distribution of the target dataset. The discriminator learns a classifier between the samples of $p_d$ and $p_g$. The optimization in the standard GAN (SGAN) formulation of Goodfellow et al. (2014), and subsequent variants such as the least-squares GAN (LSGAN) (Mao et al., 2017) or the $f$-GAN (Nowozin et al., 2016) corresponds to learning a discriminator that mimics a chosen divergence metric between $p_d$ and $p_g$ (such as the Jensen-Shannon divergence in SGAN) and a generator that minimizes the divergence.

**Integral Probability Metrics, Gradient Penalties and GANs**: The divergence metric approaches fail if $p_d$ and $p_g$ are of disjoint support (Arjovsky & Bottou, 2017), which shifted focus to *integral probability metrics* (IPMs), where a *critic* function is chosen to approximate a chosen IPM between the distributions (Arjovsky et al., 2017; Mroueh & Sercu, 2017; Bunne et al., 2019). Choosing the distance metric is equivalent to constraining the class of functions from which the critic is drawn. The most popular variant, inspired by optimal transport, is the Wasserstein GAN (WGAN) (Arjovsky et al., 2017), in which the objective is to minimize the Wasserstein-1 or *earth mover's* distance between $p_d$ and $p_g$, and the critic is constrained to be Lipschitz-1. Gulrajani et al. (2017) enforced a first-order gradient penalty on the discriminator network to approximate the Lipschitz constraint. Roth et al. (2017); Kodali et al. (2017); Fedus et al. (2018) and Mescheder et al. (2018) showed the empirical success of the first-order gradient penalty on other GAN variants, such as the SGAN or LSGAN, while Bellemare et al. (2017); Mroueh et al. (2018) and Adler & Lunz (2018) consider bounding the energy in the critic's gradients correspond to Sobolev constraint spaces.

**Kernel-based GANs**: Gretton et al. (2012) showed that the minimization of IPM losses linked to reproducing-kernel Hilbert space (RHKS) can be replaced equivalently with the minimization of kernel-based statistics. Based on this connection, Li et al. (2015) introduced generative moment matching networks (GMMNs) that minimize the the maximum-mean discrepancy (MMD) between the target and generator distributions using the RBF Gaussian (RBFG) and inverse multiquadric (IMQ) kernels. Li et al. (2017a) extended the GMMN formulation to MMD-GANs, wherein a network leans lower-dimensional embedding of the data, over which the MMD is computed. Bińkowski et al. (2018) and Arbel et al. (2018) have also incorporated gradient-based regularizers in MMD-GANs, while Wang et al. (2019) enforce a repulsive loss formulation to stabilize training. Closed-form approaches such as GMMNs benefit from stable convergence of the generator, brought about by the lack of adversarial training. A series of works by Li et al. (2017b); Zhang et al. (2018); Daskalakis et al. (2018) and Wu et al. (2020) have shown that employing the optimal discriminator in each step improves and stabilizes the generator training, while Pinetz et al. (2018); Korotin et al. (2022) showed that in most practical settings, the networks in GAN do not accurately learn the desired divergences or IPMs.

## 1.1 Our Motivation

In this paper, we strengthen the understanding of the optimal GAN discriminator by drawing connections between IPM- and divergence-based GANs, kernel-based discriminators, and high-dimensional interpolation. As shown by Arjovsky & Bottou (2017), divergence minimizing GANs suffer from vanishing gradients when $p_d$ and $p_g$ are non-overlapping. The GAN discriminator can be viewed as a two-class classifier, which learns a decision boundary between the reals and the fakes. However, as the generator optimization progresses, the generated samples and target samples get interspersed, causing multiple transitions in the discriminator. This severely impacts training due to lack of smooth gradients (Arjovsky et al., 2017). As observed by Rosca et al. (2020), gradient-based regularizers enforced on the discriminator provide a trade-off between the accuracy in classification and smoothness of the learnt discriminator.

The WGAN discriminator can be seen as assigning a positive value to the reals and a negative value to the fakes. Given an unseen sample $\boldsymbol{x}$, the output of a *smooth* discriminator should ideally depend on the values assigned to the points in the neighborhood of $\boldsymbol{x}$, which is precisely what kernel based interpolation achieves. Recently, Franceschi et al. (2022) and Zhang et al. (2022) have shown that neural networks can be interpreted as high-dimensional interpolators involving *neural tangent kernels*. In general, gradient-norm regularizers result in smooth interpolators, thereby giving rise to the well-known family of *thin-plate splines* in 2-D (Harder & Desmarais, 1972; Meinguet, 1979; Bookstein, 1989; Wahba, 1990; Bogacz et al., 2019). A natural extension to these interpolators, in a high-dimensional setting, comes in the form of higher-order gradient regularization (Duchon, 1977). It is known that the optimization of the interpolant with its higher-order derivatives bounded in $L_2$ norm has a unique solution (Duchon, 1977), which has led to successful application of higher-order gradient regularization in image processing tasks such as image interpolation (Tirosh et al., 2006) and super-resolution (Ren et al., 2013). **What are the implications of reformulating the gradient-regularized GAN optimization problem as one of solving a high-dimensional interpolation? What insights does it give about the optimal GAN discriminator?** — These are the questions that we seek to answer in this paper. While the first-order penalty has been extensively explored in GAN optimization, higher-order penalties and their effect on the learnt discriminator have not been rigorously analyzed. We establish the connection between higher-order gradient regularization of the discriminator and interpolation in LSGAN and WGAN. The most closely related work is that of Adler & Lunz (2018), where the Sobolev GAN cost evaluated in the Fourier domain is used to train a discriminator.

## 1.2 The Proposed Approach

This current work extends significantly upon the results developed as part of the **non-archival workshop preprint** Anonymous (2022), wherein we considered the optimization of the least-squares GAN (LSGAN) discriminator cost, subject to $L_2$-norm regularization on the $m^{th}$-order gradients, from a variational calculus standpoint (Section 3). Our analysis shows that the optimal LSGAN discriminator in the proposed framework involves a polyharmonic radial basis function (RBF) kernel-based interpolator. The proposed approach, referred to as *Poly-LSGAN*, can be implemented via an RBF network for the optimal discriminator whose

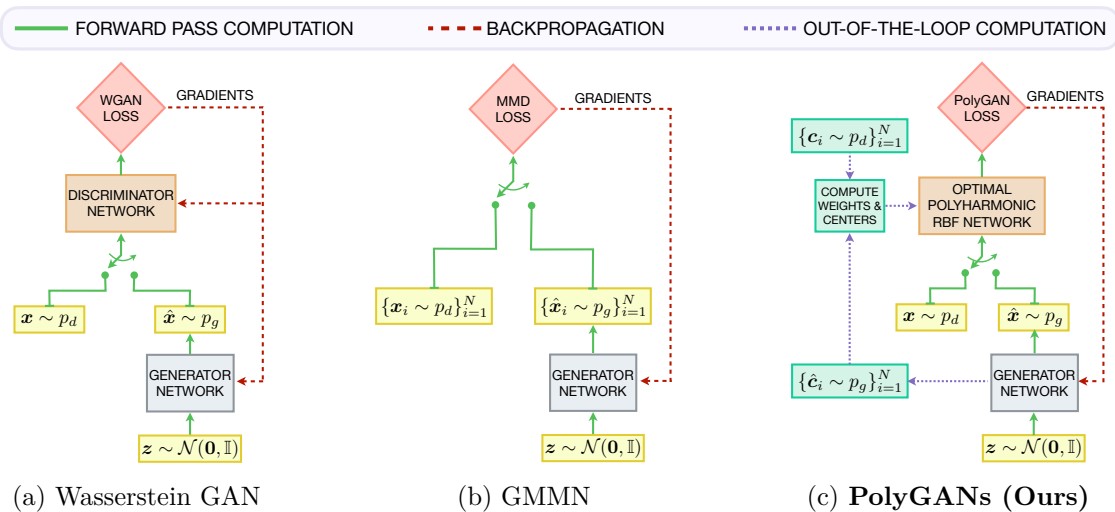

Figure 1: A comparison of generative model architectures. (a) WGAN (Arjovsky et al., 2017) trains discriminator and generator networks with the Wasserstein loss; (b) A generative moment matching network (GMMN) (Li et al., 2015) trains the generator with the closed-form maximum-mean discrepancy loss computed on a batch of samples; and (c) The proposed Poly-LSGAN/Poly-WGAN architecture uses a polyharmonic radial basis function discriminator whose weights and centers are computed based on batches of samples, whereas the generator is trained employing standard GAN losses.

weights can be computed by solving a system of linear equations. We show experimentally that the Poly-LSGAN algorithm does not scale favorably with the dimensionality of the data, owing to a combinatorial explosion in the number of coefficients, and singularity issues in solving for the weights of the RBF (Section 3.1).

To circumvent the issues in Poly-LSGAN (Anonymous, 2022), we consider the WGAN-IPM discriminator loss subjected to the $m^{th}$-order gradient regularizer, in a Lagrangian formulation. The resulting formulation, referred to as *Poly-WGAN*, does not require solving a system of equations to compute the weights. We show that this formulation of LSGAN and WGAN is equivalent to solving the GAN optimization with the discriminator constrained to belong to the *Beppo-Levi space*, which is a semi-normed/pseudo-metric space (Section 4). While standard GANs train a network to approximate a chosen metric, kernel based approaches leverage the underlying RKHS structure, from which the norms and MMD kernels are selected. However, as the discriminator is drawn from a semi-normed space, the optimum cannot be readily connected with a kernel-mean statistic, but rather must be determined using *Calculus of Variations*. Across both Poly-LSGAN and Poly-WGAN (jointly called the *PolyGAN* variants), our analysis shows that the optimal discriminators are the solution to elliptic partial differential equations (PDEs), more specifically, the iterated Laplacian/polyharmonic PDE (Section 4.2, Appendices B&C). The closed-form Poly-WGAN discriminator can be represented as an interpolator using the polyharmonic RBF kernel, which we implement through the RBF network approximation with predetermined weights and centers (Section 4.3).

Figure 1 compares WGAN, GMMN, and PolyGAN variants. We show that Poly-WGAN outperforms the baselines in terms of training stability and convergence on multivariate Gaussian and Gaussian mixture learning (Section 6 and Appendix E). Although Poly-WGANs scale favorably in comparison to Poly-LSGAN, the sample-complexity of the RBF-based discriminator remains challenging on high-dimensional image dataset. As a proof-of-concept, we apply Poly-WGAN to latent-space matching with the Wasserstein autoencoder (WAE) (Tolstikhin et al., 2018) (Section 7) and MMD-GAN (Li et al., 2017a) architectures (Appendix F), and show that the proposed RBF discriminator achieves performance improvements over comparable WAE variants in terms of various standard metrics on MNIST (LeCun et al., 1998), CIFAR-10 (Krizhevsky, 2009), CelebA (Liu et al., 2015) and LSUN-Churches (Yu et al., 2016) datasets. The emphasis in the proposed PolyGAN formulation is less on outperforming the state of the art (Karras et al., 2019; 2020; 2021), and more on gaining a deeper understanding of the underlying optimal discriminator in gradient-regularized GANs, opening up new avenues in generative modeling.

## 2 LSGAN, WGAN and Gradient Penalties

Mao et al. (2017) considered the GAN learning problem where the discriminator and generator networks minimize the least-squares loss. To mimic the classifier nature of the standard GAN (Goodfellow et al., 2014), an $a - b$ coding scheme is used, where $a$ and $b$ are the class labels of the generated samples and target data samples, respectively. On the other hand, the generator is trained to generate samples that are assigned a class label $c$ by the discriminator. The resulting formulation is as follows:

$$\mathcal{L}_D^{\text{LS}} = \frac{1}{2}\,\mathbb{E}_{\boldsymbol{x} \sim p_d}[(D(\boldsymbol{x}) - b)^2] + \frac{1}{2}\,\mathbb{E}_{\boldsymbol{x} \sim p_g}[(D(\boldsymbol{x}) - a)^2]; \qquad D^*(\boldsymbol{x}) = \arg\min_D \mathcal{L}_D^{\text{LS}},$$

$$\text{and } \mathcal{L}_G^{\text{LS}} = \frac{1}{2}\,\mathbb{E}_{\boldsymbol{x} \sim p_g}[(D^*(\boldsymbol{x}) - c)^2]; \qquad p_g^*(\boldsymbol{x}) = \arg\min_{p_g} \mathcal{L}_G^{\text{LS}},$$

where $\mathbb{E}$ denotes the expectation operator. While Mao et al. (2017) show that setting $b - a = 2$ and $c - a = 1$ lead to the generator minimizing the Pearson-$\chi^2$ divergence, a more intuitive approach is to set $c = b$, which enforces the generator to output samples that are classified as *real* by the discriminator.

Along a parallel vertical, Arjovsky et al. (2017) presented the GAN learning problem as one of *optimal transport*, wherein the critic (or discriminator) minimizes the earth mover's distance or Wasserstein-1 distance between $p_d$ and $p_g$. Through the Kantorovich–Rubinstein duality, they defined the WGAN discriminator and generator losses as follows:

$$\mathcal{L}_D^W = \mathbb{E}_{\boldsymbol{x} \sim p_g}[D^{\text{L}}(\boldsymbol{x})] - \mathbb{E}_{\boldsymbol{x} \sim p_d}[D^{\text{L}}(\boldsymbol{x})], \text{ and } \mathcal{L}_G^W = -\mathcal{L}_D^W,$$

where $D^{\text{L}}(\boldsymbol{x})$ denoted a Lipschitz-1 discriminator. While $\mathcal{L}_D^W$ was first introduced in the context of WGANs, it forms the basis for all IPM based GANs. Arjovsky et al. (2017) clip the discriminator weights to enforce the Lipschitz-1 constraint. Gulrajani et al. (2017) consider the WGAN with a gradient penalty $\mathbb{E}_{\boldsymbol{x} \sim p_{\text{int}}}\left[(\|\nabla D(\boldsymbol{x})\|_2 - 1)^2\right]$, where $p_{\text{int}}$ is an interpolated distribution between $p_d$ and $p_g$. As noted by Rosca et al. (2020), in general, the gradient penalties in GANs have the form $\Omega_D^{GP} : \mathbb{E}_{\boldsymbol{x} \sim p_r}\left[(\|\nabla D(\boldsymbol{x})\|_2 - \text{K})^2\right]$, where $p_r$ is the reference density and K is a suitable constant. Setting K $> 0$ enforces a Lipschitz constraint, while K $= 0$ promotes the smoothness of the learnt discriminator. For example, Kodali et al. (2017) employed $p_d * \mathcal{N}(\boldsymbol{0}, \mathbb{I})$, while Mescheder et al. (2018) showed that the regularizer $\Omega_D^R = \mathbb{E}_{\boldsymbol{x} \sim p_r}\left[\|\nabla D(\boldsymbol{x})\|_2^2\right]$ with either $p_r = p_d$ or $p_r = p_g$ (called WGAN-R$_d$ and WGAN-R$_g$, respectively) guarantees local convergence even in the case of discontinuous distributions. Kodali et al. (2017) also show that the gradient regularization improves the empirical performance of LSGAN and other $f$-GAN variants. Mroueh et al. (2018) considered a generalization of $\Omega_D^R$ for any choice of $p_r$ defined over $\mathbb{R}^n$, giving rise to the class of Sobolev GANs. Petzka et al. (2018) proposed WGAN with Lipschitz penalty (WGAN-LP), which applies a hinge-loss variant of the gradient penalty, while Terjék (2020) proposed WGAN with adversarial LP (WGAN-ALP) to compute the gradient penalty along the direction of maximum error. Anonymous (2023) considered the first-order gradient-norm penalty, but obtained a closed-form expression for the discriminator given the generator for the case when $p_r(\boldsymbol{x})$ is the uniform measure. They employ a Fourier-series approximation for the discriminator and do away with training a discriminator neural network. Adler & Lunz (2018) implemented $m^{th}$-order generalizations of the cost empirically through a Fourier representation of the cost, but do not explore the theoretical optimum in these scenarios.

We now consider the LSGAN cost subject to high-order gradient regularization, and show that the discriminator optimization is one of high-dimensional interpolation. Helpful background on higher-order derivatives and the *Calculus of Variations* is provided in Appendix A.

## 3 Regularized LSGAN and Least-squares Interpolation

We consider the $m^{th}$-order generalization of the gradient regularizer considered by Mroueh et al. (2018) and Anonymous (2023). The penalty is enforced uniformly for all values of $\boldsymbol{x} \in \mathcal{X}$, which is the convex hull of the supports of $p_d$ and $p_g$ (*i.e.*, we set $p_r$ to be the uniform density over $\mathcal{X}$). This can be viewed as

interpolating over infinitely many samples drawn from $p_d$ and $p_g$. The regularizer is then given by

$$\Omega_D = \left( \frac{1}{|\mathcal{X}|} \int_{\mathcal{X}} \|\nabla^m D(\boldsymbol{x})\|_2^2 \, d\boldsymbol{x} - K \right), \tag{1}$$

where $\|\nabla^m D(\boldsymbol{x})\|_2^2$ is the square of the norm of the $m^{th}$-order gradient vector (cf. Eq. (13), Appendix A), and $|\mathcal{X}|$ denotes the volume of $\mathcal{X}$. Consistent with the literature on high-dimensional interpolation (Duchon, 1977; Meinguet, 1979) and IPM-GANs (Mescheder et al., 2018; Mroueh et al., 2018), setting $K = 0$ promotes smoothness of the learnt discriminator, thereby accelerating convergence of the training algorithm. The corresponding regularized LSGAN cost is given by

$$\mathcal{L}_D^{\text{Poly}-\text{LS}} = \frac{1}{2} \, \mathbb{E}_{\boldsymbol{x} \sim p_d}[(D(\boldsymbol{x}) - b)^2] + \frac{1}{2} \, \mathbb{E}_{\boldsymbol{x} \sim p_g}[(D(\boldsymbol{x}) - a)^2] + \lambda_d \int_{\mathcal{X}} \|\nabla^m D(\boldsymbol{x})\|_2^2 \, d\boldsymbol{x}, \tag{2}$$

where $\lambda_d \geq 0$ is the Lagrange multiplier associated with the gradient penalty. When $K = 0$, the regularization of the LSGAN cost with $\Omega_D$ can be viewed as restricting the solution space to the *Beppo-Levi* space $\text{BL}^{m,p}$, comprising all functions defined over $\mathbb{R}^n$, with $m^{th}$-order gradients having finite $L_p$-norm. A more detailed discussion on drawing the discriminator from $\text{BL}^{m,p}$ is provided in Section 4.1.

Consider an $N$-sample approximation of $\mathcal{L}_D^{\text{LS}}$ in Equation (2), where $N_B$ samples are drawn from $p_d$ and $p_g$ each (therefore, $N = 2N_B$), represented by the dataset batch

$$\mathcal{D} = \left\{ (\boldsymbol{c}_i, y_i) \right\}_{i=1}^N = \left\{ (\boldsymbol{x}_i, b) \mid \boldsymbol{x}_i \sim p_d \right\}_{i=1}^{N_b} \bigcup \left\{ (\boldsymbol{x}_j, a) \mid \boldsymbol{x}_j \sim p_g \right\}_{j=1}^{N_b}.$$

The corresponding discriminator optimization problem can be formulated as follows:

$$D^* = \arg\min_D \sum_{\substack{i=1 \\ (\boldsymbol{c}_i, y_i) \sim \mathcal{D}}}^N \left( D(\boldsymbol{c}_i) - y_i \right)^2 + \lambda_d \int_{\mathcal{X}} \|\nabla^m D(\boldsymbol{x})\|_2^2 \, d\boldsymbol{x}. \tag{3}$$

The above represents a regularized least-squares interpolation problem. When $\lambda_d = 0$, the optimum $D^*$ is an interpolator that passes through the target points $(\boldsymbol{c}_i, y_i)$ exactly. On the other hand, for positive values of $\lambda_d$, the minimization leads to smoother solutions, penalizing sharp transitions in the discriminator. We found out experimentally that $\lambda_d = 10$ results in superior performance. A smoother discriminator allows for more efficient training of the generator (Li et al., 2017b; Xu et al., 2018). The following theorem shows the interpolating nature of the optimal discriminator.

**Theorem 3.1.** *The **optimal LSGAN discriminator** that minimizes the cost given in Eqn. (3) is*

$$D^*(\boldsymbol{x}) = \sum_{\substack{1=i \\ (\boldsymbol{c}_i, y_i) \sim \mathcal{D}}}^N w_i \varphi_k \left( \|\boldsymbol{x} - \boldsymbol{c}_i\| \right) + P(\boldsymbol{x}; \boldsymbol{v}), \quad \text{where} \quad \varphi_k(\boldsymbol{x}) = \begin{cases} \|\boldsymbol{x}\|^k & \text{for} \quad k = 1, 3, 5, \cdots \\ \|\boldsymbol{x}\|^k \ln(\|\boldsymbol{x}\|) & \text{for} \quad k = 2, 4, 6, \cdots \end{cases} \tag{4}$$

*is the polyharmonic radial basis function with the spline order $k = 2m - n$ for a gradient order $m$, such that $k > 0$, $P(\boldsymbol{x}; \boldsymbol{v}) \in \mathcal{P}_{m-1}^n$ is an $(m-1)^{th}$ order polynomial parametrized by the coefficients $\boldsymbol{v} \in \mathbb{R}^L$; $L = \binom{n+m-1}{m-1}$, $\boldsymbol{x} \in \mathcal{X} \subseteq \mathbb{R}^n$, $\mathcal{D} = \{(\boldsymbol{c}_i, y_i)\}$ is the set of real and fake centers about which the polyharmonic RBFs $\varphi_k(\|\cdot\|)$ are localized, $\|\cdot\|$ denotes the $\ell_2$ norm. The $N$ weights $\boldsymbol{w} = [w_1, w_2, \ldots, w_N]^{\text{T}}$ and $L$ polynomial coefficients $\boldsymbol{v} = [v_1, v_2, \ldots, v_L]^{\text{T}}$ can be obtained by solving the linear system of equations:*

$$\begin{bmatrix} \mathbf{A} + (-1)^m \lambda_d C_k \mathbf{I} & \mathbf{B} \\ \mathbf{B}^{\text{T}} & \mathbf{0} \end{bmatrix} \begin{bmatrix} \boldsymbol{w} \\ \boldsymbol{v} \end{bmatrix} = \begin{bmatrix} \boldsymbol{y} \\ \mathbf{0} \end{bmatrix}, \tag{5}$$

*where $[\mathbf{A}]_{i,j} = \varphi_k(\|\boldsymbol{c}_i - \boldsymbol{c}_j\|)$, $\mathbf{B} = \begin{bmatrix} 1 & 1 & \cdots & 1 \\ \boldsymbol{c}_1 & \boldsymbol{c}_2 & \cdots & \boldsymbol{c}_N \\ \vdots & \vdots & \ddots & \vdots \\ \boldsymbol{c}_1^{m-1} & \boldsymbol{c}_2^{m-1} & \cdots & \boldsymbol{c}_N^{m-1} \end{bmatrix}^{\text{T}}$, and $\boldsymbol{y} = [y_1, y_2, \cdots, y_N]^{\text{T}}$,*

**I** *is the $N \times N$ identity matrix, and $\boldsymbol{c}_i^j$ is a vectorized representation of all the terms of the $j^{th}$-order polynomial of $\boldsymbol{c}_i$, and $\mathrm{C}_k$ is a constant that depends only on the order $k$. The above system of equations has a unique solution iff the kernel matrix $\mathbf{A}$ is invertible and $\mathbf{B}$ is full column-rank. Matrix $\mathbf{A}$ is invertible if the set of real/fake centers are unique, and the kernel order $k$ is positive. The matrix $\mathbf{B}$ is full rank if the set of centers $\{\boldsymbol{c}_i\}$ are linearly independent, and more specifically, do not lie on any subspace of $\mathbb{R}^n$ (Iske, 2004).*

*Proof.* The proof follows by applying the Euler-Lagrange equation from the *Calculus of Variations* to the cost in Equation (3), which yields the following differential equation that the optimal discriminator satisfies:

$$\left( \sum_{i=1}^{N} (D(\boldsymbol{x}) - y_i)\delta(\boldsymbol{x} - \boldsymbol{c}_i) \right) + (-1)^m \lambda_d \Delta^m D(\boldsymbol{x}) \big|_{D=D^*(\boldsymbol{x})} = 0$$

The solution to the above PDE constitutes a polyharmonic sum which is the particular solution, and the polynomial component which represents the homogeneous component, *i.e.,* solutions to $\Delta^m f(\boldsymbol{x}) = 0$ (Iske, 2004). Substituting the optimal discriminator into the loss in Equation 3 yields the family of equations that the optimal weights and polynomial coefficients satisfy. The details are provided in Appendix B. For $k \leq 0$, the system of equations does not have a solution, as $[\mathbf{A}]_{i,i} \to \infty$. Owing to the polyharmonic radial basis kernel, the proposed approach is referred to as *Poly-LSGAN*. The solution is applicable for all $\boldsymbol{x} \in \mathcal{X}$. Outside of the domain, the loss vanishes, obviating the need for optimization. $\square$

### 3.1 Experimental Limitations of Poly-LSGAN

We evaluate the optimal Poly-LSGAN discriminator for learning synthetic 2-D Gaussian and Gaussian mixture models (GMMs), and subsequently discuss extensions to handle images. Detailed discussions are provided in Appendices E.1 and F.1. We provide only a summary of the observations here. On low-dimensional Gaussian learning tasks, using the polyharmonic RBF discriminator results in superior generator performance (lower $\mathcal{W}^{2,2}$ scores). However, the Poly-LSGAN algorithm does not scale well with the dimensionality of the data for image-space learning on datasets such as MNIST, Fashion-MNIST and CelebA. To illustrate the limitation, we consider the polyharmonic spline of order $k = 2$ for learning 784-dimensional MNIST data. This requires a $\frac{784+2}{2} - 1 = 392^{\text{nd}}$-degree polynomial consisting of $\mathcal{O}(10^{323})$ coefficients! In general, given $N$ centers in $\mathbb{R}^n$ and gradient order $m$, solving for the weights and coefficients requires inverting a matrix of size $M = N + \binom{n+m-1}{m-1}$, which requires $\mathcal{O}(M^3)$ computations. For example, given a batch size of $N = 100$ and data in $\mathbb{R}^{128}$, we have $M \approx \mathcal{O}(10^4)$ for $m = 3$, and $M \approx \mathcal{O}(10^5)$ for $m = 4$. However, as the Poly-LSGAN solution is only valid for $k > 0$ or $m > \lceil \frac{n}{2} \rceil$ (cf. Appendix C.5), the problem becomes intractable, with $M \approx \mathcal{O}(10^{51})$ or higher! We therefore restrict the solution to include only $3^{rd}$ order polynomials. Although clearly sub-optimal, this work-around results in an implementable solution.

The results of training Poly-LSGAN on image datasets are discussed in Appendix F.1. While the underlying structure is learnt, the generated images are far from being realistic and below par compared with standard GAN results. Poly-LSGAN failed to converge as the matrix $\mathbf{B}$ turned out to be rank-deficient. As noted in the literature on mesh-free interpolation (Iske, 2004), $\mathbf{B}$ must be full column-rank for the system of equations (Eq. (5)) to have a unique solution. This requires the centers $\boldsymbol{c}_i$ to not lie on a subspace/manifold of $\mathbb{R}^n$. However, from the manifold hypothesis (Kelley, 2017; Vershynin, 2018), we know that structured image datasets lie precisely in such low-dimensional manifolds. One possible workaround is to avoid training GANs on images, and instead perform adversarial score matching (Jolicoeur-Martineau et al., 2021). Yet another approach is to not compute the weights through matrix inversion. In the remainder of this paper, we consider the latter approach, wherein we enforce the higher-order gradient constraint on the Wasserstein GAN cost. Through a variational analysis, we show that even in this setting, the links to high-dimensional interpolation hold, but the weights can be computed without the need for solving a system of equations. We therefore focus on WGAN with the higher-order gradient-norm constraint in the remainder of the paper.

# 4 WGAN with Higher-order Gradient Regularization

We consider the WGAN-IPM loss, with the $m^{th}$-order gradient-norm regularizer $\Omega_D$ (cf. Equation (1)). The resulting Lagrangian of the discriminator cost is given by:

$$\mathcal{L}_D^{\text{Poly}-\text{W}} = \mathbb{E}_{\boldsymbol{x} \sim p_g}[D(\boldsymbol{x})] - \mathbb{E}_{\boldsymbol{x} \sim p_d}[D(\boldsymbol{x})] + \lambda_d \left( \int_{\mathcal{X}} \|\nabla^m D(\boldsymbol{x})\|_2^2 \, \mathrm{d}\boldsymbol{x} - \mathrm{K}|\mathcal{X}| \right) \tag{6}$$

$$= \int_{\mathcal{X}} \underbrace{D(\boldsymbol{x}) \left( p_g(\boldsymbol{x}) - p_d(\boldsymbol{x}) \right) + \lambda_d \left( \|\nabla^m D(\boldsymbol{x})\|_2^2 - \mathrm{K} \right)}_{\mathcal{F}(D, \partial^{\boldsymbol{\alpha}} D; |\boldsymbol{\alpha}| = m)} \, \mathrm{d}\boldsymbol{x}, \tag{7}$$

where $\lambda_d$ is the Lagrange multiplier associated with $\Omega_D$, which is optimized as a dual variable. We show in Appendix C.2 that the choice of K simply scales the optimal dual variable $\lambda_d^*$ by a factor of $\frac{1}{\sqrt{\mathrm{K}}}$, but the optimal generator distribution $p_g^*(\boldsymbol{x})$ remains unaffected. Therefore, to maintain consistency with the LSGAN formulation, we consider $\mathrm{K} = 0$ in the remainder of this paper, while the generalization to positive K is discussed in Appendix C.2. Before proceeding with the optimality of Poly-WGANs, we discuss the implications of the chosen regularizer on the constraint space of the discriminator.

## 4.1 Constraint Space of the Discriminator

Both the Poly-LSGAN and the Poly-WGAN discriminator functions are solutions to gradient-regularized optimization problems. The Poly-LSGAN optimization results in discriminator functions that are *sufficiently smooth* (large $m$) and interpolate between the positive and negative class labels. On the other hand, the *smooth* Poly-WGAN discriminator can be seen as approximating large positive values corresponding to the reals, and large negative values corresponding to the fakes.

In both PolyGANs, the optimization problem can be interpreted as restricting solutions to belong to the *Beppo-Levi* space $\mathrm{BL}^{m,p}$, endowed with the semi-norm $\|D\|_{\mathrm{BL}^{m,p}} = \|\nabla^m D(\boldsymbol{x})\|_{\mathrm{L}_p}$. The $m^{th}$-order gradient penalty considered in Eq. (6) corresponds to $\mathrm{BL}^{m,2}$. Unlike a norm, the semi-norm does not satisfy the point-separation property, *i.e.,* $\|D\|_{\mathrm{BL}^{m,p}} = 0 \not\Rightarrow D = 0$. Contrast this with the Sobolev space $\mathrm{W}^{m,p}$, which comprises all functions with finite $\mathrm{L}_p$-norms of the gradients *up to* order $m$, endowed with the norm $\|D\|_{\mathrm{W}^{m,p}} = \sum_{k=0}^{m} \|\nabla^k D(\boldsymbol{x})\|_{\mathrm{L}_p}$. The Sobolev space $\mathrm{W}^{m,p}$ is a Banach space, and for the case of $p = 2$, it is a Hilbert space. The null-space of the Beppo-Levi semi-norm comprises all $(m-1)$-degree polynomials defined over $\mathbb{R}^n$, denoted by $\mathcal{P}_{m-1}^n(\boldsymbol{x})$. The Sobolev semi-norm considered by Mroueh et al. (2018) is the first-order Beppo-Levi semi-norm. Adler & Lunz (2018) consider Sobolev spaces in Banach WGAN and implement the loss through a Bessel potential approach, relying on a Fourier transform of the loss. They provide experimental results, but an in-depth analysis of the discriminator optimization is lacking. We optimize the GAN loss defined in Eq. (6) within a variational framework and choose Beppo-Levi $\mathrm{BL}^{m,2}$ as the constraint space and provide a closed-form solution for the optimal discriminator. Our approach also highlights the interplay between the gradient order $m$ and the dimensionality of the data $n$, and its influence on the performance of the GAN.

## 4.2 The Optimal Poly-WGAN Discriminator and Generator

Consider the integral form of the **discriminator loss** given in Eq. (6). The following Theorem gives us the optimal Poly-WGAN discriminator.

**Theorem 4.1.** *The **optimal discriminator** that minimizes the loss $\mathcal{L}_D$ is a solution to the following PDE:*

$$\Delta^m D(\boldsymbol{x}) = \frac{(-1)^{m+1}}{2\lambda_d} \left( p_g(\boldsymbol{x}) - p_d(\boldsymbol{x}) \right), \ \forall \ \boldsymbol{x} \in \mathcal{X}, \tag{8}$$

*where $\Delta^m$ is the polyharmonic operator of order $m$. The particular solution $D_p^*(\boldsymbol{x})$ is given by*

$$D_p^*(\boldsymbol{x}) = \frac{(-1)^{m+1}}{2\lambda_d} \left( (p_g - p_d) * \psi_{2m-n} \right)(\boldsymbol{x}), \tag{9}$$

*which is a multidimensional convolution with the polyharmonic radial basis function $\psi_{2m-n}(\boldsymbol{x})$, which in turn is the fundamental solution to the polyharmonic equation: $\Delta^m \varrho \psi_{2m-n}(\boldsymbol{x}) = \delta(\boldsymbol{x})$, for some constant $\varrho$, and is given by*

$$\psi_{2m-n}(\boldsymbol{x}) = \begin{cases} \|\boldsymbol{x}\|^{2m-n} & \text{if } 2m-n < 0 \text{ or } n \text{ is odd,} \\ \|\boldsymbol{x}\|^{2m-n} \ln(\|\boldsymbol{x}\|) & \text{if } 2m-n \geq 0 \text{ and } n \text{ is even.} \end{cases} \tag{10}$$

*The general solution $D^*(\boldsymbol{x})$ is given by $D^*(\boldsymbol{x}) = D_p^*(\boldsymbol{x}) + P(\boldsymbol{x})$, where $P(\boldsymbol{x}) \in \mathcal{P}_{m-1}^n$, which is the space of all $(m-1)^{th}$ order polynomials defined over $\mathbb{R}^n$.*

*Proof.* From the integrand $\mathcal{F}$ in Eq. (7), we have

$$\frac{\partial \mathcal{F}}{\partial D} = p_g(\boldsymbol{x}) - p_d(\boldsymbol{x}), \qquad \text{and} \qquad (-1)^m \sum_{\boldsymbol{\alpha}:|\boldsymbol{\alpha}|=m} \partial^{\boldsymbol{\alpha}} \left( \frac{\partial \mathcal{F}}{\partial(\partial^{\boldsymbol{\alpha}} D)} \right) = (-1)^m 2\lambda_d \Delta^m D.$$

Substituting the above in the Euler-Lagrange condition from the *Calculus of Variations* (Eq. (14)) results in the PDE stated in Eq. (8). The solution to the PDE can be obtained in terms of the solution to the inhomogeneous equation $\Delta^m f(\boldsymbol{x}) = \delta(\boldsymbol{x})$. PDEs of this type have been extensively researched. The book on *Polyharmonic Functions* by Aronszajn et al. (1983) is an authoritative reference on the topic. It has been shown that $\psi_{2m-n}(\boldsymbol{x})$ defined in Eq. (10) is the fundamental solution, up to a constant $\varrho$. Convolving both sides of Eq. (8) with $\psi_{2m-n}(\boldsymbol{x})$ yields Eq. (9). The value of $\varrho$ for various $m$ and $n$ is given by in Appendix C.1.

Equation 9 provides the particular solution to the PDE governing the discriminator. As in the case of Poly-LSGAN, the general solution also includes the homogeneous component. The homogeneous component belongs to the null-space $\mathcal{P}_{m-1}^n$ of the Beppo-Levi semi-norm. The general solution to the discriminator is $D^*(\boldsymbol{x}) = D_p^*(\boldsymbol{x}) + P(\boldsymbol{x})$, where $P(\boldsymbol{x}) \in \mathcal{P}_{m-1}^n$. The exact choice of the polynomial depends on the boundary conditions and will be discussed in Appendix C.5. $\qquad \square$

The optimal Lagrange multiplier $\lambda_d^*$ can be determined by solving the dual optimization problem. A discussion is provided in Appendix C.2. The polyharmonic function $\psi_{2m-n}$ can be seen as an extension of Poly-LSGAN kernel $\varphi_k$ that permits negative orders. Since the optimal discriminator does not require any weight computation, the associated singularity of the kernel matrix can be ignored. The optimal GAN discriminator defined in WGAN-FS (Anonymous, 2023) and Sobolev GANs (Mroueh et al., 2018) are a special case of Theorem 4.2 for $m = 1$.

Obtaining the optimal discriminator is only one-half of the problem, with the optimal generator constituting the other half. In baseline GANs, the discriminator can be interpreted as approximating the divergence or IPM between distributions. Consequently, the generator is known to minimize the corresponding divergence, or distance function between distributions, and therefore, the optimum is attained when the two distributions match. However, in PolyGANs, the discriminator does not correspond to an IPM, as the Beppo-Levi space is a semi-normed space with a null-space component. Therefore, it must be shown that a generator that minimizes a loss employing the Poly-WGAN discriminator indeed results in the desired convergence of the generator distribution to that of the target. Although in practice, the push-forward distribution of the generator is well-defined, it remains to be shown that training the generator in PolyGANs indeed results in the generator distribution approaching the target. As a mathematical safeguard, we incorporate constraints to ensure that the learnt function is indeed a valid distribution, along the lines of Anonymous (2023). In particular, we consider the integral constraint $\Omega_p : \int_{\mathcal{X}} p_g(\boldsymbol{x}) d\boldsymbol{x} = 1$, and the point-wise non-negativity constraint $\Phi_p : p_g(\boldsymbol{x}) \geq 0, \forall \boldsymbol{x} \in \mathcal{X}$. While $\Omega_p$ readily fits into the Euler-Lagrange framework, $\Phi_p$ must be cast into an integral form with a point-wise Lagrange multiplier function $\mu_p(\boldsymbol{x}) : \mathcal{X} \to \mathbb{R}_-$, where $\mathbb{R}_-$ is the set of negative real numbers. Effectively, $\mu_p(\boldsymbol{x}) \leq 0 \; \forall \; \boldsymbol{x} \in \mathcal{X}$ (Gelfand & Fomin, 1964), which yields the Lagrangian of the **generator loss function**:

$$\mathcal{L}_G = \mathbb{E}_{\boldsymbol{x} \sim p_d}[D^*(\boldsymbol{x})] - \mathbb{E}_{\boldsymbol{x} \sim p_g}[D^*(\boldsymbol{x})] + \lambda_p \left( \int_{\mathcal{X}} p_g(\boldsymbol{x}) \, d\boldsymbol{x} - 1 \right) + \int_{\mathcal{X}} \mu_p(\boldsymbol{x}) p_g(\boldsymbol{x}) \, d\boldsymbol{x}, \tag{11}$$

where $\lambda_p \in \mathbb{R}$ and $\mu_p(\boldsymbol{x})$ are the Lagrange multipliers. The following theorem specifies the optimal generator density that minimizes $\mathcal{L}_G$ given the optimal discriminator.

**Theorem 4.2.** ***Optimal generator density:*** *Consider the minimization of the generator loss $\mathcal{L}_G$. The optimal generator density is given by $p_g^*(\boldsymbol{x}) = p_d(\boldsymbol{x})$, $\forall \; \boldsymbol{x} \in \mathcal{X}$. The optimal Lagrange multipliers are*

$$\lambda_p^* \in \mathbb{R} \quad and \quad \mu_p^*(\boldsymbol{x}) = \begin{cases} 0, & \forall \; \boldsymbol{x} : \; p_d(\boldsymbol{x}) > 0, \\ Q(\boldsymbol{x}) \in \mathcal{P}_{m-1}^n(\boldsymbol{x}), & \forall \; \boldsymbol{x} : \; p_d(\boldsymbol{x}) = 0, \end{cases}$$

*respectively, where $Q(\boldsymbol{x})$ is a non-positive polynomial of degree $m-1$, i.e., $Q(\boldsymbol{x}) \leq 0 \; \forall \; \boldsymbol{x}$, such that $p_d(\boldsymbol{x}) = 0$. The solution is valid for all choices of the homogeneous component $P(\boldsymbol{x}) \in \mathcal{P}_{m-1}^n(\boldsymbol{x})$ in the optimal discriminator.*

*Proof.* As the cost function involves convolution terms, the Euler-Lagrange condition cannot be applied readily, and the optimum must be derived using the *Fundamental Lemma of Calculus of Variations* (Gelfand & Fomin, 1964), as in the case of WGAN-FS (Anonymous, 2023).

Consider the Lagrangian of the generator loss $\mathcal{L}_G$. Enforcing the first-order necessary conditions for a minimizer of the cost yields the following equation that the optimum solution $p_g^*(\boldsymbol{x})$ satisfies the equation $p_g^*(\boldsymbol{x}) = p_d(\boldsymbol{x}) + \left(\frac{\lambda_d^*}{\xi}\right) \Delta^m \mu_p^*(\boldsymbol{x})$. It is clear from the above solution that the optimum, $p_g^*(\boldsymbol{x})$, does not depend on the choice of the homogeneous component $P(\boldsymbol{x})$ in the optimal discriminator. The optimal Lagrange multipliers can be determined through dual optimization and enforcing the complementary slackness condition to obtain the result in Theorem 4.2. The detailed proof is provided in Appendix C.3. $\square$

### 4.3 Practical Considerations

The closed-form discriminator in Equation (9) involves multidimensional convolution in a high-dimensional space. For instance, considering MNIST database with $28 \times 28$ size images, the convolution must be carried out in $\mathbb{R}^{784}$! Further, since closed-form expressions for $p_g$ and $p_d$ are not available in practice, the convolutions cannot be computed. A practical alternative is needed, for which we propose a sample approximation to $D^*(\boldsymbol{x})$, which also links well with other kernel-based generative models such as GMMNs. The following lemma presents an implementable form of the optimal discriminator.

**Theorem 4.3.** *The particular optimal discriminator $D_p^*(\boldsymbol{x})$ given in Eq. (9) can be approximated through the following sample estimate:*

$$\tilde{D}_p^*(\boldsymbol{x}) = \underbrace{\frac{\xi}{\lambda_d^* N} \sum_{\boldsymbol{c}_i \sim p_g} \psi_{2m-n}(\boldsymbol{x} - \boldsymbol{c}_i)}_{S_{p_g}} - \underbrace{\frac{\xi}{\lambda_d^* N} \sum_{\boldsymbol{c}_j \sim p_d} \psi_{2m-n}(\boldsymbol{x} - \boldsymbol{c}_j)}_{S_{p_d}} \tag{12}$$

*where $\psi_{2m-n}$ is the polyharmonic kernel, as described in Eq. 10.*

Theorem 4.3 shows that the sample approximation of Poly-WGAN discriminator can be implemented through an RBF network. The proof is given in Appendix C.4. Incorporating the homogeneous component in the solution becomes impractical in higher dimensions. However, by virtue of Theorem 4.2 and the first-order methods employed in updating the generator parameters, we argue that not incorporating the homogeneous component is not too detrimental to GAN optimization. Therefore, we set $P(\boldsymbol{x})$ to be the zero polynomial (*i.e.,* set $P(\boldsymbol{x}) = \boldsymbol{0}$). This argument is presented in Appendix C.5.

## 5 Interpreting The Optimal Discriminator in PolyGANs

Theorem 4.1 shows that, in gradient-regularized LSGAN and WGAN, the optimal discriminators that trainable neural networks learn to approximate are expressible as kernel-based convolutions. In particular, the gradient-norm penalty induces a polyharmonic kernel interpolator. By virtue of Theorems 3.1 and 4.2, this takes the form of a weighted sum of distance functions (for example, when $n$ is odd, we have $\sum_{i=1}^N \boldsymbol{w}_i \|\boldsymbol{x} - \boldsymbol{c}_i\|^{2m-n}$). While in the case of Poly-LSGAN, the weights must be computed by matrix inversion, in the case of Poly-WGAN, the analysis is tractable, because the weights reduce to $\pm \frac{\xi}{\lambda_d^* N}$ (cf. Equation (12)).

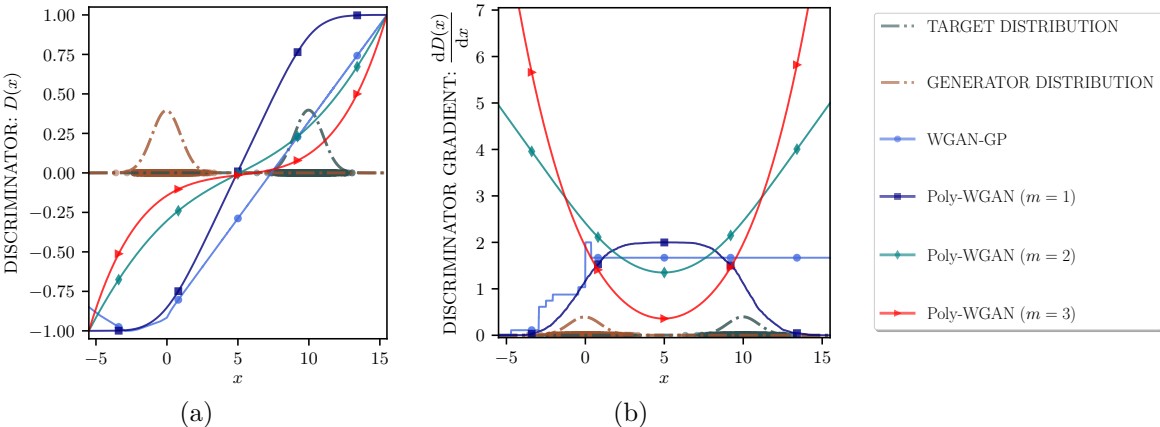

Figure 2: A comparison of the discriminator function and its gradient for WGAN-GP and Poly-WGAN for various choices of $m$, when the target and generator distributions are Gaussian. (a) The discriminator functions are normalized to $[-1, 1]$ to facilitate comparison. While the WGAN-GP discriminator is a three-layer feedforward network trained until convergence, which took about 50 iterations, the Poly-WGAN discriminator is computed in closed-form (Equations (12)). The Poly-WGAN discriminator for $m = 1$ is the optimal form of the discriminator learnt in WGAN-GP. Across all variants, the Poly-WGAN discriminator accurately changes in sign at the mid-point between the two distributions. (b) The unnormalized discriminator gradients illustrate the convergence speed-up observed in Poly-WGAN. The gradient magnitude increases with the penalty order. For orders $m \geq 5$, the generator training is unstable due to *exploding gradients.*

For order $2m - n < 0$, the optimal discriminator acts as an inverse-distance weighted (IDW) interpolator, where the centers closest to the sample $\boldsymbol{x}$ under evaluation have a stronger influence, while for $2m - n > 0$, the effect of the far-off centers is stronger. The latter is particularly helpful in pulling the generator distribution towards the target distribution when the two are far apart. Additionally, when $2m - n < 0$, the weights in Poly-LSGAN cannot be computed, as $\mathbf{A}$ is no longer a valid kernel matrix, while in Poly-WGANs, although tractable, the solution is singular at the target centers, causing training instability. As an illustration, Figure 2 presents the learnt discriminator, and its unnormalized gradient in the case of 1-D learning with WGAN-GP, and those implemented in Poly-WGAN for $m \in \{1, 2, 3\}$. While the WGAN-GP discriminator is a three-layer feedforward network trained until convergence, the Poly-WGAN discriminator is a closed-form RBF network. The discriminator functions are normalized to the range $[-1, 1]$ to facilitate visual comparison, but the gradients are presented unnormalized. For $n = 1$, the value of $2m - n$ is positive for all $m$. From Figure 2(b), we observe that the magnitude of the gradient increases with the gradient order, resulting in a stronger gradient for training the generator. We observed empirically (cf. Section 6.1) that this causes exploding gradients for large $m$, and in practice, the generator training is superior when the order $m \approx \frac{n}{2}$.

The discriminator function $\tilde{D}_p^*(\boldsymbol{x})$ comprises the difference between two RBF interpolations: $S_{p_d}$ operating entirely on the real data ($\boldsymbol{c}_i \sim p_d$), and $S_{p_g}$ operating on the fake ones ($\boldsymbol{c}_j \sim p_g$). For a test sample $\boldsymbol{x}$ drawn from the generator, the value of $S_{p_g}$ is smaller than $S_{p_d}$ with a high probability, and vice versa for samples drawn from $p_d$. A reasonable generator should output samples that result in a lower value for $S_{p_d}$ than $S_{p_g}$, and eventually, over the course of learning, *transport* $p_g$ towards $p_d$, *i.e.*, $S_{p_g} \to S_{p_d} \Rightarrow \tilde{D}_p^*(\boldsymbol{x}) \to 0$.

## 5.1 Related Works

**GANs and Gradient Flows**: A prominent example where an RBF network has been used for the discriminator is that of Hu et al. (2020), who solve 2-D flow-field reconstruction problems. KALE Flow (Glaser et al., 2021) and MMD-Flow (Mroueh & Nguyen, 2021) also consider explicit forms of the discriminator function in terms of kernels, as opposed to training a neural network to approximate a chosen divergence between the distributions. However, unlike in PolyGANs, they leverage the closed-form function to derive the associated gradient field of the discriminator, over which a flow-based approach is employed to *transform* samples drawn from a parametric noise distribution into those following the target distribution. Similarly, in Sobolev descent (Mroueh et al., 2019; Mroueh & Rigotti, 2020), a gradient flow over the Sobolev GAN critic

is implemented. A similar approach could also be explored in this context, considering the gradient field of the Poly-WGAN discriminator, which is a promising direction for future research.

**GANs and Neural Tangent Kernels**: Along another vertical, Franceschi et al. (2022) and Zhang et al. (2022) analyze the IPM-GAN losses from the perspective of neural tangent kernels (NTKs), and show that the existing IPM-GAN losses optimize an MMD-kernel loss associated with the NTK of an infinite-width discriminator network (under suitable assumptions on the network architecture) with the kernel drawn from an associated RKHS. In contrast, we consider the functional form of the discriminator optimization, and derive a kernel-based optimum considering both the WGAN and LSGAN losses with higher-order gradient regularizers. Our formulation does not consider a distance metric, but a pseudo-norm, and the optimization schemes used provides a general approach to analyzing regularized GAN losses.

## 6 Experimental Validation on Synthetic Data

We now compare Poly-WGAN with the following baselines: WGAN-GP, WGAN-LP, WGAN-ALP, WGAN-$R_d$ and WGAN-$R_g$ variants of WGANs; and GMMN with the Gaussian (GMMN-RBFG) and the inverse multiquadric (GMMN-IMQ) kernels (cf. Section 2).

### 6.1 Two-dimensional Gaussian Learning

To serve as an illustration, consider the tasks of learning 2-D unimodal and multimodal Gaussian distributions. The data preparation and network architectures are described in Appendix D.2. For performance quantification, we use the Wasserstein-2 distance between the target and generator distributions $\left(\mathcal{W}^{2,2}(p_d, p_g)\right)$. Figure 3(a) shows $\mathcal{W}^{2,2}(p_d, p_g)$ versus the iteration count, on the 2-D Gaussian learning task. Two variants of Poly-WGAN were considered, one with $m = 1$ and the other with $m = 2$. In both cases, the convergence of Poly-WGAN is about two times faster than WGAN-$R_d$, which is the best performing baseline. Figure 3(b) shows $\mathcal{W}^{2,2}(p_d, p_g)$ as a function of iterations for GMM learning. Again, Poly-WGAN converges faster than the baselines and to a better score (lower $\mathcal{W}^{2,2}(p_d, p_g)$ value). Images comparing the performance of the GAN models are included in Appendix E.2 and ablation experiments showcasing the computational speedup in Poly-WGAN (of about two orders of magnitude) over baselines are provided in Appendix 6.3.

***Choice of the Gradient Order***: Figure 3(e) shows the Wasserstein-2 distance $\mathcal{W}^{2,2}(p_d, p_g)$ for Poly-WGAN as a function of iterations for various $m$. We observe that $m = \frac{n}{2} = 1$ is the fastest in terms of convergence speed, while penalties up to order $m = 6$ also result in favorable convergence behavior. For values of $m$ such that $2m - n \geq 10$, we encountered numerical instability issues. In view of these findings, we suggest $m \approx \lceil \frac{n}{2} \rceil$. A discussion on why this choice of $m$ is also theoretically sound, based on the *Sobolev embedding theorem*, is given in Appendix C.5. Poly-WGAN with $m = 1$ is also robust to the choice of the learning rate parameter. For instance, it converges stably even for learning rates as high as $10^{-1}$.

### 6.2 Higher-dimensional Gaussian Learning

Next, we demonstrate the success of Poly-WGANs in a high-dimensional setting. The target distribution is the Gaussian $\mathcal{N}(0.7\mathbf{1}_n, 0.2\,\mathbb{I}_n)$, where $\mathbf{1}_2$ is the 2-D vector of ones, and $\mathbb{I}_2$ is the $2 \times 2$ identity matrix. To consider both even and odd variants of the polyharmonic solution, we perform two experiments, one with $n = 16$ and the other with $n = 63$. We also analyze the effect of varying $m$ for the case when $n = 6$. The generator has the DCGAN architecture (Radford et al., 2016), while the other training parameters and architectures are identical to the 2-D learning scenario (cf. Appendix D.2). The convergence is measured in terms of the Wasserstein-2 distance, $\mathcal{W}^{2,2}(p_d, p_g)$. Figure 3(f) shows $\mathcal{W}^{2,2}$ as a function of iterations for Poly-WGAN learning for various $m$. We observe that $m = \frac{n}{2}$ performs the best, as suggested by the theoretical analysis in Section 4.3. Similar to the 2-D Gaussian case, numerical instability was encountered for $2m - n \geq 10$. The instability can be overcome to a certain extent by reducing the learning rate, but at the expense of slow training. Therefore, we consider $m = \lceil \frac{n}{2} \rceil$ as the most stable choice in the subsequent experiments. Figures 3(c) & (d) present the results for learning on 16-D and 63-D Gaussians, respectively, where Poly-WGAN with $\lceil \frac{n}{2} \rceil^{th}$-order penalty outperforms the baselines, converging by an order of magnitude faster in both cases.

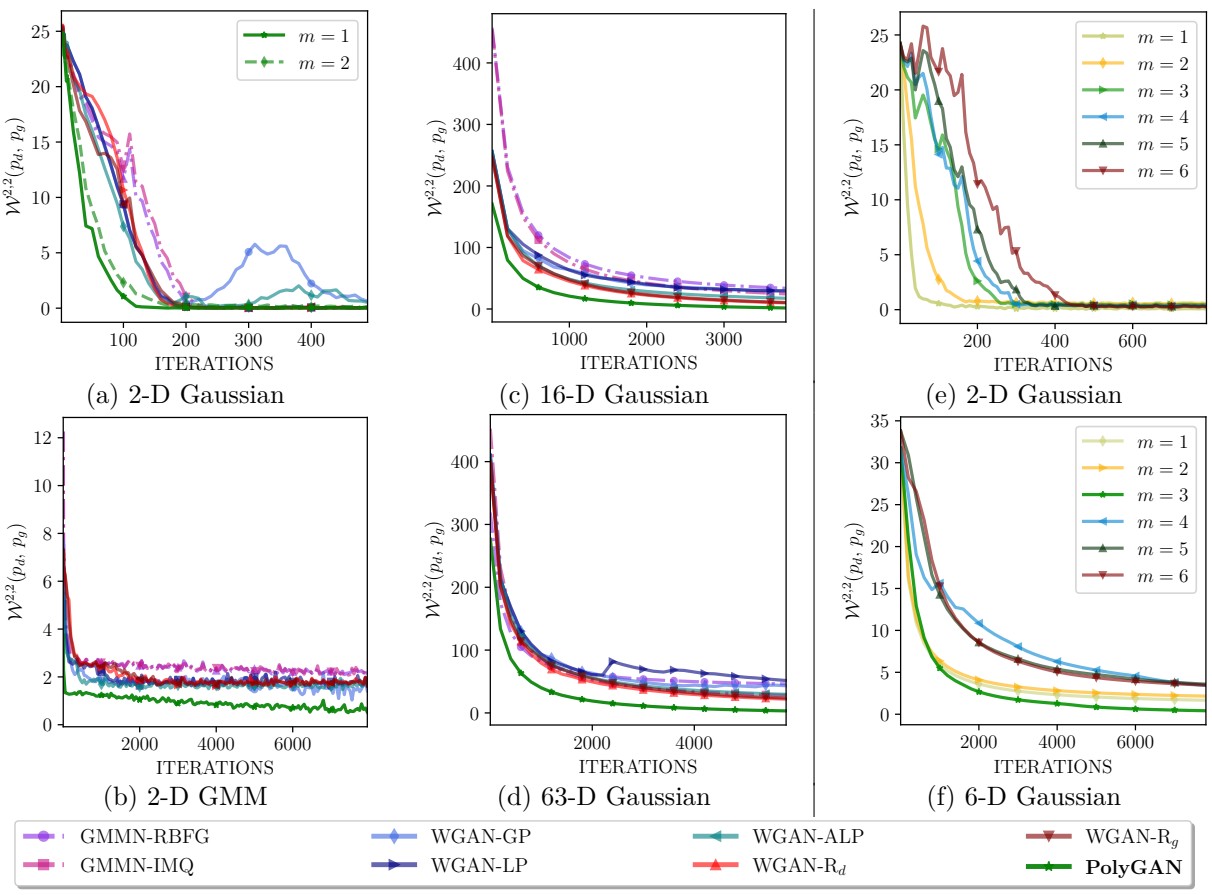

Figure 3: Training GAN variants on multivariate Gaussians. Wasserstein-2 distance between the data and generator distributions $\left(\mathcal{W}^{2,2}(p_d, p_g)\right)$ on learning (a) a 2-D Gaussian; (b) a 2-D Gaussian mixture model (GMM) ; (c) a 16-D Gaussian; (d) a 63-D Gaussian; (e) a 2-D Gaussian using Poly-WGAN for various $m$; and (f) a 6-D Gaussian using Poly-WGAN for various $m$. The legend is common to subfigures (a)-(d). Subfigure (a) further depicts two scenarios of Poly-WGAN for $m = 1$ (solid line) and $m = 2$ (dashed line). In subfigures (b)-(d) $m = \frac{n}{2}$ (solid line). $\mathcal{W}^{2,2}$ based comparison shows that Poly-WGAN outperforms the WGAN variants and the GMMN baselines in all the scenarios considered. Among the Poly-WGAN variants, the performance is the best for $m = \left\lceil \frac{n}{2} \right\rceil$, which corresponds to (e) $m = 1$ in the case of 2-D Gaussian learning; and (f) $m = 3$ for 6-D Gaussian learning.

## 6.3 Ablation Experiments

Having shown that the Poly-WGAN formulation indeed results in superior performance compared to the baselines, we now perform ablation experiments to gain a deeper understanding into the advantages of implementing the RBF-based Poly-WGAN discriminator, over the baselines.

To evaluate the computational speed-up achieved by Poly-WGAN over GANs with trainable discriminators, we perform ablation experiments comparing the convergence of Poly-WGAN and the best-case baseline WGAN-R$_d$ (cf. Section 6.1). The RBF discriminator in Poly-WGAN is compared against the WGAN-R$_d$ discriminator trained for $D_{\text{iters}} \in \{1, 2, 5, 10, 20, 100\}$ steps per generator update. We report results on the 2-D and 63-D Gaussian learning tasks. The convergence plots for $\mathcal{W}^{2,2}(p_d, p_g)$ as a function of iterations are provided in Figure 4, while the converged $\mathcal{W}^{2,2}(p_d, p_g)$ scores, and the time taken between generator updates for different choices of $D_{\text{iters}}$ (referred to as *Compute Time*) are presented in Table 1. We observe that the baseline GAN performance converges to that of Poly-WGAN, as $D_{\text{iters}}$ increases. On the 2-D learning task, the compute time in Poly-WGAN is on par with the WGAN-R$_d$ with $D_{\text{iters}} \approx 5$. However, as

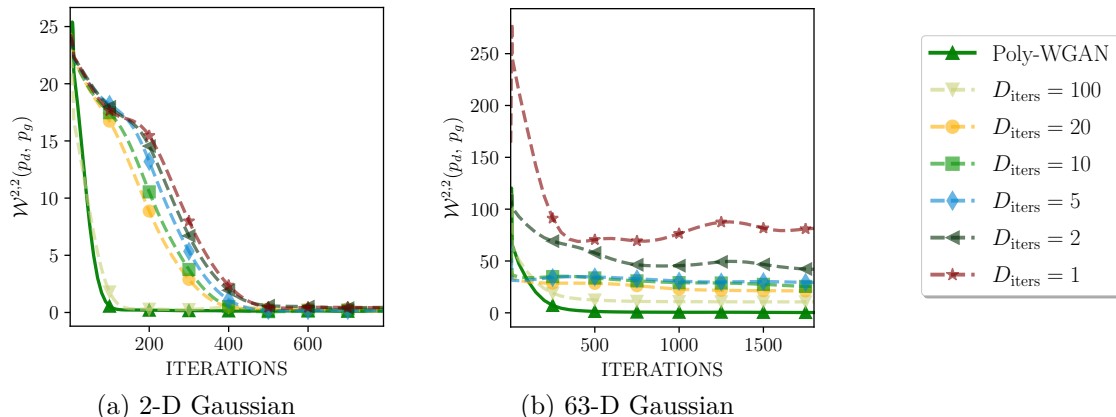

Figure 4: A comparison of the Wasserstein-2 distance between $p_d$ and $p_g$ ($\mathcal{W}^{2,2}(p_d, p_g)$) on (a) 2-D Gaussian; and (b) 63-D Gaussian learning. Poly-WGAN is compared against WGAN-$R_d$, which is the best-performing baseline (cf. Figure 3). While Poly-WGAN employs a closed-form discriminator, the discriminator in the baseline is updated for $D_{\text{iters}}$ iterations per generator update. We observe that, as the number of discriminator updates increases, the baseline performance approaches the optimal discriminator considered in Poly-WGAN.

the dimensionality of the data increases, the advantage of Poly-WGAN becomes apparent. Poly-WGAN is nearly twice as fast as WGAN-$R_d$ with $D_{\text{iters}} = 1$, while achieving $\mathcal{W}^{2,2}(p_d, p_g)$ scores that are two orders of magnitude lower than the baseline. Experimentally, Poly-WGAN is two orders of magnitude faster in training than WGAN-$R_d$ with $D_{\text{iters}} = 100$. These results clearly show that Poly-WGAN achieves superior performance over the best-case baselines, in a fraction of the training time of the generator.

To gain insights into the discriminator in PolyGAN, we consider comparisons against two baseline discriminator Scenarios – (i) A neural-network discriminator with four fully-connected layers and the *hyperbolic tangent* activation, consisting of 256, 64 and 32 and one node(s), trained using the regularized Poly-WGAN loss (cf. Equation (6)). We set K = 0 and replace the integral in the constraint with its sample estimate, akin to WGAN-$R_d$ and WGAN-$R_g$. The higher-order gradients are computed by means of nested *automatic differentiation* loops. (ii) A trainable version of the Poly-WGAN discriminator, wherein the centers and weights are initialized as in Poly-WGAN, but are subsequently updated by means of an un-regularized WGAN loss. The regularization is implicit, enforced by the choice of the activation function, which corresponds to the polyharmonic kernel of order $m$. We consider the 5-D Gaussian learning task (cf. Section 6.2).

Figure 5(a) compares the convergence of the Wasserstein-2 metric $\mathcal{W}^{2,2}(p_d, p_g)$ as a function of iterations for Poly-WGAN (solid lines), and the trainable discrimination in Scenario (i) (dashed lines), for various choices of $m$. Akin to baseline GANs, the discriminator is trained for 5 updates per generator update. In accordance with the observations in Section 6.1, we observed that $m = 3$ results in the best performance. For each $m$, the trainable baseline GAN is inferior to the corresponding Poly-WGAN. Kernel orders $m = 1, 2$ lead to a poorer performance as the kernel is singular when $m < \lceil \frac{n}{2} \rceil$. However, the trainable discriminator approach from Scenario (i) does not scale with the dimensionality of the data, as the memory requirement in computing the nested high-order gradients grows exponentially. For example, in the 5-D experiment considered above, for $m = 1$, we require $\approx 700$ MB of system memory to store the value necessary to compute the gradient penalty via back-propagation. However, for $m = 4$ we require $\approx 9$ GB of system memory! Given the choice $m = \lceil \frac{n}{2} \rceil = 3$, we also compare the effect of training the discriminator for $D_{\text{iters}}$ updaters per generator update. From Figure 5(b), we observe that the performance of the GAN with the trainable discriminator converged to the performance of Poly-GAN as $D_{\text{iters}}$ increases. To isolate the effect of the discriminator architecture on Poly-WGAN's performance, we compare Poly-WGAN with $m = 3$ against a trainable RBF discriminator as described in Scenario (ii). From Figure 5(c), we observe that, given the kernel order and the network architecture, training the RBF discriminate via stochastic-gradient updates results in significantly higher training instability. For small values of $D_{\text{iters}}$, there are large oscillations in the early stages of training, as the quality of the discriminator is sub-par. As $D_{\text{iters}}$ increases, we observed mode collapse in GANs with a trainable RBF discriminator, as the variance of the learnt gaussian converges to a small value. These

performance issues can be attributed to the learnable centers and weights, as the resulting discriminator would be a poor approximation to the ideal classifier.

These ablation experiments show that the performance of Poly-WGAN is superior to that of a GAN with a trainable RBF discriminator, which in turn is superior to the GAN with the discriminator trained on the Poly-WGAN loss. However, the scalability of Poly-WGAN to high dimensions remains a bottleneck, which we circumvent using latent-space optimization so that PolyGANs become viable on image datasets.

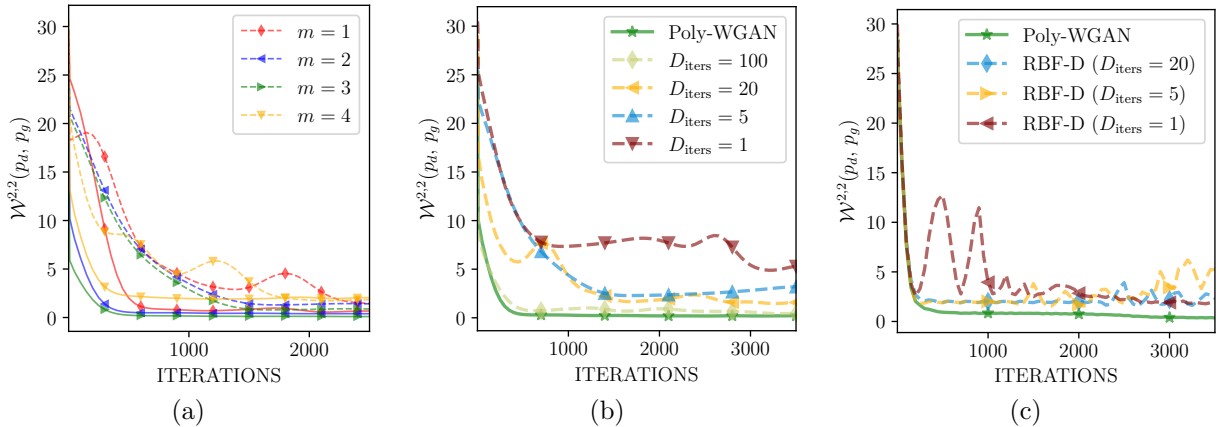

Figure 5: Comparison of Poly-WGAN against baseline GANs with a trainable discriminator on learning 5-D unimodal Gaussian data. Subfigures (a) & (b) consider a neural-network discriminator trained using the regularized Poly-WGAN loss (cf. Equation (6)), while in (c) the RBF discriminator is trained to minimize the unregularized WGAN loss. (a) For each $m$, the trainable baseline GAN is inferior to the corresponding Poly-WGAN, while Poly-WGAN with $m = 3$ results in the best performance. (b) The performance of the GAN with the trainable discriminator converges to the performance of Poly-GAN as $D_{\text{iters}}$ increases. (c) Training the RBF discriminator via stochastic-gradient updates results in significantly higher training instability compared to Poly-WGAN with the closed-form weights and centers.

Table 1: A comparison of training times for Poly-WGAN against the WGAN-$R_d$ baseline, considering various number of updates steps of the discriminator ($D_{\text{iters}}$), per generator update. The models are trained on a workstation with a single NVIDIA 3090 GPU with 24 GB of Visual RAM, and 64 GB of system RAM. Results are presented for learning 2-D and 63-D Gaussian data. The models are trained with a batch size of 500 in the case of 2-D Gaussian data, and 100 in the 63-D learning task. The table presents the (i) *Compute Time* (in seconds) per generate update; and (ii) the Wasserstein-2 distance $\mathcal{W}^{2,2}$ between the generator and data distributions of the trained model. The compute requirement in Poly-WGAN is on par with the baseline WGAN with 5 discriminator updates in low-dimensional learning tasks. As the dimensionality of the data increases, the computational load in the baselines increase drastically, with Poly-WGAN achieving superior performance with a fraction of the training time.

| WGAN flavor | $D_{\text{iters}}$ | 2-D Gaussian | | 63-D Gaussian | |
| --- | --- | --- | --- | --- | --- |
| | | Compute Time (s) ↓ | $\mathcal{W}^{2,2}(p_d, p_g)$ ↓ | Compute Time (s) ↓ | $\mathcal{W}^{2,2}(p_d, p_g)$ ↓ |
| **Poly-WGAN** | – | $0.4951 \pm 0.0034$ | **0.0107** | $\mathbf{0.0987 \pm 0.0067}$ | **0.3187** |
| WGAN-$R_d$ | 100 | $1.2451 \pm 0.0044$ | 0.0889 | $9.6541 \pm 0.0085$ | 10.5561 |
| | 20 | $0.6834 \pm 0.0055$ | 0.0743 | $1.9622 \pm 0.0071$ | 21.1581 |
| | 10 | $0.5283 \pm 0.0023$ | 0.0695 | $0.9412 \pm 0.0083$ | 25.3296 |
| | 5 | $0.4972 \pm 0.0043$ | 0.1571 | $0.5325 \pm 0.0073$ | 29.3043 |
| | 2 | $0.4856 \pm 0.0072$ | 0.3880 | $0.2358 \pm 0.0032$ | 42.0041 |
| | 1 | $\mathbf{0.4420 \pm 0.0045}$ | 0.3880 | $0.1416 \pm 0.0051$ | 68.8278 |

# 7 Experimental Validation on Standard Image Datasets

We now apply the Poly-WGAN framework to benchmark image datasets considering image- and latent-space matching. Akin to kernel based methods, Poly-WGAN is also affected by the *curse of dimensionality* (Bellman, 1957). Our aim is to develop a better understanding of the optimal discriminator in GANs and gain deeper insights, and not necessarily to outperform state-of-the-art generative techniques such as StyleGAN (Karras et al., 2021) or Diffusion models (Ho et al., 2020). Therefore, to demonstrate the feasibility of implementing the optimal discriminator, as opposed to designing networks in an uninformed way, we compare the RBF discriminator against comparable latent-space learning algorithms. There are two approaches to learning the latent-space representation in GANs – by introducing an encoder in the generator, or by introducing an encoder in the discriminator. In the image-space setting, Poly-WGAN converges faster and outperforms GMMN variants in terms of FID, although both variants generate images of poor visual quality. The experiments are presented in Appendix F.2.

**Latent-space encoders and GANs:** Encoding networks were originally introduced in a GAN setting in the context of adversarial autoencoders (AAEs) (Makhzani et al., 2015) considering the Jensen-Shannon divergence (JSD) between the target standard Gaussian $p_z \sim \mathcal{N}(\mathbf{0}, \mathbb{I})$ and the latent distribution of data $p_{d_\ell}$. A generalization incorporating Wasserstein costs was presented in the Wasserstein autoencoder (WAE) (Tolstikhin et al., 2018), where the generator also plays the role of an encoder network and the discriminator is an IPM between $p_{d_\ell}$ (the *fake* class) and $p_z$ (the *real* class). The decoder network learns a mapping from the latent space to the image space. Training the encoder-decoder pair in WAEs is a stable alternative to training GANs in the image space (Khayatkhoei et al., 2018; Pinetz et al., 2020; Feng et al., 2021). Tolstikhin et al. (2018); Patrini et al. (2018); Kolouri et al. (2019) and Gong et al. (2021) also consider kernel-based metrics to improve computational efficiency. We compare the performance of PolyGAN approach applied to WAE (PolyGAN-WAE) against the following baselines — WAE-GAN with the JSD based discriminator cost (Tolstikhin et al., 2018), the Wasserstein adversarial autoencoder with the Lipschitz penalty (WAAE-LP) (Anonymous, 2023), WAE-MMD with RBFG and IMQ kernels (Tolstikhin et al., 2018), the sliced WAE (SWAE) (Kolouri et al., 2019), the Cramér-Wold autoencoder (CWAE) (Knop et al., 2020) and WAE with a Fourier-series representation for the discriminator (WAEFR) (Anonymous, 2023). While WAE-GAN and WAAE-LP have a trainable discriminator network, the other variants use kernel metrics between $p_z$ and $p_{d_\ell}$.

An alternative to encoding in the generator space, is to learn latents representations of the real and fake images by an autoencoder discriminator architecture, as proposed in MMD-GANs by Li et al. (2017a). The need for adversarial training in the MMD-GAN discriminator results in less stable training in comparison with the MMD kernel-based methods in WAE. We therefore carry out experiments on the WAE formulation here, and provide comparisons with MMD-GAN in terms of FID and time-complexity in Appendix F.5.

## 7.1 Experimentation on Wasserstein Autoencoders

We consider four image datasets: MNIST, CIFAR-10, CelebA, and LSUN-Churches. The learning parameters are identical to those reported by Knop et al. (2020), while the network architectures are described in Appendix D.2. We consider a 16-D latent space for MNIST, 64-D for CIFAR-10, and 128-D for CelebA and LSUN-Churches. PolyGAN-WAE uses $m = \left\lceil \frac{n}{2} \right\rceil$ in all the cases. The performance metrics considered are Fréchet inception distance (FID) (Heusel et al., 2018), kernel inception distance (KID) (Bińkowski et al., 2018), image sharpness (Arjovsky et al., 2017) and reconstruction error $\langle RE \rangle$. We present FID comparisons here, while the other metrics are compared in Appendix F.2.

Figure 6 presents examples of images generated by PolyGAN-WAE when decoding samples drawn from the target latent distribution. Table 11 presents the FID of the best case converged models. We reiterate that, while these experiments are not designed to compete against state-of-the-art GANs, they compare the performance of applying a network- or kernel-based discriminator agains the RBF approach. PolyGAN-WAE outperforms the baseline WAEs in terms of FID on all datasets, with about 20% improvement on low-dimensional data such as MNIST. All the models considered failed to generate good quality images when trained on multi-class CIFAR-10, whereas SWAE failed to generate meaningful results on LSUN-Churches.

Table 2: A comparison of the WAE variants including PolyGAN-WAE in terms of Fréchet inception distance (FID). While WAE-GAN and WAAE-LP have a trainable discriminator, WAE-MMD, SWAE and CWAE use closed-form kernel functions. PolyGAN-WAE attains the optimal discriminator in closed form, while overcoming the instabilities of computing the Fourier-series expansions in WAEFR. PolyGAN-WAE achieves the best (lowest) FID compared to the baseline latent-space matching variants.

| WAE flavor | MNIST | CIFAR-10 | CelebA | LSUN-Churches |
|---|---|---|---|---|
| WAE-GAN | 21.6762 | 123.8843 | 42.9431 | 161.3421 |
| WAAE-LP | 21.2401 | 110.2232 | 43.5090 | 160.4971 |
| WAE-MMD (RBFG) | 51.2025 | 143.7128 | 56.0618 | 160.4867 |
| WAE-MMD (IMQ) | 25.9116 | 106.1817 | 43.6560 | 155.9920 |
| SWAE | 28.7962 | 107.4853 | 51.0265 | 195.6828 |
| CWAE | 25.0545 | 108.4172 | 44.8659 | 170.9388 |
| WAEFR | 21.2387 | 100.7347 | 38.3044 | 156.2485 |
| **PolyGAN-WAE** | **17.2273** | **97.3268** | **34.1568** | **139.6939** |

Figure 6: Images generated by PolyGAN-WAE upon decoding Gaussian distributed inputs.

## 8 Discussion and Conclusions

Considering the LSGAN frameworks, we showed that the GAN discriminator effectively functions as a high-dimensional interpolator involving the polyharmonic kernel. While Poly-LSGAN precisely matches the interpolation schemes, limitations in computing the inverse of very large matrices, and singularity issues potentially caused due to the manifold structure of images, made the approach impractical. We then extended the formulation to the WGAN-IPM, and showed that the interpolating nature of the optimal discriminator continues to hold. Poly-WGAN lies at the intersection between IPM based GANs and RKHS based MMD kernel losses, where the loss constrains the discriminator to come from the semi-normed *Beppo-Levi* space. Through a variational optimization, we showed that the optimal discriminator in both variants is the solution to an iterated Laplacian PDE, involving the polyharmonic RBF. We explored implementations of the *one-shot* optimal RBF discriminator and demonstrated speed up in GAN convergence, compared to both gradient-penalty based GANs and kernel based GMMNs in terms of standard convergence metrics such as $\mathcal{W}^{2,2}$. However, scaling the RBF discriminator to classify images in high-dimensional settings becomes impractical as the number of centers required in $\mathbb{R}^n$ grows as $n$ (Tavkhelidze, 2007). We therefore restricted the discriminator to work on the latent-space distribution of the data, learnt by a Wasserstein autoencoder, which reduces the dimensionality from $\mathcal{O}\left(10^6\right)$ to $\mathcal{O}\left(10^2\right)$ (cf. Section 7). While this PolyGAN-WAE framework does not outperform top-end high-resolution GAN architectures with massive compute requirements, such as StyleGAN (Karras et al., 2019; 2020; 2021) or vector quantized GANs (VQGAN) (Esser et al., 2021; Yu et al., 2022) in terms of the image quality or FID, it does outperform alternative WAE frameworks that deploy Cramér-Wold or sliced Wasserstein metric based losses. A key takeaway is that the proposed approach results in superior FID and convergence performance with a closed-form optimal discriminator as opposed to trainable discriminators or MMD based losses.

Developing improved algorithms to compute the optimal closed-form discriminator in high-dimensional spaces is a promising direction of research. Alternatives to the finite-sample RBF estimate, such as efficient mesh-free sampling strategies (Iske, 2004), or numerical PDE solvers (Ho et al., 2020; Song et al., 2021) could also be employed. The *curse of dimensionality* encountered in scaling PolyGAN variants to high-dimensional data could be circumvented by employing separable kernels for interpolation (Debarre et al., 2019). Higher-order gradient regularizers could also be incorporated into other popular GAN frameworks.

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

# Appendix

## Table of Contents

## Appendix

The Supporting Documents comprise the appendices, anonymized versions of Anonymous (2023) and Anonymous (2022) for the reviewers' reference, and the source code for PolyGANs. Anonymous (2023) has been accepted for publication, conditioned on minor revisions, at *Journal of Machine Learning Research* Anonymous (2022) has been accepted for publication at the non-archival venue *INTERPOLATE: First workshop on Interpolation Regularizers and Beyond*, NeurIPS 2022 To comply with the *double-blind* review policy, we provide an anonymized version of the manuscript as part of the *Supplementary Material.* The appendices contain the proofs of the theorems stated in the *Main Manuscript* and results of additional experimentation on synthetic Gaussians and image datasets.

## A  Mathematical Preliminaries

We recall results from the calculus of variations, which play an important role in the optimization of the new GAN flavors introduced in this paper.

Consider a vector $\boldsymbol{x} = [x_1, x_2, \ldots, x_n]^{\mathrm{T}} \in \mathbb{R}^n$ and a function $f : \mathbb{R}^n \to \mathbb{R}$. The notation $\nabla^m f(\boldsymbol{x})$ denotes the vector of $m^{th}$-order partial derivatives of $f$ with respect to the entries of $\boldsymbol{x}$. $\nabla^0$ is the identity operator. The elements of $\nabla^m f$ are represented using the multi-index $\boldsymbol{\alpha} = [\alpha_1, \alpha_2, \ldots, \alpha_n]^{\mathrm{T}}$, as:

$$\partial^{\boldsymbol{\alpha}} f = \frac{\partial^{|\boldsymbol{\alpha}|}}{\partial x_1^{\alpha_1} \partial x_2^{\alpha_2} \ldots \partial x_n^{\alpha_n}} f, \qquad \text{where} \qquad \boldsymbol{\alpha} \in \mathbb{Z}_*^n, \quad |\boldsymbol{\alpha}| = \sum_{i=1}^n \alpha_i,$$

where in turn $\mathbb{Z}_*^n$ is the set of $n$-dimensional vectors with non-negative integer entries. For example, with $n = 4, m = 3$, the index $\boldsymbol{\alpha} = [2, 0, 0, 1]^{\mathrm{T}}$ yields the element $\frac{\partial^3}{\partial x_1^2 \partial x_4} f(\boldsymbol{x})$. The square of the $L_2$-norm of $\nabla^m f$ is given by a multidimensional sum:

$$\|\nabla^m f(\boldsymbol{x})\|_2^2 = \sum_{\boldsymbol{\alpha}: \, |\boldsymbol{\alpha}|=m} \left( \frac{m!}{\boldsymbol{\alpha}!} \right) \left( \partial^{\boldsymbol{\alpha}} f(\boldsymbol{x}) \right)^2, \tag{13}$$

where $\boldsymbol{\alpha}! = \alpha_1! \alpha_2! \ldots \alpha_n!$. The iterated Laplacian, also known as the polyharmonic operator, is defined as:

$$\Delta^m f(\boldsymbol{x}) = \Delta(\Delta^{m-1} f(\boldsymbol{x})), \qquad \text{where} \qquad \Delta f(\boldsymbol{x}) = \nabla \cdot \nabla f(\boldsymbol{x}) = \sum_{i=1}^n \frac{\partial^2}{\partial x_i^2} f(\boldsymbol{x})$$

is the Laplacian operator acting on $f(\boldsymbol{x})$. Applying the multi-index notation yields the standard form of the polyharmonic operator:

$$\Delta^m f(\boldsymbol{x}) = \sum_{\boldsymbol{\alpha}: \, |\boldsymbol{\alpha}|=m} \left( \frac{m!}{\boldsymbol{\alpha}!} \right) \partial^{\boldsymbol{\alpha}} \left( \partial^{\boldsymbol{\alpha}} f(\boldsymbol{x}) \right).$$

**Calculus of Variations**: Consider an integral cost $\mathcal{L}$ with the integrand $\mathcal{F}$ dependent on $f$ and all its partial derivatives up to and including order $\ell$, given by

$$\mathcal{L}\left(f(\boldsymbol{x}), \partial^{\boldsymbol{\alpha}} f; |\boldsymbol{\alpha}| \le \ell\right) = \int_{\mathcal{X}} \mathcal{F}\left(f(\boldsymbol{x}), \partial^{\boldsymbol{\alpha}} f; |\boldsymbol{\alpha}| \le \ell\right) \mathrm{d}\boldsymbol{x},$$

defined on a suitable domain $\mathcal{X}$ over which $f$ and its partial derivatives up to and including order $\ell$ are continuously differentiable.

The optimizer $f^*$ must satisfy the Euler-Lagrange condition:

$$\frac{\partial \mathcal{F}}{\partial f} + \sum_{j=1}^{\ell} \left( (-1)^j \sum_{\boldsymbol{\alpha}: \, |\boldsymbol{\alpha}|=j} \partial^{\boldsymbol{\alpha}} \left( \frac{\partial \mathcal{F}}{\partial(\partial^{\boldsymbol{\alpha}} f)} \right) \right) \Bigg|_{f=f^*} = 0. \tag{14}$$

# B The Optimal Poly-LSGAN Discriminator

The proof of Theorem 3.1 follows from the results in mesh-free interpolation literature (Aronszajn et al., 1983; Iske, 2004; Fasshauer, 2007) that deal with the generic polyharmonic spline interpolation problem. For completeness, we provide the proof here. While the assumption may appear strong, we show that this is implicitly satisfied by the optimal solution. Recall the discriminator optimization problem given in Eq. (3)

$$\arg\min_{D} \left\{ \sum_{\substack{i=1 \\ (\boldsymbol{c}_i, y_i) \sim \mathcal{D}}}^{N} (D(\boldsymbol{c}_i) - y_i)^2 + \lambda_d \int_{\mathcal{X}} \|\nabla^m D(\boldsymbol{x})\|_2^2 \, \mathrm{d}\boldsymbol{x} \right\}. \tag{15}$$

To compute the functional optimum in the Calculus of Variations setting, the above cost must be cast into an integral form. Using the Dirac delta function, we have:

$$\sum_{\substack{i=1 \\ (\boldsymbol{c}_i, y_i) \sim \mathcal{D}}}^{N} (D(\boldsymbol{c}_i) - y_i)^2 = \int_{\mathcal{X}} \sum_{\substack{i=1 \\ (\boldsymbol{c}_i, y_i) \sim \mathcal{D}}}^{N} (D(\boldsymbol{x}) - y_i)^2 \, \delta(\boldsymbol{x} - \boldsymbol{c}_i) \, \mathrm{d}\boldsymbol{x}.$$

Then, Equation 15 can be rewritten as an integral-cost minimization:

$$\arg\min_{D} \left\{ \int_{\mathcal{X}} \underbrace{\sum_{\substack{i=1 \\ (\boldsymbol{c}_i, y_i) \sim \mathcal{D}}}^{N} (D(\boldsymbol{x}) - y_i)^2 \, \delta(\boldsymbol{x} - \boldsymbol{c}_i) + \lambda_d \|\nabla^m D(\boldsymbol{x})\|_2^2}_{\mathcal{F}(D, \partial^{\boldsymbol{\alpha}} D; |\boldsymbol{\alpha}|=m)} \, \mathrm{d}\boldsymbol{x} \right\}.$$

Computing the derivatives of the integrand $\mathcal{F}$ with respect to $D$ and $\partial^{\alpha} D$ yields

$$\frac{\partial \mathcal{F}}{\partial D} = 2 \sum_{\substack{i=1 \\ (\boldsymbol{c}_i, y_i) \sim \mathcal{D}}}^{N} (D(\boldsymbol{x}) - y_i) \, \delta(\boldsymbol{x} - \boldsymbol{c}_i), \qquad \text{and} \qquad \sum_{\boldsymbol{\alpha}:|\boldsymbol{\alpha}|=m} \partial^{\boldsymbol{\alpha}} \left( \frac{\partial \mathcal{F}}{\partial(\partial^{\boldsymbol{\alpha}} D)} \right) = 2\lambda_d \Delta^m D(\boldsymbol{x}).$$

Substituting the above into the Euler-Lagrange equation (Eq. 14) gives us the partial differential equation that the optimal discriminator $D^*(\boldsymbol{x})$ must satisfy:

$$\left( \sum_{i=1}^{N} (D(\boldsymbol{x}) - y_i)\delta(\boldsymbol{x} - \boldsymbol{c}_i) \right) + (-1)^m \lambda_d \Delta^m D(\boldsymbol{x}) \, \Bigg|_{D=D^*(\boldsymbol{x})} = 0.$$

While the above condition is applicable for a strong solution, a weak solution to $D(\boldsymbol{x})$ satisfies:

$$\int_{\mathcal{X}} \left( \left( \sum_{i=1}^{N} (D(\boldsymbol{x}) - y_i)\delta(\boldsymbol{x} - \boldsymbol{c}_i) \right) + (-1)^m \lambda_d \Delta^m D(\boldsymbol{x}) \right) \eta(\boldsymbol{x}) \, \mathrm{d}\boldsymbol{x} \, \Bigg|_{D=D^*(\boldsymbol{x})} = 0, \tag{16}$$

where $\eta(\boldsymbol{x})$ is any test function drawn from the family of compactly-supported infinitely-differentiable functions. Aronszajn et al. (1983), an authoritative resource on polyharmonic functions, has shown that, functions of the form

$$f(\boldsymbol{x}) = \sum_{\substack{i=1 \\ (\boldsymbol{c}_i, y_i) \sim \mathcal{D}}}^{N} w_i \varphi_k (\|\boldsymbol{x} - \boldsymbol{c}_i\|) + P(\boldsymbol{x}; \boldsymbol{v}), \text{ where } \varphi_k(\|\boldsymbol{x}\|) = \begin{cases} \|\boldsymbol{x}\|^k & \text{for } k = 1, 3, \cdots \\ \|\boldsymbol{x}\|^k \ln(\|\boldsymbol{x}\|) & \text{for } k = 2, 4, \cdots, \end{cases} \tag{17}$$

satisfy the polyharmonic PDE:

$$\Delta^m f(\boldsymbol{x}) = \sum_{i=1}^{N} C_k w_i \delta(\boldsymbol{x} - \boldsymbol{c}_i),$$

where $P(\boldsymbol{x}; \boldsymbol{v}) \in \mathcal{P}_{m-1}^n$ is the $(m-1)^{th}$ order polynomial parametrized by the coefficients $\boldsymbol{v} \in \mathbb{R}^L$, where:

$$L = \sum_{\ell=0}^{m-1} \binom{n+\ell-1}{\ell} = \binom{n+m-1}{m-1}$$

For example, with $m = 2$, we have $\mathcal{P}(\boldsymbol{x}; \boldsymbol{v}) = \langle \boldsymbol{v}_1, \boldsymbol{x} \rangle + v_0;\ \boldsymbol{v} = \begin{bmatrix} v_0 \\ \boldsymbol{v}_1 \end{bmatrix} \in \mathbb{R}^{n+1}$. Substituting the above back into Equation (16), we get

$$\int_{\mathcal{X}} \left( \sum_{i=1}^N \left( (D(\boldsymbol{x}) - y_i) + (-1)^m \lambda_d C_k w_i \right) \delta(\boldsymbol{x} - \boldsymbol{c}_i) \right) \eta(\boldsymbol{x})\ \mathrm{d}\boldsymbol{x} \ \Bigg|_{D=D^*(\boldsymbol{x})} = 0$$

$$\Rightarrow \sum_{i=1}^N \left( (D(\boldsymbol{c}_i) - y_i) + (-1)^m \lambda_d C_k w_i \right) \eta(\boldsymbol{c}_i) \ \Bigg|_{D=D^*(\boldsymbol{x})} = 0$$

Since the above condition must hold for all possible test functions $\eta$, we have:

$$D^*(\boldsymbol{c}_i) - y_i + (-1)^m \lambda_d \mathrm{C}_k w_i = 0 \qquad \forall\ i = 1, 2, \cdots N,$$

where $D^*$ is given by Equation (17). Substituting for $D^*$ and stacking for all $i$ gives the following condition that the weights and polynomial coefficients satisfy:

$$\left( \mathbf{A} + (-1)^m \lambda_d \mathrm{C}_k \mathbf{I} \right) \boldsymbol{w} + \mathbf{B}\boldsymbol{v} = \boldsymbol{y}, \tag{18}$$

$$\text{where} \quad [\mathbf{A}]_{i,j} = \psi_k(\|\boldsymbol{c}_i - \boldsymbol{c}_j\|); \quad \boldsymbol{w} = [w_1, w_2, \ldots, w_N]^{\mathrm{T}}, \quad \boldsymbol{y} = [y_1, y_2, \cdots, y_N]^{\mathrm{T}},$$

$$\mathbf{B} = \begin{bmatrix} 1 & 1 & \cdots & 1 \\ \boldsymbol{c}_1 & \boldsymbol{c}_2 & \cdots & \boldsymbol{c}_N \\ \vdots & \vdots & \ddots & \vdots \\ \boldsymbol{c}_1^{m-1} & \boldsymbol{c}_2^{m-1} & \cdots & \boldsymbol{c}_N^{m-1} \end{bmatrix}^{\mathrm{T}}, \text{ and } \boldsymbol{v} = [v_0, v_1, v_2, \ldots, v_L]^{\mathrm{T}}.$$

The matrix $\mathbf{B}$ corresponds to a Vandermonde matrix when $n = 1$. The above system of equations has a unique solution when the kernel matrix $\mathbf{A}$ is invertible and $\mathbf{B}$ is full column-rank. Matrix $\mathbf{A}$ is invertible if the set of real/fake centers are unique, and the kernel order $2m - n$ is positive. On the other hand, matrix $\mathbf{B}$ is full rank if the set of centers $\{\boldsymbol{c}_i\}$ are linearly independent, and more specifically, do not lie on any subspace of $\mathbb{R}^n$ (Iske, 2004). The above system of linear equations only provides us with the conditions on the weights and coefficients that the discriminator radial basis function expansion satisfies. In order to derive the optimal discriminator, the one that minimizes the discriminator loss, we substitute the RBF form of the discriminator into Equation (15) and solve for the weights and polynomial coefficients.

To derive the second condition present in Equation (5), we first consider deriving the higher-order gradient penalty in terms of the optimal RBF discriminator $D^*$. Consider the inner-product space associated with the higher-order gradient (the Beppo-Levi space $\mathrm{BL}^{m,2}$), given by (Aronszajn et al., 1983):

$$\langle f, g \rangle \triangleq \int_{\mathbb{R}^n} (\nabla^m f) \cdot (\nabla^m g)\ \mathrm{d}\boldsymbol{x} = (-1)^m \int_{\mathbb{R}^n} f \cdot (\Delta^m g)\ \mathrm{d}\boldsymbol{x}$$

where the second inequality is via integration by parts. For any function $D^*$ of the form given in Equation (17):

$$\langle D^*, D^* \rangle = (-1)^m \int_{\mathbb{R}^n} D^*(\boldsymbol{x}) \left( \sum_{i=1}^N C_k w_i \delta(\boldsymbol{x} - \boldsymbol{c}_i) \right)\ \mathrm{d}\boldsymbol{x} = (-1)^m C_k \sum_{i=1}^N w_i D^*(\boldsymbol{c}_i).$$

Substituting for $D^*$ from Equation (17) gives:

$$\int_{\mathbb{R}^n} \|\nabla^m D^*\|_2^2\ \mathrm{d}\boldsymbol{x} = \langle D^*, D^* \rangle = (-1)^m C_k \sum_{i=1}^N \left( w_i \left( \sum_{\substack{j=1 \\ (\boldsymbol{c}_j, y_j) \sim \mathcal{D}}}^N w_j \psi_k \left( \|\boldsymbol{c}_j - \boldsymbol{c}_i\| \right) + [\mathbf{B}\boldsymbol{v}]_i \right) \right)$$

$$= (-1)^m C_k \boldsymbol{w}^{\mathrm{T}} \mathbf{A} \boldsymbol{w}, \tag{19}$$

where the second equality holds as a result of Equation 21. Substituting in $D^*$ and Equation 19 into the optimization problem in Equation 15 yields:

$$
\arg\min_{D} \left\{ \sum_{\substack{i=1 \\ (\boldsymbol{c}_i, y_i) \sim \mathcal{D}}}^{N} (D(\boldsymbol{c}_i) - y_i)^2 + \lambda_d \int_{\mathcal{X}} \|\nabla^m D(\boldsymbol{x})\|_2^2 \, \mathrm{d}\boldsymbol{x} \right\}
$$
$$
= \arg\min_{\boldsymbol{w}, \boldsymbol{v}} \left\{ \underbrace{\|\mathbf{A}\boldsymbol{w} + \mathbf{B}\boldsymbol{v} - \boldsymbol{y}\|_2^2 + \lambda_d C_k \boldsymbol{w}^{\mathrm{T}} \mathbf{A}\boldsymbol{w}}_{\mathrm{F}(\boldsymbol{w}, \boldsymbol{v})} \right\}.
\tag{20}
$$

Minimizing the cost function in Equation (20) with respect to $\boldsymbol{w}$ and $\boldsymbol{v}$ yields:

$$
\frac{\partial \mathrm{F}}{\partial \boldsymbol{w}} = 2\mathbf{A}^{\mathrm{T}} \left( \mathbf{A}\boldsymbol{w} + \mathbf{B}\boldsymbol{v} - \boldsymbol{y} + 2\lambda_d C_k \boldsymbol{w} \right) = 0, \quad \text{and} \quad \frac{\partial \mathrm{F}}{\partial \boldsymbol{v}} = 2\mathbf{B}^{\mathrm{T}} \left( \mathbf{A}\boldsymbol{w} + \mathbf{B}\boldsymbol{v} - \boldsymbol{y} \right) = 0
$$
$$
\Rightarrow \mathbf{B}^{\mathrm{T}} \boldsymbol{w} = \mathbf{0},
\tag{21}
$$

which gives us the second necessary condition that the optimal weights and polynomial coefficients must satisfy. Equation (21) ensure that the solution obtained is such that the sum of the unbounded polyharmonic kernels vanishes as $\boldsymbol{x}$ tends to infinity. Essentially, in regions close to the centers $\boldsymbol{c}_i$, there is a large contribution in $D^*(\boldsymbol{x})$ from the kernel function, and when far away from the centers, the polynomial has a large contribution in $D^*(\boldsymbol{x})$. This ensures that the the discriminator obtained by solving the system of equations does not grow to infinity. This completes the proof of Theorem 3.1.

## C   Optimality of Poly-WGAN

In this appendix, we present the proofs of theorems associated with the optimality of Poly-WGAN, and derive bounds for the optimal Lagrange multiplier of the regularized Poly-WGAN cost.

### C.1   Constants in the Fundamental Solution

Theorem 4.1 contains a constant $\varrho$ that is a function of $m$ and $n$. The exact expression for the constant is provided here. Consider the fundamental solution $\Delta^m \varrho \psi_{2m-n}(\boldsymbol{x}) = \delta(\boldsymbol{x})$, where the polyharmonic radial basis function is given by

$$
\psi_{2m-n}(\boldsymbol{x}) = \begin{cases} \|\boldsymbol{x}\|^{2m-n}, & \text{if } 2m - n < 0 \text{ or } n \text{ is odd}, \\ \|\boldsymbol{x}\|^{2m-n} \ln(\|\boldsymbol{x}\|), & \text{if } 2m - n \geq 0 \text{ and } n \text{ is even}. \end{cases}
$$

The value of $\varrho$ is given by (Aronszajn et al., 1983):

$$
\varrho = \begin{cases} \dfrac{2^{2-2m}}{(m-1)!} \dfrac{\Gamma(2-\tau)}{\Gamma(m+1-\tau)}, & \text{for } m = 1, 2, 3, \ldots, \text{ and } n \text{ is odd}, \\[2.5ex] (-1)^{(m-1)} \dfrac{2^{2-2m}}{(m-1)!} \dfrac{(\tau - m - 1)!}{(\tau - 1)!}, & \text{for } m = 1, 2, 3, \ldots, (\tau - 2) \text{ and } n \geq 4 \text{ is even}, \\[2.5ex] (-1) \dfrac{2^{2-2m}}{(m-1)!(\tau-2)!(m-\tau)!}, & \text{for } m = (\tau - 1), \tau, \ldots \text{ and } n \geq 4 \text{ is even}, \\[2.5ex] \left( \dfrac{2^{1-m}}{(m-1)!} \right)^2, & \text{for } m = 1, 2, 3, \ldots, \text{ and } n = 2, \end{cases}
$$

where $\tau = \dfrac{n}{2}$, and $\Gamma(z)$ is the Gamma function given in terms of the factorial expression as $\Gamma(z) = (z-1)!$ for integer $z$, and by the improper integral $\Gamma(z) = \displaystyle\int_0^{\infty} x^{z-1} e^{-x} \, \mathrm{d}x$, $\mathrm{Re}(z) > 0$, for $z \in \mathbb{C}$. As shown in the subsequent sections, the exact value of $\varrho$ turns out to be inconsequential for the optimal discriminator $D^*(\boldsymbol{x})$ as its effect gets nullified by the optimal Lagrange parameter $\lambda_d^*$.

## C.2 Optimal Lagrange Multiplier

The optimal Lagrange multiplier $\lambda_d^*$ can be computed by enforcing the gradient constraint $\Omega_D$ on the optimal discriminator:

$$\Omega_D : \quad \int_{\mathcal{X}} \|\nabla^m D^*(\boldsymbol{x})\|_2^2 \, \mathrm{d}\boldsymbol{x} = \mathrm{K}|\mathcal{X}|, \tag{22}$$

where $|\mathcal{X}|$ denotes the volume of the domain $\mathcal{X}$, and $D^*(\boldsymbol{x}) = D_p^*(\boldsymbol{x}) + P(\boldsymbol{x})$, where in turn $P(\boldsymbol{x}) \in \mathcal{P}_{m-1}^n(\boldsymbol{x})$ is an $(m-1)$-degree polynomial, and $D_p^*(\boldsymbol{x})$ is the particular solution. We have $\partial^{\boldsymbol{\alpha}} P(\boldsymbol{x}) = 0$, $\forall \, \boldsymbol{\alpha}$, such that $|\boldsymbol{\alpha}| = m$. Hence, we only consider $D_p^*(\boldsymbol{x})$ in the subsequent analysis. Without loss of generality, assume that $n$ is odd. The analysis is similar for the case when $n$ is even.

First, consider the radially symmetric function $p(\boldsymbol{x}) = Q_0 \|\boldsymbol{x}\|^k$, where $|Q_0| \leq 1$ is a zeroth-degree polynomial (a constant), whose magnitude is bounded by 1. For multi-index $\boldsymbol{\alpha}$, we have

$$\partial^{\boldsymbol{\alpha}} p(\boldsymbol{x}) = \partial^{\boldsymbol{\alpha}} Q_0 \|\boldsymbol{x}\|^k = Q_{|\boldsymbol{\alpha}|}(k) \|\boldsymbol{x}\|^{k-2|\boldsymbol{\alpha}|},$$

where $Q_{|\boldsymbol{\alpha}|}(k)$ is a $|\boldsymbol{\alpha}|$-degree polynomial in $k$ consisting of at most $n^{|\boldsymbol{\alpha}|}$ terms, with the coefficient of each term bounded by $(|k|+1)(|k|+3)\ldots(|k|+2|\boldsymbol{\alpha}|-1)$. Aronszajn et al. (1983) showed that, when $k = 2m - n$ and $|\boldsymbol{\alpha}| = m$, the following simplified bound holds:

$$\partial^{\boldsymbol{\alpha}} \|\boldsymbol{x}\|^{2m-n} \leq (2n)^m \frac{\Gamma\left(2m + \frac{n+1}{2}\right)}{\Gamma\left(m + \frac{n+1}{2}\right)} \|\boldsymbol{x}\|^{-n}. \tag{23}$$

Consider the integral form of the particular solution:

$$D_p^*(\boldsymbol{x}) = \frac{(-1)^{m+1}}{2\lambda_d^*} \int_{\boldsymbol{y} \in \mathcal{X}} (p_g(\boldsymbol{y}) - p_d(\boldsymbol{y})) \, r(\boldsymbol{x} - \boldsymbol{y}) \, \mathrm{d}\boldsymbol{y}.$$

Computing the $\boldsymbol{\alpha}^{th}$ partial derivative with respect to $\boldsymbol{x}$ gives

$$\partial_{\boldsymbol{x}}^{\boldsymbol{\alpha}} D_p^*(\boldsymbol{x}) = \frac{(-1)^{m+1}\varrho}{2\lambda_d^*} \left( \int_{\boldsymbol{y} \in \mathcal{X}} p_g(\boldsymbol{y}) \partial_{\boldsymbol{x}}^{\boldsymbol{\alpha}} \|\boldsymbol{x} - \boldsymbol{y}\|^{2m-n} \, \mathrm{d}\boldsymbol{y} - \int_{\boldsymbol{y} \in \mathcal{X}} p_d(\boldsymbol{y}) \partial_{\boldsymbol{x}}^{\boldsymbol{\alpha}} \|\boldsymbol{x} - \boldsymbol{y}\|^{2m-n} \, \mathrm{d}\boldsymbol{y} \right).$$

Squaring on both sides yields:

$$\left(\partial_{\boldsymbol{x}}^{\boldsymbol{\alpha}} D_p^*(\boldsymbol{x})\right)^2 = \left(\frac{\xi}{\lambda_d^*}\right)^2 \left( \int_{\boldsymbol{y} \in \mathcal{X}} (p_g(\boldsymbol{y}) - p_d(\boldsymbol{y})) \, \partial_{\boldsymbol{x}}^{\boldsymbol{\alpha}} \|\boldsymbol{x} - \boldsymbol{y}\|^{2m-n} \, \mathrm{d}\boldsymbol{y} \right)^2$$

$$\leq \frac{\xi^2 \varepsilon^2}{\lambda_d^{*2}} \left( \int_{\boldsymbol{y} \in \mathcal{X}} (p_g(\boldsymbol{y}) - p_d(\boldsymbol{y})) \, \|\boldsymbol{x} - \boldsymbol{y}\|^{-n} \, \mathrm{d}\boldsymbol{y} \right)^2,$$

where $\varepsilon = \dfrac{(2n)^m \Gamma\left(2m + \frac{n+1}{2}\right)}{\Gamma\left(m + \frac{n+1}{2}\right)}$ and $\xi = \dfrac{(-1)^{m+1}\varrho}{2}$ and the inequality is a consequence of Eq. (23). A similar analysis can be carried out for the case when $n$ is even. In general, we have

$$\left(\partial_{\boldsymbol{x}}^{\boldsymbol{\alpha}} D_p^*(\boldsymbol{x})\right)^2 \leq \frac{\varepsilon^2 \xi^2}{\lambda_d^{*2}} \left( ((p_g - p_d) * \psi_{-n}) (\boldsymbol{x}) \right)^2,$$

where $\psi$ is as defined in Section 4.2 of the *Main Manuscript*. The square of the L$_2$-norm of $\nabla^m D_p^*(\boldsymbol{x})$ can be bounded as follows:

$$\|\nabla^m D_p^*(\boldsymbol{x})\|_2^2 = \sum_{|\boldsymbol{\alpha}|=m} \frac{m!}{\boldsymbol{\alpha}!} \left(\partial_{\boldsymbol{x}}^{\boldsymbol{\alpha}} D_p^*(\boldsymbol{x})\right)^2 \leq \frac{m! \, \varepsilon^2 \xi^2}{\lambda_d^{*2}} \left( ((p_g - p_d) * \psi_{-n}) (\boldsymbol{x}) \right)^2 \sum_{|\boldsymbol{\alpha}|=m} \frac{1}{\boldsymbol{\alpha}!}.$$

Substituting the above into Eq. (22), we obtain:

$$|\mathcal{X}| = \int_{\mathcal{X}} \|\nabla^m D^*(\boldsymbol{x})\|_2^2 \, \mathrm{d}\boldsymbol{x} \leq \frac{m! \, \varepsilon^2 \xi^2}{\lambda_d^{*2}} \underbrace{\sum_{\substack{\boldsymbol{\alpha} \\ |\boldsymbol{\alpha}|=m}} \left( \frac{1}{\boldsymbol{\alpha}!} \right)}_{S_{\boldsymbol{\alpha}}} \int_{\mathcal{X}} \left( ((p_g - p_d) * \psi_{-n})(\boldsymbol{x}) \right)^2 \, \mathrm{d}\boldsymbol{x}.$$

Rearranging the terms and simplifying gives us an upper bound on the square of $\lambda_d^*$:

$$\lambda_d^{*2} \leq \frac{m! \, \varepsilon^2 \xi^2 S_{\boldsymbol{\alpha}}}{|\mathcal{X}|} \int_{\mathcal{X}} \left( ((p_g - p_d) * \psi_{-n})(\boldsymbol{x}) \right)^2 \, \mathrm{d}\boldsymbol{x}. \tag{24}$$

**Practical Implementation**: While Eq. (24) gives a theoretical bound on $\lambda_d^*$, the integral, which in turn involves a convolution integral, cannot be computed practically. We therefore replace it with a feasible alternative, $\tilde{\lambda}_d^*$ based on sample approximations. Replacing the integral over $\boldsymbol{x}$ with a sample estimate yields:

$$\tilde{\lambda}_d^{*2} \leq \frac{m! \, \varepsilon^2 \xi^2 S_{\boldsymbol{\alpha}}}{\mathrm{K}|\mathcal{X}|} \frac{1}{M} \sum_{\ell=1}^{M} \left( ((p_g - p_d) * \psi_{-n})(\boldsymbol{x}_\ell) \right)^2,$$

where $|\mathcal{X}| = M$. Simplifying the convolutions similar to the approach used in Section 4.3 of the *Main Manuscript*, we obtain:

$$\tilde{\lambda}_d^{*2} \leq \frac{m! \, \varepsilon^2 \xi^2 S_{\boldsymbol{\alpha}}}{\mathrm{K}M^2} \sum_{\ell=1}^{M} \left( \mathbb{E}_{\boldsymbol{y} \sim p_g}[\psi_{-n}(\boldsymbol{x}_\ell - \boldsymbol{y})] - \mathbb{E}_{\boldsymbol{y} \sim p_d}[\psi_{-n}(\boldsymbol{x}_\ell - \boldsymbol{y})] \right)^2.$$

Replacing the expectations with their $N$-sample estimates could be used as an estimate of the upper bound:

$$\tilde{\lambda}_d^{*2} \leq \frac{m! \, \varepsilon^2 \xi^2 S_{\boldsymbol{\alpha}}}{\mathrm{K}N^2 M^2} \sum_{\ell=1}^{M} \left( \sum_{\boldsymbol{c}_i \sim p_g; \, i=1}^{N} \psi_{-n}(\boldsymbol{x}_\ell - \boldsymbol{c}_i) - \sum_{\boldsymbol{c}_j \sim p_d; \, j=1}^{N} \psi_{-n}(\boldsymbol{x}_\ell - \boldsymbol{c}_j) \right)^2, \tag{25}$$

where $\boldsymbol{c}_i$ and $\boldsymbol{c}_j$ are drawn from $p_g$ and $p_d$, respectively.

**The sign of $\lambda_d^*$:** The choice of the sign on $\tilde{\lambda}_d^*$ is determined by the optimization problem. While the solution to the Euler-Lagrange equation gives an extremum, whether it is a maximizer or a minimizer must be ascertained based on the second variation of the cost, which is derived below.

Before proceeding further, we recapitulate the second variation of an integral cost. Consider the cost:

$$\mathcal{L}\left(f, \partial^{\boldsymbol{\alpha}} f; |\boldsymbol{\alpha}| \leq k\right) = \int_{\mathcal{X}} \mathcal{F}\left(f(\boldsymbol{x}), \partial^{\boldsymbol{\alpha}} f; |\boldsymbol{\alpha}| \leq k\right) \mathrm{d}\boldsymbol{x}. \tag{26}$$

Let $f^*(\boldsymbol{x})$ denote the optimizer of the cost. Consider the perturbations $f(\boldsymbol{x}) = f^*(\boldsymbol{x}) + \epsilon\eta(\boldsymbol{x})$, characterized by $\eta(\boldsymbol{x})$ drawn from the family of compactly supported and infinitely differentiable functions. Then, the second-order Taylor-series approximation of $g(\epsilon) = \mathcal{L}(f(\boldsymbol{x})) = \mathcal{L}(f^*(\boldsymbol{x}) + \epsilon\eta(\boldsymbol{x}))$ is given by

$$g(\epsilon) = \mathcal{L}\left(f^*(\boldsymbol{x}) + \epsilon\eta(\boldsymbol{x})\right)$$
$$= \mathcal{L}\left(f^*(\boldsymbol{x})\right) + \epsilon \, \partial\mathcal{L}(f^*; \eta) + \frac{1}{2}\epsilon^2 \, \partial^2\mathcal{L}(f^*; \eta),$$

where $\partial\mathcal{L}(\cdot)$ and $\partial^2\mathcal{L}(\cdot)$ denote the first variation and second variation of $\mathcal{L}$, respectively, and can be evaluated through the scalar optimization problems (Gelfand & Fomin, 1964):

$$\partial\mathcal{L}(f^*(\boldsymbol{x})) = g'(0) = \left.\frac{\partial g(\epsilon)}{\partial\epsilon}\right|_{\epsilon=0}, \quad \text{and}$$
$$\partial^2\mathcal{L}(f^*(\boldsymbol{x})) = g''(0) = \left.\frac{\partial^2 g(\epsilon)}{\partial\epsilon^2}\right|_{\epsilon=0},$$

respectively. Evaluating $\partial \mathcal{L}(f^*(\boldsymbol{x}))$ corresponding to Eq. (26) and setting it equal to zero yields the Euler-Lagrange condition (first-order necessary condition) that the optimizer $f^*$ must satisfy (cf. Eq. (14), *Main Manuscript*). The second-order Legendre condition for $f^*(\boldsymbol{x})$ to be a minimizer of the cost $\mathcal{L}$ is $g''(0) > 0$ (Gelfand & Fomin, 1964).

We now derive the sign of the optimal Lagrange multiplier. Recall the discriminator cost:

$$\mathcal{L}_D = \int_{\mathcal{X}} D(\boldsymbol{x}) \left(p_g(\boldsymbol{x}) - p_d(\boldsymbol{x})\right) + \lambda_d \left(\|\nabla^m D(\boldsymbol{x})\|_2^2 - 1\right) \ \mathrm{d}\boldsymbol{x}.$$

The scalar function associated with the above cost is

$$g_D(\epsilon) = \int_{\mathcal{X}} \left(\left(D^*(\boldsymbol{x}) + \epsilon\eta(\boldsymbol{x})\right)\left(p_g(\boldsymbol{x}) - p_d(\boldsymbol{x})\right) + \lambda_d \left(\|\nabla^m \left(D^*(\boldsymbol{x}) + \epsilon\eta(\boldsymbol{x})\right)\|_2^2 - 1\right)\right) \ \mathrm{d}\boldsymbol{x}$$

$$= \int_{\mathcal{X}} \left(\left(D^*(\boldsymbol{x}) + \epsilon\eta(\boldsymbol{x})\right)\left(p_g(\boldsymbol{x}) - p_d(\boldsymbol{x})\right) + \lambda_d \sum_{\boldsymbol{\alpha}: \ |\boldsymbol{\alpha}|=m} \left(\frac{m!}{\boldsymbol{\alpha}!}\right) \left(\partial^{\boldsymbol{\alpha}} \left(D^*(\boldsymbol{x}) + \epsilon\eta(\boldsymbol{x})\right)\right)^2 - \lambda_d\right) \ \mathrm{d}\boldsymbol{x}.$$

Differentiating with respect to $\epsilon$ yields

$$g_D'(\epsilon) = \frac{\partial}{\partial \epsilon} \int_{\mathcal{X}} \left(\left(D^*(\boldsymbol{x}) + \epsilon\eta(\boldsymbol{x})\right)\left(p_g(\boldsymbol{x}) - p_d(\boldsymbol{x})\right) + \lambda_d \sum_{\substack{\boldsymbol{\alpha} \\ |\boldsymbol{\alpha}|=m}} \left(\frac{m!}{\boldsymbol{\alpha}!}\right) \left(\partial^{\boldsymbol{\alpha}} \left(D^*(\boldsymbol{x}) + \epsilon\eta(\boldsymbol{x})\right)\right)^2 - \lambda_d\right) \ \mathrm{d}\boldsymbol{x}.$$

$$= \int_{\mathcal{X}} \left(\eta(\boldsymbol{x})\left(p_g(\boldsymbol{x}) - p_d(\boldsymbol{x})\right) + 2\lambda_d \sum_{\substack{\boldsymbol{\alpha} \\ |\boldsymbol{\alpha}|=m}} \left(\frac{m!}{\boldsymbol{\alpha}!}\right) \left(\partial^{\boldsymbol{\alpha}} \left(D^*(\boldsymbol{x}) + \epsilon\eta(\boldsymbol{x})\right)\right) \left(\partial^{\boldsymbol{\alpha}} \eta(\boldsymbol{x})\right) \right) \mathrm{d}\boldsymbol{x}.$$

The second variation can be obtained by differentiating the above with respect to $\epsilon$ and equating it to zero. The second derivative of the scalar function $g$ is given by

$$g_D''(\epsilon) = \frac{\partial}{\partial \epsilon} g_D'(\epsilon) = 2\lambda_d \int_{\mathcal{X}} \sum_{\substack{\boldsymbol{\alpha} \\ |\boldsymbol{\alpha}|=m}} \left(\frac{m!}{\boldsymbol{\alpha}!}\right) \partial^{\boldsymbol{\alpha}} \left(\frac{\partial \left(D^*(\boldsymbol{x}) + \epsilon\eta(\boldsymbol{x})\right)}{\partial \epsilon}\right) \partial^{\boldsymbol{\alpha}} \eta(\boldsymbol{x}) \ \mathrm{d}\boldsymbol{x}$$

$$= 2\lambda_d \int_{\mathcal{X}} \sum_{\substack{\boldsymbol{\alpha} \\ |\boldsymbol{\alpha}|=m}} \left(\frac{m!}{\boldsymbol{\alpha}!}\right) \left(\partial^{\boldsymbol{\alpha}} \eta(\boldsymbol{x})\right)^2 \ \mathrm{d}\boldsymbol{x}$$

$$= 2\lambda_d \int_{\mathcal{X}} \|\nabla^m \eta(\boldsymbol{x})\|_2^2 \ \mathrm{d}\boldsymbol{x}.$$

The Legendre condition for $D^*(\boldsymbol{x})$ to be a minimizer of $\mathcal{L}_D$ is then given by

$$g_D''(0) = 2\lambda_d \int_{\mathcal{X}} \|\nabla^m \eta(\boldsymbol{x})\|_2^2 \ \mathrm{d}\boldsymbol{x} > 0,$$

which must be true for all compactly supported, infinitely differentiable functions $\eta(\boldsymbol{x})$. Therefore, we have $\lambda_d^* > 0$. The following bound holds on the sample estimate of the optimal Lagrange multiplier given in Eq. (25):

$$0 < |\tilde{\lambda}_d^*| \leq \left|\frac{\varepsilon\xi}{M}\right| \frac{\sqrt{(m!)S_{\boldsymbol{\alpha}}}}{N\sqrt{\mathrm{K}}} \left(\sum_{\ell=1}^{M} \left(\sum_{\substack{\boldsymbol{c}_i \sim p_g \\ i=1}}^{N} \psi_{-n}(\boldsymbol{x}_\ell - \boldsymbol{c}_i) - \sum_{\substack{\boldsymbol{c}_j \sim p_d \\ j=1}}^{N} \psi_{-n}(\boldsymbol{x}_\ell - \boldsymbol{c}_j)\right)^2\right)^{\frac{1}{2}}. \tag{27}$$

One could consider the right-hand side of the above inequality as the worst case bound on $\tilde{\lambda}_d^*$. Substituting for $\tilde{\lambda}_d^*$ in $\tilde{D}_p^*(\boldsymbol{x})$ yields

$$\tilde{D}_p^*(\boldsymbol{x}) = \left( \frac{\xi M \mathrm{K}^{\frac{1}{2}}}{|\xi| \varepsilon \, (m! \, S_{\boldsymbol{\alpha}})^{\frac{1}{2}}} \right) \frac{\displaystyle\sum_{\boldsymbol{c}_i \sim p_g} \psi_{2m-n}(\boldsymbol{x} - \boldsymbol{c}_i) - \sum_{\boldsymbol{c}_j \sim p_d} \psi_{2m-n}(\boldsymbol{x} - \boldsymbol{c}_j)}{\sqrt{\displaystyle\sum_{\ell=1}^{M} \left( \sum_{\boldsymbol{c}_i \sim p_g} \psi_{-n}(\boldsymbol{x}_\ell - \boldsymbol{c}_i) - \sum_{\boldsymbol{c}_j \sim p_d} \psi_{-n}(\boldsymbol{x}_\ell - \boldsymbol{c}_j) \right)^2}}$$

$$= \left( \frac{\mathrm{sgn}(\xi) M \mathrm{K}^{\frac{1}{2}}}{\varepsilon \, (m! \, S_{\boldsymbol{\alpha}})^{\frac{1}{2}}} \right) \frac{S_{p_g} - S_{p_d}}{\sqrt{\displaystyle\sum_{\ell=1}^{M} \left( \sum_{\boldsymbol{c}_i \sim p_g} \psi_{-n}(\boldsymbol{x}_\ell - \boldsymbol{c}_i) - \sum_{\boldsymbol{c}_j \sim p_d} \psi_{-n}(\boldsymbol{x}_\ell - \boldsymbol{c}_j) \right)^2}},$$

where $\mathrm{sgn}(\cdot)$ denotes the signum function. This gives a closed-form solution to the polyharmonic PDE. However, in practice, from experiments on synthetic Gaussian learning presented in Section 6 of the *Main Manuscript*, we observe that $\tilde{\lambda}_d^*$ can be ignored, and its effect can be accounted for by the choice of the learning rate of the generator optimizer.

### C.3 Optimal Generator Distribution

We now present the derivation of the optimal generator distribution, given the optimal discriminator. The derivation is along the same lines as presented by Anonymous (2023). Consider the Lagrangian of the generator loss, described in Section 4.2 of the *Main Manuscript*:

$$\mathcal{L}_G = \int_{\mathcal{X}} D^*(\boldsymbol{x}) \, (p_d(\boldsymbol{x}) - p_g(\boldsymbol{x})) + (\lambda_p + \mu_p(\boldsymbol{x})) \, p_g(\boldsymbol{x}) \mathrm{d}\boldsymbol{x} - \lambda_p.$$

Since the integral cost in turn involves a convolution integral, the Euler-Lagrange condition cannot be applied readily. Instead, the optimum must be obtained from first principles. Let $p_g^*(\boldsymbol{x})$ be the optimal solution. Consider the perturbed version $p_g(\boldsymbol{x}) = p_g^*(\boldsymbol{x}) + \epsilon \eta(\boldsymbol{x})$, where $\eta(\boldsymbol{x})$ is drawn from a family of compactly supported, absolutely integrable, infinitely differentiable functions that vanish on the boundary of $\mathcal{X}$. The corresponding perturbed loss is given by

$$\mathcal{L}_{G,\epsilon}(p_g) = \mathcal{L}_G(p_g^*(\boldsymbol{x}) + \epsilon \eta(\boldsymbol{x}))$$
$$= \int_{\mathcal{X}} \left( D_\epsilon^*(\boldsymbol{x}) \, (p_d(\boldsymbol{x}) - p_g^*(\boldsymbol{x}) - \epsilon \eta(\boldsymbol{x})) + (\lambda_p + \mu_p(\boldsymbol{x})) \, (p_g^*(\boldsymbol{x}) - \epsilon \eta(\boldsymbol{x})) \right) \, \mathrm{d}\boldsymbol{x} - \lambda_p,$$

where $D_\epsilon^*(\boldsymbol{x})$ is the optimal discriminator corresponding to the perturbed generator and is given by

$$D_\epsilon^*(\boldsymbol{x}) = -\frac{\xi}{\lambda_d^*} \left( (p_d - p_g^* - \epsilon \eta) * \psi_{2m-n} \right)(\boldsymbol{x}) + P(\boldsymbol{x}).$$

The derivatives of $\mathcal{L}_{G,\epsilon}$ and $D_\epsilon^*(\boldsymbol{x})$ with respect to $\epsilon$ are given by

$$\frac{\mathrm{d}\mathcal{L}_{G,\epsilon}}{\mathrm{d}\epsilon} = \int_{\mathcal{X}} \left( \frac{\mathrm{d}D_\epsilon^*(\boldsymbol{x})}{\mathrm{d}\epsilon} \, (p_d(\boldsymbol{x}) - p_g^*(\boldsymbol{x}) - \epsilon \eta(\boldsymbol{x})) + (\lambda_p + \mu_p(\boldsymbol{x}) - D_\epsilon^*(\boldsymbol{x})) \, \eta(\boldsymbol{x}) \right) \, \mathrm{d}\boldsymbol{x}, \text{ and}$$
$$\frac{\mathrm{d}D_\epsilon^*(\boldsymbol{x})}{\mathrm{d}\epsilon} = \frac{\xi}{\lambda_d^*} \left( \eta * \psi_{2m-n} \right)(\boldsymbol{x}),$$

respectively. The first variation of the loss $\mathcal{L}_G$, denoted by $\partial \mathcal{L}_G$, is given by $\partial \mathcal{L}_G = \left. \frac{\mathrm{d}\mathcal{L}_{G,\epsilon}}{\mathrm{d}\epsilon} \right|_{\epsilon=0}$. For the loss at hand, the first variation is given by

$$\partial \mathcal{L}_G = \underbrace{\frac{\xi}{\lambda_d^*} \int_{\mathcal{X}} (\eta * \psi_{2m-n})(\boldsymbol{x}) \, (p_d(\boldsymbol{x}) - p_g^*(\boldsymbol{x})) \mathrm{d}\boldsymbol{x}}_{\mathrm{T}_0} + \left( \lambda_p + \mu_p(\boldsymbol{x}) + \frac{\xi}{\lambda_d^*} \left( (p_d - p_g^*) * \psi_{2m-n} \right)(\boldsymbol{x}) - P(\boldsymbol{x}) \right) \eta(\boldsymbol{x}) \mathrm{d}\boldsymbol{x}.$$

Consider the term

$$\mathrm{T}_0 = \frac{\xi}{\lambda_d^*} \int_{\boldsymbol{x} \in \mathcal{X}} \int_{\boldsymbol{y} \in \mathcal{X}} \eta(\boldsymbol{y}) \, \psi_{2m-n}(\boldsymbol{x} - \boldsymbol{y}) \left( p_d(\boldsymbol{x}) - p_g^*(\boldsymbol{x}) \right) \, \mathrm{d}\boldsymbol{y} \, \mathrm{d}\boldsymbol{x},$$

with the convolution integral expanded. Swapping the order of integration requires absolutely integrability over the domain of interest $\mathcal{X}$. Assume $p_d$ and $p_g$ to be compactly supported, *i.e.,* $\mathcal{X}$ is compact. This is a reasonable assumption even in practice because the data always has a finite dynamic range, pixel intensities of images, for instance. The family of perturbations $\eta(\boldsymbol{x})$ is assumed to be compactly supported and absolutely integrable over $\mathcal{X}$. It remains to show that the fundamental solution $r(\boldsymbol{x})$ is finite-valued over $\mathcal{X}$. Consider the case when $2m - n > 0$. Then, $r(\boldsymbol{x})$ is absolutely integrable over $\mathcal{X}$ for odd $n$. When $n$ is even, we consider the following approximation (Fasshauer, 2007; Iske, 2004):

$$\|\boldsymbol{x}\|^{2m-n} \ln(\|\boldsymbol{x}\|) \approx \begin{cases} \|\boldsymbol{x}\|^{2m-n-1} & \text{for } \|\boldsymbol{x}\| < 1, \\ \|\boldsymbol{x}\|^{2m-n} \ln(\|\boldsymbol{x}\|) & \text{for } \|\boldsymbol{x}\| \geq 1, \end{cases}$$

which overcomes the singularity of $\ln(\|\boldsymbol{x}\|)$ at the origin. With this approximation, $r(\boldsymbol{x})$ becomes finite. By Fubini's theorem, the order of integration can be swapped resulting in

$$\mathrm{T}_0 = \frac{\xi}{\lambda_d^*} \int_{\boldsymbol{y} \in \mathcal{X}} \int_{\boldsymbol{x} \in \mathcal{X}} \eta(\boldsymbol{y}) \, \psi_{2m-n}(\boldsymbol{x} - \boldsymbol{y}) \left( p_d(\boldsymbol{x}) - p_g^*(\boldsymbol{x}) \right) \, \mathrm{d}\boldsymbol{x} \, \mathrm{d}\boldsymbol{y}.$$

Owing to radial symmetry of $r$, we write

$$\mathrm{T}_0 = \frac{\xi}{\lambda_d^*} \int_{\boldsymbol{y} \in \mathcal{X}} \eta(\boldsymbol{y}) \int_{\boldsymbol{x} \in \mathcal{X}} \psi_{2m-n}(\boldsymbol{y} - \boldsymbol{x}) \left( p_d(\boldsymbol{x}) - p_g^*(\boldsymbol{x}) \right) \, \mathrm{d}\boldsymbol{x} \, \mathrm{d}\boldsymbol{y},$$

$$= \frac{\xi}{\lambda_d^*} \int_{\boldsymbol{y} \in \mathcal{X}} \eta(\boldsymbol{y}) \left( \left( p_d - p_g^* \right) * \psi_{2m-n} \right)(\boldsymbol{y}) \, \mathrm{d}\boldsymbol{y}.$$

For the case when $2m - n \leq 0$, the above analysis holds on $\mathcal{X} - \mathcal{B}_{\mathbf{0},\delta}$, where $\mathcal{B}_{\mathbf{0},\delta}$ represents a ball of radius $\delta$ centered around the origin (which is where the singularity is). Substituting $\mathrm{T}_0$ back into $\partial \mathcal{L}_G$ yields

$$\partial \mathcal{L}_G = \int_{\mathcal{X}} \left( \lambda_p + \mu_p(\boldsymbol{x}) + \frac{\xi}{\lambda_d^*} \left( \left( p_d - p_g^* \right) * \psi_{2m-n} \right)(\boldsymbol{x}) - P(\boldsymbol{x}) \right) \eta(\boldsymbol{x}) \mathrm{d}\boldsymbol{x}$$

$$= 0,$$

where the second equality is due to the fact that, when $\epsilon = 0$, $p_g = p_g^*$, which implies that $\partial \mathcal{L}_G = 0$. By the *Fundamental Lemma of Calculus of Variations* (Gelfand & Fomin, 1964), we have

$$\lambda_p + \mu_p(\boldsymbol{x}) + \frac{\xi}{\lambda_d^*} \left( \left( p_d - p_g^* \right) * \psi_{2m-n} \right)(\boldsymbol{x}) - P(\boldsymbol{x}) = 0.$$

Rearranging terms, we get

$$\left( p_g^* * \psi_{2m-n} \right)(\boldsymbol{x}) = \left( p_d * \psi_{2m-n} \right)(\boldsymbol{x}) + \left( \frac{\lambda_d^*}{\xi} \right) \left( \lambda_p + \mu_p(\boldsymbol{x}) - P(\boldsymbol{x}) \right). \tag{28}$$

In order to "deconvolve" the effect of $r(\boldsymbol{x})$ on $p_g^*$, we take advantage of the following property of the polyharmonic operator: $\Delta^m r(\boldsymbol{x}) = \delta(\boldsymbol{x})$. Applying $\Delta^m$ to both sides of Eq. (28) yields:

$$p_g^*(\boldsymbol{x}) = p_d(\boldsymbol{x}) + \left( \frac{\lambda_d^*}{\xi} \right) \Delta^m \mu_p(\boldsymbol{x}), \tag{29}$$

where $\Delta^m P(\boldsymbol{x}) = 0$, since $P(\boldsymbol{x})$ is an $(m-1)$-degree polynomial. This implies that the optimal generator distribution $p_g^*$ is independent of the choice of the homogeneous component $P(\boldsymbol{x}) \in \mathcal{P}_{m-1}^n(\boldsymbol{x})$. The solution is

also independent of $\lambda_p$. We now focus our attention on computing $\mu_p^*(\boldsymbol{x})$. Applying the integral constraint $\Omega_P$ on $p_g^*(\boldsymbol{x})$ gives

$$\int_{\mathcal{X}} p_g^*(\boldsymbol{x}) \, \mathrm{d}\boldsymbol{x} = \int_{\mathcal{X}} p_d(\boldsymbol{x}) + \left(\frac{\lambda_d^*}{\xi}\right) \Delta^m \mu_p(\boldsymbol{x}) \, \mathrm{d}\boldsymbol{x} = 1,$$

$$\Rightarrow \int_{\mathcal{X}} \Delta^m \mu_p(\boldsymbol{x}) \, \mathrm{d}\boldsymbol{x} = 0. \tag{30}$$

The non-negativity constraint implies that $\mu_p(\boldsymbol{x}) \leq 0$, $\forall \, \boldsymbol{x} \in \mathcal{X}$. Further, from the complementary slackness condition, we have

$$\mu_p^*(\boldsymbol{x}) \, p_g^*(\boldsymbol{x}) = \mu_p^*(\boldsymbol{x}) \, p_d(\boldsymbol{x}) + \left(\frac{\lambda_d^*}{\xi}\right) \mu_p^*(\boldsymbol{x}) \, \Delta^m \mu_p^*(\boldsymbol{x}) = 0, \tag{31}$$

for all $\boldsymbol{x} \in \mathcal{X}$. Two scenarios arise: (a) $p_d(\boldsymbol{x}) = 0$; and (b) $p_d(\boldsymbol{x}) > 0$. The solutions $\mu_p^*(\boldsymbol{x})$ that satisfy the conditions in Equations (30) and (31) are:

$$\mu_p^*(\boldsymbol{x}) = 0, \ \forall \, \boldsymbol{x} \in \mathcal{X}, \quad \text{or}$$

$$\mu_p^*(\boldsymbol{x}) = \begin{cases} 0, & \forall \, \boldsymbol{x} \text{ such that } p_d(\boldsymbol{x}) > 0, \\ Q(\boldsymbol{x}) \in \mathcal{P}_{m-1}^n(\boldsymbol{x}), & \forall \, \boldsymbol{x} \text{ such that } p_d(\boldsymbol{x}) = 0, \end{cases}$$

where $Q(\boldsymbol{x})$ must be a non-positive polynomial of degree $m - 1$, i.e., $Q(\boldsymbol{x}) \leq 0 \ \forall \, \boldsymbol{x}$, such that $p_d(\boldsymbol{x}) = 0$. In either case, $p_g^*(\boldsymbol{x}) = p_d(\boldsymbol{x})$, i.e., the optimal generator distribution that minimizes the chosen cost subject to non-negativity and integral constraints is indeed the data distribution. This completes the proof of Theorem 4.2.

## C.4 Sample Estimate of the Optimal Discriminator

Consider the closed-form optimal discriminator given in Equation (9):

$$D_p^*(\boldsymbol{x}) = \frac{(-1)^{m+1}\varrho}{2\lambda_d^*} \left( (p_g - p_d) * \psi_{2m-n} \right)(\boldsymbol{x}).$$

Without loss of generality, we assume that $n$ is odd. From the definition of the convolution, we have

$$D_p^*(\boldsymbol{x}) = \frac{(-1)^{m+1}\varrho}{2\lambda_d^*} \int_{\mathcal{X}} (p_g(\boldsymbol{y}) - p_d(\boldsymbol{y})) \, \|\boldsymbol{x} - \boldsymbol{y}\|^{2m-n} \, \mathrm{d}\boldsymbol{y}$$

$$= \frac{\xi}{\lambda_d^*} \left( \mathbb{E}_{\boldsymbol{y} \sim p_g} \left[ \|\boldsymbol{x} - \boldsymbol{y}\|^{2m-n} \right] - \mathbb{E}_{\boldsymbol{y} \sim p_d} \left[ \|\boldsymbol{x} - \boldsymbol{y}\|^{2m-n} \right] \right),$$

where $\xi = \frac{(-1)^{m+1}\varrho}{2}$. The expectations can be replaced with $N$-sample estimates as follows:

$$\tilde{D}_p^*(\boldsymbol{x}) = \frac{\xi}{\lambda_d^* N} \left( \sum_{\boldsymbol{c}_i \sim p_g} \|\boldsymbol{x} - \boldsymbol{c}_i\|^{2m-n} - \frac{\xi}{\lambda_d^* N} \sum_{\boldsymbol{c}_j \sim p_d} \|\boldsymbol{x} - \boldsymbol{c}_j\|^{2m-n} \right),$$

where $\tilde{D}_p^*$ is a polyharmonic RBF expansion. A similar analysis could be carried out for even $n$, and the corresponding discriminators is:

$$\tilde{D}_p^*(\boldsymbol{x}) = \frac{\xi}{\lambda_d^* N} \left( \sum_{\boldsymbol{c}_i \sim p_g} \|\boldsymbol{x} - \boldsymbol{c}_i\|^{2m-n} \ln(\|\boldsymbol{x} - \boldsymbol{c}_i\|) - \sum_{\boldsymbol{c}_j \sim p_d} \|\boldsymbol{x} - \boldsymbol{c}_j\|^{2m-n} \ln(\|\boldsymbol{x} - \boldsymbol{c}_j\|) \right)$$

The generic form for $\tilde{D}_p^*$ is given by The generic form for $\tilde{D}_p^*$ is given by

$$\tilde{D}_p^*(\boldsymbol{x}) = \frac{\xi}{\lambda_d^* N} \left( \sum_{\boldsymbol{c}_i \sim p_g} \psi_{2m-n}(\boldsymbol{x} - \boldsymbol{c}_i) - \sum_{\boldsymbol{c}_j \sim p_d} \psi_{2m-n}(\boldsymbol{x} - \boldsymbol{c}_j) \right) \tag{32}$$

which completes the proof of Theorem 4.3.

### C.5 Practical considerations

In this appendix, we discuss additional practical considerations in implementing the polyharmonic RBF discriminator. In particular, we discuss the choice of the homogeneous component $P(\boldsymbol{x})$ and the gradient order $m$.

***Issues with the Homogeneous Component***: The polynomial term $P(\boldsymbol{x})$ in the PolyGAN discriminator represents the homogeneous component of the solution. While in Poly-LSGAN, the coefficients can be computed via matrix inversion, in Poly-WGAN, boundary conditions must be defined to determine the optimal values of the coefficients. In either case, as discussed in Section 3.1, the number of coefficients grows exponentially with both the data dimension and the gradient order, which makes it impractical to incorporate the homogeneous component in the solution. We argue that dropping the homogeneous component does not significantly impact the gradient-descent optimization. The justification is as follows. The result provided in Theorem 4.2 shows that the optimal generator is independent of the homogeneous component. As far as gradient-descent is concerned, ignoring the homogeneous component is not too detrimental to the optimization. Consider the generator optimization in practice, given the empirical loss:

$$\hat{\mathcal{L}}_G(\theta) = \frac{\xi}{\lambda_d^* NM} \sum_{\boldsymbol{z}_k \sim p_Z(\boldsymbol{z})} \left( \sum_{\boldsymbol{c}_i \sim p_g} \psi_{2m-n}(G_\theta(\boldsymbol{z}_k) - \boldsymbol{c}_i) - \sum_{\boldsymbol{c}_j \sim p_d} \psi_{2m-n}(G_\theta(\boldsymbol{z}_k) - \boldsymbol{c}_j) + P(G_\theta(\boldsymbol{z}_k)) \right),$$

obtained by simplifying Equation (11), considering only those terms that involve the generator $G_\theta$, parameterized by $\theta$. Updating the generator parameters $\theta_t \to \theta_{t+1}$ through first-order methods such as gradient-descent involve locally linear approximation of the loss surface $\mathcal{L}_G$ about the point of interest $\theta_t$ giving rise to the update $\theta_{t+1} = \theta_t + \tau \nabla_\theta \hat{\mathcal{L}}_G(\theta_t)$, where $\nabla_\theta \hat{\mathcal{L}}_G(\theta_t)$ denotes the gradient of the loss evaluated at $\theta = \theta_t$ and $\tau$ is the learning rate parameter. The gradient of the loss involves the derivatives of the kernel $\partial_{x_i} \psi_{2m-n}(\boldsymbol{x} - \cdot)$, and the polynomial $\partial_{x_i} P(\boldsymbol{x})$. Given a gradient direction associated with the particular solution, the gradient of the homogeneous component serves as a correction term, the effect of which can be neglected when the learning rate $\tau$ is small. As iterations progress and the optimization converges, the effect of the polynomial term in the discriminator diminishes as the optimal WGAN discriminator is a constant function (Arjovsky et al., 2017). In view of the above considerations, we do not incorporate the the homogeneous component $P(\boldsymbol{x})$ in the Poly-WGAN discriminator.

***Choice of the Gradient Order***: For $2m - n > 0$, the RBFs $\{\psi_{2m-n}(\boldsymbol{x} - \boldsymbol{c}_\ell)\}$ increase with $\boldsymbol{x}$, which might result in large gradients particularly in the initial phases of training when $p_g$ is away from $p_d$. On the other hand, if $2m - n \leq 0$, $\psi_{2m-n}(\boldsymbol{x} - \boldsymbol{c}_\ell)$ has a singularity at $\boldsymbol{x} = \boldsymbol{c}_\ell$, which could result in convergence issues in the later stages of training. Experimental results in support of this claim are presented in Section 6.2. Though these observations were empirical, they can also be explained through the Sobolev embeddings into continuous spaces. It is known that functions in the $L_2$-normed Sobolev spaces of order $m$, $\mathrm{W}^{m,2}$ will be Hölder continuous, *i.e.,* $f \in \mathrm{C}^{R,\alpha}$ such that $|f(\boldsymbol{x}) - f(\boldsymbol{y})| = R\|\boldsymbol{x} - \boldsymbol{y}\|^\alpha$, where $R, \alpha > 0$, and $R + \alpha = m - \frac{n}{2}$ (Stein, 1970). If the discriminator has its $m^{th}$ order derivatives bounded in $L_2$-norm and $m > \frac{n}{2}$, then the discriminator will be continuous. Additionally, for $\alpha = 1$, we get Lipschitz discriminators. For an in-depth analysis on the embedding of Beppo-Levi spaces in Hölder-Zygmund spaces, the reader is referred to Beatson et al. (2005). In a similar vein, the relationship between Sobolev embeddings of the discriminator and generator in Sobolev GANs was explored by Liang (2021).

## D   Implementation Details

In this section, we provide details regarding the network architectures and training parameters associated with the experiments reported, and the Poly-LSGAN, Poly-WGAN and PolyGAN-WAE training algorithms.

### D.1   Network Architectures

Tables 3-8 describe the network architectures, a summary of which is given below.

**2-D Gaussians and GMMs**: The generator accepts 100-D standard Gaussian data as input. The network consists of three fully connected layers, with 64, 32, and 16 nodes. The activation in each layer is ReLU. The

output layer consists of two nodes. The discriminator is also a three-layer fully connected ReLU network with 10, 20, and 5 nodes, in order. The discriminator outputs a 1-D prediction. Table 3 depicts these architectures.

**$n$-D Gaussians**: In order to simulate DCGAN (Radford et al., 2016) based image generation, we use a convolutional neural network for the generator as shown in Table 4. The 100-dimensional Gaussian data is input to a fully connected layer with $32 \times 32 \times 3 = 3072$ nodes and subsequently reshaped to $32 \times 32 \times 3$, and provided as input to five convolution layers. Each convolution filter is of size $4 \times 4$ and a stride of two resulting in a downsampling of the input by a factor of two. All convolution layers include batch normalization (Ioffe & Szegedy, 2015). The discriminator is a four-layer fully-connected network with 512, 256, 64 and 32 nodes.

**Autoencoder architecture**: For the MNIST learning task, as shown in Table 5, we consider a 4-layer fully connected network with leaky ReLU activation for the encoder with 784, 256, 128 and 64 nodes in the first, second, third, and fourth layers, respectively. The decoder has a similar architecture but exactly in the reverse order. For CIFAR-10 and CelebA learning, the convolutional autoencoder architectures based on DCGAN are used as shown in Tables 6 and 7. For LSUN-Churches, we consider a convolutional ResNet architecture. As shown in Table 8, the encoder consists of four ResNet convolution layers with both batch and spectral normalization (Roth et al., 2019). The decoder similarly consists of ResNet deconvolution layers. The CelebA and LSUN-Churches images are center-cropped and resized to $64 \times 64 \times 3$ using built-in bilinear interpolation. We employ the ResNet based BigGAN architecture (Tolstikhin et al., 2018) from Table 8 for high-resolution ($192 \times 192$) experiments on CelebA, presented in Appendix F.4. In experiments involving the Wasserstein autoencoder with a discriminator network, the discriminator uses the standard DCGAN architecture.

Table 3: GAN architectural details for 2-D Gaussian and Gaussian mixture learning tasks.

| | Layer | Batch Norm | Activation | Output size |
|---|---|---|---|---|
| **Generator** | Input | - | - | (100,1) |
| | Dense 1 | ✗ | ReLU | (64,1) |
| | Dense 2 | ✗ | ReLU | (32,1) |
| | Dense 3 | ✗ | ReLU | (16,1) |
| | Output | ✗ | none | (2,1) |
| **Discriminator** | Input | - | - | (2,1) |
| | Dense 1 | ✗ | ReLU | (10,1) |
| | Dense 2 | ✗ | ReLU | (20,1) |
| | Dense 3 | ✗ | ReLU | (5,1) |
| | Output | ✗ | none | (1,1) |

Table 4: GAN architectural details for $n$-dimensional Gaussian learning tasks.

| | Layer | Batch Norm | Filters | (Size, Stride) | Activation | Output size |
|---|---|---|---|---|---|---|
| **Generator** | Input | - | - | - | - | (100,1) |
| | Dense 1 | ✗ | - | - | leaky ReLU | $(32 \times 32 \times 3,1)$ |
| | Reshape | - | - | - | - | (32,32,3) |
| | Conv2D 1 | ✓ | 1024 | (4, 2) | leaky ReLU | (16,16,1024) |
| | Conv2D 2 | ✓ | 256 | (4, 2) | leaky ReLU | (8,8,256) |
| | Conv2D 3 | ✓ | 128 | (4, 2) | leaky ReLU | (4,4,128) |
| | Conv2D 4 | ✓ | 128 | (4, 2) | leaky ReLU | (2,2,128) |
| | Conv2D 5 | ✓ | $n$ | (4, 2) | leaky ReLU | (1,1,$n$) |
| | Flatten | - | - | - | - | ($n$,1) |
| **Discriminator** | Input | - | - | - | - | ($n$,1) |
| | Dense 1 | ✗ | - | - | leaky ReLU | (512,1) |
| | Dense 2 | ✗ | - | - | leaky ReLU | (256,1) |
| | Dense 3 | ✗ | - | - | leaky ReLU | (64,1) |
| | Dense 4 | ✗ | - | - | none | (32,1) |
| | Output | ✗ | - | - | none | (1,1) |

Table 5: Autoencoder architectural details for learning MNIST with a 11-D latent space.

| | Layer | Batch Norm | Activation | Output size |
|---|---|---|---|---|
| **Encoder** | Input | - | - | (28,28,1) |
| | Flatten | - | - | (784,1) |
| | Dense 1 | ✓ | leaky ReLU | (512,1) |
| | Dense 2 | ✓ | leaky ReLU | (256,1) |
| | Dense 3 | ✓ | leaky ReLU | (128,1) |
| | Dense 4 | ✓ | leaky ReLU | (64,1) |
| | Output | ✗ | none | (11,1) |
| **Decoder** | Input | - | - | (11,1) |
| | Dense 1 | ✓ | leaky ReLU | (64,1) |
| | Dense 2 | ✓ | leaky ReLU | (128,1) |
| | Dense 3 | ✓ | leaky ReLU | (256,1) |
| | Dense 4 | ✓ | leaky ReLU | (784,1) |
| | Reshape | - | - | (28,28,1) |
| | Activation | - | tanh | (28,28,1) |

Table 6: Autoencoder architectural details for learning CIFAR-10 with a 64-D latent space.

| | Layer | Batch Norm | Filters | (Size, Stride) | Activation | Output size |
|---|---|---|---|---|---|---|
| Encoder | Input | - | - | - | - | (32,32,3) |
| | Conv2D 1 | ✓ | 128 | (4, 2) | leaky ReLU | (16,16,128) |
| | Conv2D 2 | ✓ | 256 | (4, 2) | leaky ReLU | (8,8,256) |
| | Conv2D 3 | ✓ | 512 | (4, 2) | leaky ReLU | (4,4,512) |
| | Conv2D 4 | ✓ | 1024 | (4, 2) | leaky ReLU | (2,2,1024) |
| | Flatten | - | - | - | - | $(2 \times 2 \times 1024, 1)$ |
| | Dense 1 | ✗ | - | - | none | (64,1) |
| Decoder | Input | - | - | - | - | (64,1) |
| | Dense 1 | ✗ | - | - | none | $(4 \times 4 \times 1024, 1)$ |
| | Reshape | - | - | - | - | (4,4,1024) |
| | Deconv2D 1 | ✓ | 512 | (4, 2) | leaky ReLU | (8,8,512) |
| | Deconv2D 2 | ✓ | 256 | (4, 2) | leaky ReLU | (16,16,256) |
| | Deconv2D 3 | ✓ | 128 | (4, 2) | leaky ReLU | (32,32,128) |
| | Deconv2D 4 | ✗ | 3 | (4, 1) | tanh | (32,32,3) |

Table 7: Autoencoder architectural details for learning 64-D CelebA with a 128-D latent space.

| | Layer | Batch Norm | Filters | (Size, Stride) | Activation | Output size |
|---|---|---|---|---|---|---|
| Encoder | Input | - | - | - | - | (64,64,3) |
| | Conv2D 1 | ✓ | 128 | (4, 2) | leaky ReLU | (32,32,128) |
| | Conv2D 2 | ✓ | 256 | (4, 2) | leaky ReLU | (16,16,256) |
| | Conv2D 3 | ✓ | 512 | (4, 2) | leaky ReLU | (8,8,512) |
| | Conv2D 4 | ✓ | 1024 | (4, 2) | leaky ReLU | (4,4,1024) |
| | Flatten | - | - | - | - | $(4 \times 4 \times 1024, 1)$ |
| | Dense 1 | ✗ | - | - | none | (128,1) |
| Decoder | Input | - | - | - | - | (128,1) |
| | Dense 1 | ✓ | - | - | none | $(8 \times 8 \times 1024, 1)$ |
| | Reshape | - | - | - | - | (8,8,1024) |
| | Deconv2D 1 | ✓ | 512 | (4, 2) | leaky ReLU | (16,16,512) |
| | Deconv2D 2 | ✓ | 256 | (4, 2) | leaky ReLU | (32,32,256) |
| | Deconv2D 3 | ✓ | 128 | (4, 2) | leaky ReLU | (64,64,128) |
| | Deconv2D 4 | ✗ | 3 | (4, 1) | tanh | (64,64,3) |

Table 8: Autoencoder architectural details for learning 64-D LSUN-Churches with a 128-D latent space.

| | Layer | Spectral Norm | Filters | Activation | Output size |
|---|---|---|---|---|---|
| **Encoder** | Input | - | - | - | (64,64,3) |
| | ResBlock Down 1 | ✓ | 128 | leaky ReLU | (32,32,128) |
| | ResBlock Down 2 | ✓ | 256 | leaky ReLU | (16,16,256) |
| | ResBlock Down 3 | ✓ | 512 | leaky ReLU | (8,8,512) |
| | ResBlock Down 4 | ✓ | 1024 | leaky ReLU | (4,4,1024) |
| | Flatten | - | - | - | $(4 \times 4 \times 1024, 1)$ |
| | Dense 1 | ✗ | - | leaky ReLU | (128,1) |
| | Dense 2 | ✗ | - | none | (128,1) |
| **Decoder** | Input | - | - | - | (128,1) |
| | Dense 1 | ✗ | - | none | $(4 \times 4 \times 1024, 1)$ |
| | Reshape | - | - | - | (4,4,1024) |
| | ResBlock Up 1 | ✓ | 512 | leaky ReLU | (8,8,512) |
| | ResBlock Up 2 | ✓ | 256 | leaky ReLU | (16,16,256) |
| | ResBlock Up 3 | ✓ | 128 | leaky ReLU | (32,32,128) |
| | ResBlock Up 4 | ✓ | 64 | leaky ReLU | (64,64,64) |
| | Conv2D | ✗ | 3 | tanh | (64,64,3) |

### D.2 Training Specifications

**System Specifications**: The codes for both PolyGANs are written in Tensorflow 2.0 (Abadi et al., 2016). All experiments were conducted on workstations with one of two configurations: (I) 256 GB of system RAM and 2×NVIDIA GTX 3090 GPUs with 24 GB of VRAM; or (II) 512 GB of system RAM and 8×NVIDIA Tesla V100 GPUs with 32 GB of VRAM.

**Experiments on 2-D Gaussians with Poly-LSGAN**: On the unimodal learning task, the target is $\mathcal{N}(5\mathbf{1}_2, 1.5\mathbb{I}_2)$, where $\mathbf{1}_2$ is the 2-D vector of ones, and $\mathbb{I}_2$ is the $2 \times 2$ identity matrix. On the GMM learning task, we consider eight components distributed uniformly about the unit circle, each having a standard deviation of 0.02.

On the unimodal Gaussian learning task, the generator is a single layer affine transformation of the noise $\boldsymbol{z} \in \mathbb{R}^2$, given by $\boldsymbol{x} = M\boldsymbol{z} + b$, while on the GMM task, it is a three-layer neural network with Leaky ReLU activations. The discriminator in baseline LSGANs variants is a three-layer neural network with Leaky ReLU activation in both the Gaussian and GMM learning tasks. Poly-LSGAN employs the RBF discriminator while weights are computed by solving the system of equation given in Equation (5), while in Poly-WGAN the weights are constant across all iterations. The networks are trained using the Adam optimizer (Kingma & Ba, 2015) with a learning rate of $\eta_g = 0.002$ for the generator and $\eta_d = 0.0075$ for the discriminator. A batch size of 500 is employed.

**Experiments on 2-D Gaussians with Poly-WGAN**: The experimental setup is as follows. In the unimodal Gaussian learning task, the target distribution is $\mathcal{N}(3.5\mathbf{1}_2, 1.25\mathbb{I}_2)$. For the multimodal learning task, we consider an 8-component Gaussian mixture model (GMM), with components having standard deviation equal to 0.02, identical to the Poly-LSGAN case.

The generators and discriminators in the baselines are three-layer neural networks with Leaky ReLU activation. The noise $\boldsymbol{z}$ is 100-dimensional. Poly-WGAN employs the RBF discriminator whereas the other models use a discriminator network. All models use the ADAM optimizer (Kingma & Ba, 2015) with a learning rate of $\eta_g = 0.002$ for the generator network. The learning rate for the baseline discriminator networks is $\eta_d = 0.0075$. The batch size employed is 500.

**Experiments on Image-space Learning**: On image learning tasks, we employ the DCGAN (Radford et al., 2016) generator, trained using the Adam optimizer. The batch size is set to 100. The generator learning rate is set to $\eta_g = 10^{-4}$. The discriminator is the polyharmonic RBF with a gradient penalty of order $m = \lceil \frac{n+1}{2} \rceil$.

**Experiments on PolyGAN-WAE**: On MNIST, we consider a 4-layer dense-ReLU architecture for the encoder, whereas on CIFAR-10 and CelebA, we use the DCGAN model. For LSUN-Churches, we consider a ResNet based encoder model with spectral normalization (Miyato et al., 2018). The decoder is an inversion of the encoder layers in all cases. We used the published TensorFlow implementations of SWAE (Kolouri et al., 2019) and CWAE (Knop et al., 2020), and the published PyTorch implementations of MMD-GAN (Li et al., 2017a) and MMD-GAN-GP (Bińkowski et al., 2018), while other GAN and GMMN variants were coded anew. The learning rates for all MMD variants follow the specifications provided by Knop et al. (2020), while the learning rates for WAE-GAN and WAAE-LP follow those used by Tolstikhin et al. (2018). The WAE models with a trainable discriminator fail to converge for higher learning rates, and consequently, the WAE-MMD variants are an order (in terms of number of iterations) faster than the WAE-GAN variants.

### D.3 Evaluation Metrics

**Wasserstein-2 distance**: We use the Wasserstein-2 ($\mathcal{W}^{2,2}$) distance between the generator and target distributions to quantify the performance in Gaussian learning (Section 6 of the main manuscript). Given two Gaussians $p_d = \mathcal{N}(\boldsymbol{\mu}_d, \Sigma_d)$ and $p_g = \mathcal{N}(\boldsymbol{\mu}_g, \Sigma_g)$, the Wasserstein-2 distance between them is given by

$$\mathcal{W}^{2,2}(p_d, p_g) = \|\boldsymbol{\mu}_d - \boldsymbol{\mu}_g\|^2 + \text{Tr}\left(\Sigma_d + \Sigma_g - 2\sqrt{\Sigma_d \Sigma_g}\right),$$

where $\text{Tr}(\cdot)$ denotes the trace operator and the matrix square-root is computed via singular-value decomposition. In the case of Gaussian mixture data, $\mathcal{W}^{2,2}(p_d, p_g)$ is computed using a sample estimate provided by the *python optimal transport* library (Flamary et al., 2021).

**Fréchet Inception distance (FID)**: Proposed by Heusel et al. (2018), the FID is used to quantify how *realistic* the samples generated by GANs are. To compute FID, we consider the InceptionV3 (Szegedy et al., 2015) model without the topmost layer, loaded with pre-trained ImageNet (Deng et al., 2009) classification weights. The network accepts inputs ranging from $75 \times 75 \times 3$ to $299 \times 299 \times 3$. We therefore rescale all images to $299 \times 299 \times 3$. Grayscale images are duplicated across the color channels. FID is computed as the $\mathcal{W}^{2,2}$ between the InceptionV3 embeddings of real and fake images. The means and covariances are computed using $10,000$ samples. The publicly available TensorFlow based *Clean-FID* library (Parmar et al., 2021) is used to compute FID.

**Kernel Inception distance (KID)**: Proposed by Bińkowski et al. (2018), the kernel inception distance is an unbiased alternative to FID. The KID computes the squared MMD between the InceptionV3 embeddings, akin to the FID framework. The polynomial kernel $\left(\frac{1}{n}\boldsymbol{x}^{\text{T}}\boldsymbol{y} + 1\right)^3$ is computed over batches of 5000 samples. In the interest of reproducibility, we use the publicly available *Clean-FID* (Parmar et al., 2021) library implementation of KID.

**Average reconstruction error** ($\langle RE \rangle$): The WAE autoencoders are trained using the $\ell_1$ loss between the true samples $\boldsymbol{x} \sim p_d$ and their reconstructions $\tilde{\boldsymbol{x}} = \text{Dec}(\text{Enc}(\boldsymbol{x}))$, given by $\mathcal{L}_{AE} = \|\boldsymbol{x} - \tilde{\boldsymbol{x}}\|_1$. On the MNIST, CIFAR-10, and LSUN-Churches datasets, $\langle RE \rangle$ is computed by averaging $\mathcal{L}_{AE}$ over $10^4$ samples drawn from the predefined *Test* sets, whereas on CelebA, a held-out validation set of $10^4$ images is used.

**Image sharpness**: We employ the approach proposed by Tolstikhin et al. (2018) to compute image sharpness. The edge-map is obtained using the Laplacian operator. The variance in pixel intensities on the edge-map is computed and averaged over batches of 50,000 images to determine the average sharpness.

### D.4   Training Algorithm

The PolyGAN implementations uses a radial basis function (RBF) network as the discriminator. The RBF is implemented as a custom layer in Tensorflow 2.0 (Abadi et al., 2016). The network weights and centers are computed *out-of-the-loop* at each step of the training and updated, resulting in the optimal discriminator at each iteration. The output of the RBF discriminator network is used to train the generator through gradient-descent.

Algorithm 1 summarizes the training procedure of Poly-LSGAN, wherein the weights are polynomial coefficients are computed through matrix inversion, adding additional compute overhead. Algorithm 2 presents the Poly-WGAN training procedure, wherein the weights associated with all centers are identical, as given in Equation (12). Additionally, as discussed in Appendix C.5, the polynomial component can be ignored, allowing for fewer computations. Algorithm 3 summarizes the PolyGAN-WAE training framework, where an autoencoder is used to learn the latent-space representations of the data, on which the Poly-WGAN algorithm is applied.

### D.5   Source Code

The source code for the TensorFlow 2.0 (Abadi et al., 2016) implementation of PolyGAN and high-resolution counterparts of the images presented in the paper will be made available on GitHub as part of the camera-ready submission of the manuscript. For the review process, anonymized source code has been attached as a *PolyGANs.zip* folder in the *Supplementary Material*

---

**Algorithm 1:** Poly-LSGAN − LSGAN with a trainable generator and radial basis function (RBF) discriminator and solvable RBF weights.

---

**Input:** Training data $\boldsymbol{x} \sim p_d$, Gaussian prior distribution $p_z = \mathcal{N}(\mu_{\boldsymbol{z}}, \Sigma_{\boldsymbol{z}})$
**Parameters:** Batch size $M$, optimizer learning rate $\eta$, number of radial basis function (RBF) centers $N$
**Models:** Generator: $\text{Gen}_\theta$; Polyharmonic RBF Discriminator: $D_p^*$.
**while** $\text{Gen}_\theta$ *not converged* **do**
  **Sample:** $\boldsymbol{x} \sim p_d$ – A batch of $M$ real samples.
  **Sample:** $\boldsymbol{z} \sim p_z$ – A batch of $M$ noise samples.
  **Sample:** $\tilde{\boldsymbol{x}} = \text{Gen}_\theta(\boldsymbol{z})$ – Generator output.
  **Sample:** $\boldsymbol{z}_c \sim p_z$ – A batch of $N$ noise samples for computing RBF centers.
  **Sample:** $\boldsymbol{c}_i \sim \text{Gen}_\theta(\boldsymbol{z}_c)$ – A batch of $N$ *generator data* centers for the generator RBF interpolator $S_{p_g}$.
  **Sample:** $\boldsymbol{c}_j \sim p_d$ – A batch of $N$ *target data* centers for the target RBF interpolator $S_{p_d}$.
  **Compute:** Matrices $\mathbf{A}$ and $\mathbf{B}$.
  **Solve: System of equations** for RBF weights $\boldsymbol{w}$ and polynomial coefficients $\boldsymbol{v}$ (Eq. (5)).
  **Update: Discriminator RBF** $D^*$ with centers $\{\boldsymbol{c}_i\}, \{\boldsymbol{c}_j\}$, weights $\boldsymbol{w}$ and coefficients $\boldsymbol{v}$ (Eq. (4)).
  **Evaluate: Generator Loss** $\mathcal{L}_G^{LS}(D_p^*(\boldsymbol{x}), D_p^*(\tilde{\boldsymbol{x}}))$
  **Update: Generator** $\text{Gen}_\theta \leftarrow \eta \nabla_\theta[\mathcal{L}_G]$
**Output:** Samples output by the Generator: $\tilde{\boldsymbol{x}}$

---

**Algorithm 2:** Poly-WGAN − GAN with a trainable generator and radial basis function (RBF) discriminator and fixed RBF weights.

---

**Input:** Training data $\boldsymbol{x} \sim p_d$, Gaussian prior distribution $p_z = \mathcal{N}(\mu_{\boldsymbol{z}}, \Sigma_{\boldsymbol{z}})$
**Parameters:** Batch size $M$, optimizer learning rate $\eta$, number of radial basis function (RBF) centers $N$, gradient order $m$.
**Models:** Generator: $\text{Gen}_\theta$; Polyharmonic RBF Discriminator: $D_p^*$.
**while** $\text{Gen}_\theta$ *not converged* **do**
  **Sample:** $\boldsymbol{x} \sim p_d$ – A batch of $M$ real samples.
  **Sample:** $\boldsymbol{z} \sim p_z$ – A batch of $M$ noise samples.
  **Sample:** $\tilde{\boldsymbol{x}} = \text{Gen}_\theta(\boldsymbol{z})$ – Generator output.
  **Sample:** $\boldsymbol{z}_c \sim p_z$ – A batch of $N$ noise samples for computing RBF centers.
  **Sample:** $\boldsymbol{c}_i \sim \text{Gen}_\theta(\boldsymbol{z}_c)$ – A batch of $N$ *generator data* centers for the generator RBF interpolator $S_{p_g}$.
  **Sample:** $\boldsymbol{c}_j \sim p_d$ – A batch of $N$ *target data* centers for the target RBF interpolator $S_{p_d}$.
  **Compute:** RBF weights $w = \frac{\xi}{N\lambda_d^*}$.
  **Update: Discriminator RBF** $D_p^*$ with centers $\{\boldsymbol{c}_i\}, \{\boldsymbol{c}_j\}$ and weight $w$ (Eqn. (9)).
  **Evaluate: Generator Loss** $\mathcal{L}_G(D_p^*(\boldsymbol{x}), D_p^*(\tilde{\boldsymbol{x}})$
  **Update: Generator** $\text{Gen}_\theta \leftarrow \eta \nabla_\theta[\mathcal{L}_G]$
**Output:** Samples output by the Generator: $\tilde{\boldsymbol{x}}$

---

# E  Additional Experiments On Gaussians

In this section, we present additional experimental results on learning 2-D Gaussians and 2-D Gaussian mixtures with the Poly-LSGAN and Poly-WGAN frameworks. The network architectures and training parameters are as described in Appendix E.2.

## E.1  Learning 2-D Gaussians with Poly-LSGAN

We now evaluate the optimal Poly-LSGAN discriminator on learning synthetic 2-D Gaussian and Gaussian mixture models (GMMs), and subsequently discuss extensions to learning images. We consider polyharmonic spline order $k = 2$. For larger $k$, we encountered numerical instability. We compare against the base LSGAN (Mao et al., 2017), and LSGAN subjected to the gradient penalty (GP) (Gulrajani et al., 2017) , $R_d$

---

**Algorithm 3:** PolyGAN-WAE − Wasserstein autoencoder with a radial basis function discriminator.

---

**Input:** Training data $\boldsymbol{x} \sim p_d$, Gaussian prior distribution $\boldsymbol{z} \sim p_z = \mathcal{N}(\mu_{\boldsymbol{z}}, \Sigma_{\boldsymbol{z}})$
**Parameters:** Batch size $M$, optimizer learning rate $\eta$, number of radial basis function (RBF) centers $N$
**Models:** Encoder/Generator: $\text{Enc}_\phi$; Decoder: $\text{Dec}_\theta$; Polyharmonic RBF Discriminator: $D_p^*$.
**while** $\text{Enc}_\phi, \text{Dec}_\theta$ *not converged* **do**

> **Sample:** $\boldsymbol{x} \sim p_d$ – A batch of $M$ real samples.
> **Sample:** $\tilde{z} = \text{Enc}_\phi(\boldsymbol{x})$ – Latent encoding of real samples.
> **Sample:** $\tilde{\boldsymbol{x}} = \text{Dec}_\theta(\tilde{\boldsymbol{z}})$ – Reconstructed samples.
> **Evaluate: Autoencoder Loss**: $\mathcal{L}_{AE}(\boldsymbol{x}, \tilde{\boldsymbol{x}})$
> **Update: Autoencoder** $\text{Enc}_\phi \leftarrow \eta \nabla_\phi [\mathcal{L}_{AE}]$; $\text{Dec}_\theta \leftarrow \eta \nabla_\theta [\mathcal{L}_{AE}]$
> **Sample:** $\boldsymbol{c}_i \sim \mathcal{N}(\mu_{\boldsymbol{z}}, \Sigma_{\boldsymbol{z}})$ – A batch of $N$ centers for the target RBF interpolator.
> **Sample:** $\boldsymbol{x}_c \sim p_d$ – A batch of $N$ real samples to compute data centers.
> **Sample:** $\boldsymbol{c}_j = \text{Enc}_\phi(\boldsymbol{x}_c) \sim p_{d_\ell}$ – A batch of $N$ centers for the generator RBF interpolator.
> **Compute:** RBF weights $w = \frac{\xi}{N \lambda_d^*}$.
> **Update: Discriminator RBF** $D_p^*$ with centers $\{\boldsymbol{c}_i\}, \{\boldsymbol{c}_j\}$ and weight $w$ (Eqn. (9)).
> **Sample:** $\boldsymbol{z} \sim \mathcal{N}(\mu_{\boldsymbol{z}}, \Sigma_{\boldsymbol{z}})$ – A batch of $M$ prior distribution samples.
> **Evaluate: Generator Loss** $\mathcal{L}_G(D_p^*(\tilde{\boldsymbol{z}}), D_p^*(\boldsymbol{z}))$
> **Update: Generator** $\text{Enc}_\phi \leftarrow \eta \nabla_\phi [\mathcal{L}_G]$

**Output:** Reconstructed random prior samples: $\text{Dec}_\theta(\boldsymbol{z})$

---

and $R_g$ (Mescheder et al., 2018), Lipschitz penalty (LP) (Petzka et al., 2018) and the DRAGAN (Roth et al., 2017) regularizers.

On the unimodal learning task, the target is $\mathcal{N}(5\mathbf{1}_2, 1.5\mathbb{I}_2)$, where $\mathbf{1}_2$ is the 2-D vector of ones, and $\mathbb{I}_2$ is the $2 \times 2$ identity matrix. On the GMM learning task, we consider eight components distributed uniformly about the unit circle, each having a standard deviation of 0.02. To quantify performance, we use the Wasserstein-2 distance between the target and generator distributions $(\mathcal{W}^{2,2}(p_d, p_g))$. Network parameters are given in Appendix D.1. Figure 7 presents the $\mathcal{W}^{2,2}$ distance as a function of iterations on the Gaussian and Gaussian mixture learning tasks. On both datasets, using the polyharmonic RBF discriminator results in superior generator performance (lower $\mathcal{W}^{2,2}$ scores). In all scenarios considered, the polyharmonic RBF discriminator learns the perfect classifier, compared to LSGAN with a trainable network discriminator.

Figures 8 and 9 present the generated and target data samples, superimposed on the level-sets of the discriminator, for the 2-D Gaussian, and 8-component Gaussian mixture learning tasks, respectively. For the Gaussian learning problem, we observe that Poly-LSGAN does not *mode collapse* upon convergence to the target distribution. However, in the baseline GANs, depending on the learning rate, the generator converges to a distribution of smaller support than the target, before latching on to the desired target. Similarly, on the GMM learning task, Poly-LSGAN learns the target distribution more accurately compared to the baselines.

### E.2 Learning 2-D Gaussians with Poly-WGAN

Figure 10 shows the Wasserstein-2 distance $(\mathcal{W}^{2,2}(p_d, p_g))$ as a function of iterations for various number of centers $N$ of the RBFs on the 2-D unimodal and multimodal Gaussian and 8-component GMM learning task. On the unimodal data, we observe that the performance is comparable for all $N$ on a linear scale. Compared on a logarithmic scale (Figure 10(b)), it is clear that as $N$ increases, the model results in better performance, as indicated by the lower $\mathcal{W}^{2,2}(p_d, p_g)$ scores. From figure 10(c) we infer that, on multimodal data, choosing insufficient number of centers could lead to *mode hopping*, where $p_g$ latches on to different modes of the data as iterations progress (observable as sharp spikes in $\mathcal{W}^{2,2}$ for the case of $N = 5$). This is attributed to the fact that the number of centers drawn is insufficient to represent the underlying modes in the target data. On the other hand, for larger values, such as $N \geq 500$, the additional computational overhead slows down training. We found that $N = 100$ is an acceptable compromise. Figure 11 shows the samples from $p_d$ alongside those drawn from $p_g$ as the iterations progress. The contour plot shows the level-sets of the optimal

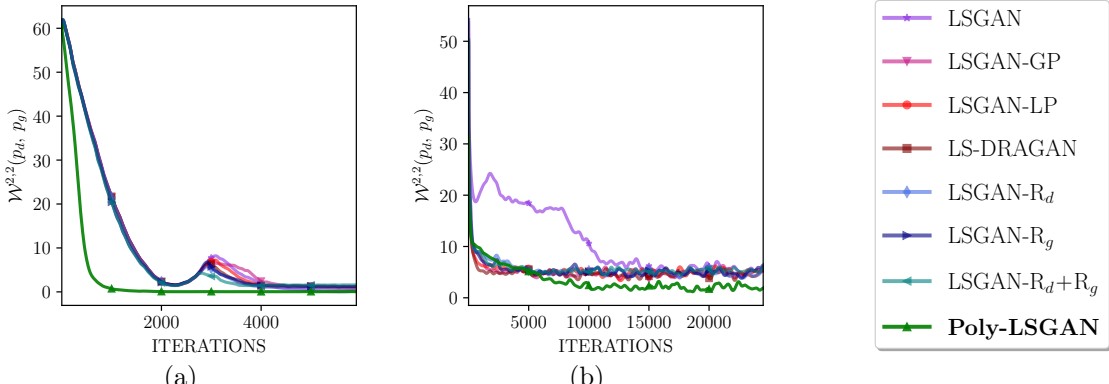

Figure 7: The Wasserstein-2 distance versus iterations on learning (a) a 2-D Gaussian; and (b) a 2-D Gaussian mixture, for various LSGAN variants. The performance of the Poly-LSGAN with the RBF discriminator is superior to the baselines in both scenarios. The convergence is also relatively smoother and stabler, unlike the baselines, which have fluctuations on the 2-D Gaussian learning task.

discriminator $D^*(\boldsymbol{x})$ (blue: low; yellow: high). When $N$ is small, some of the modes in $p_d$ are missed out by the discriminator, thus destabilizing training, whereas for large $N$, all the modes are captured accurately.

Similar convergence plots, juxtaposed with the discriminator level-sets (in WGAN based variants), for the two experiments conducted in Section 6.1 of the *Main Manuscript* are shown in Figures 13 and 12. We observe that in both the unimodal and multimodal cases, Poly-WGAN converges faster than the baselines, and the *one-shot* optimal discriminator learns a better representation of the underlying distributions than the baseline GANs and GMMNs. Poly-WGAN also outperforms the non-adversarial GMMN variants, and there is no mode-collapse upon convergence.

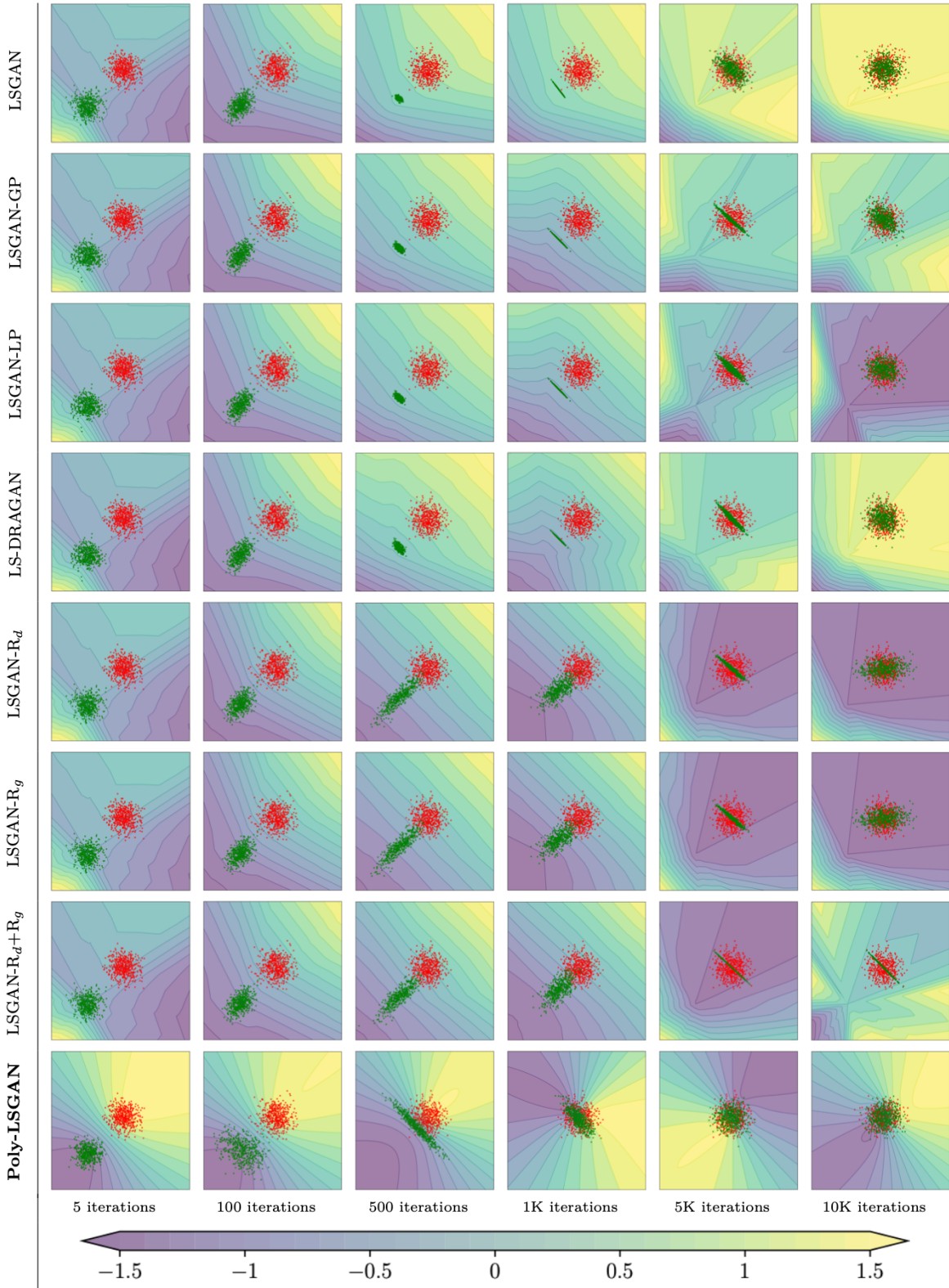

Figure 8: Convergence of generator distribution (*green*) to the target 2-D Gaussian data (*red*) on the considered LSGAN variants. The heatmap represents the values taken by the discriminator. The Poly-LSGAN approach leads to a better representation of the discriminator function during the initial training iterations when compared to baseline approaches, leading to a faster convergence. Poly-LSGAN also does not experience *mode collapse*.

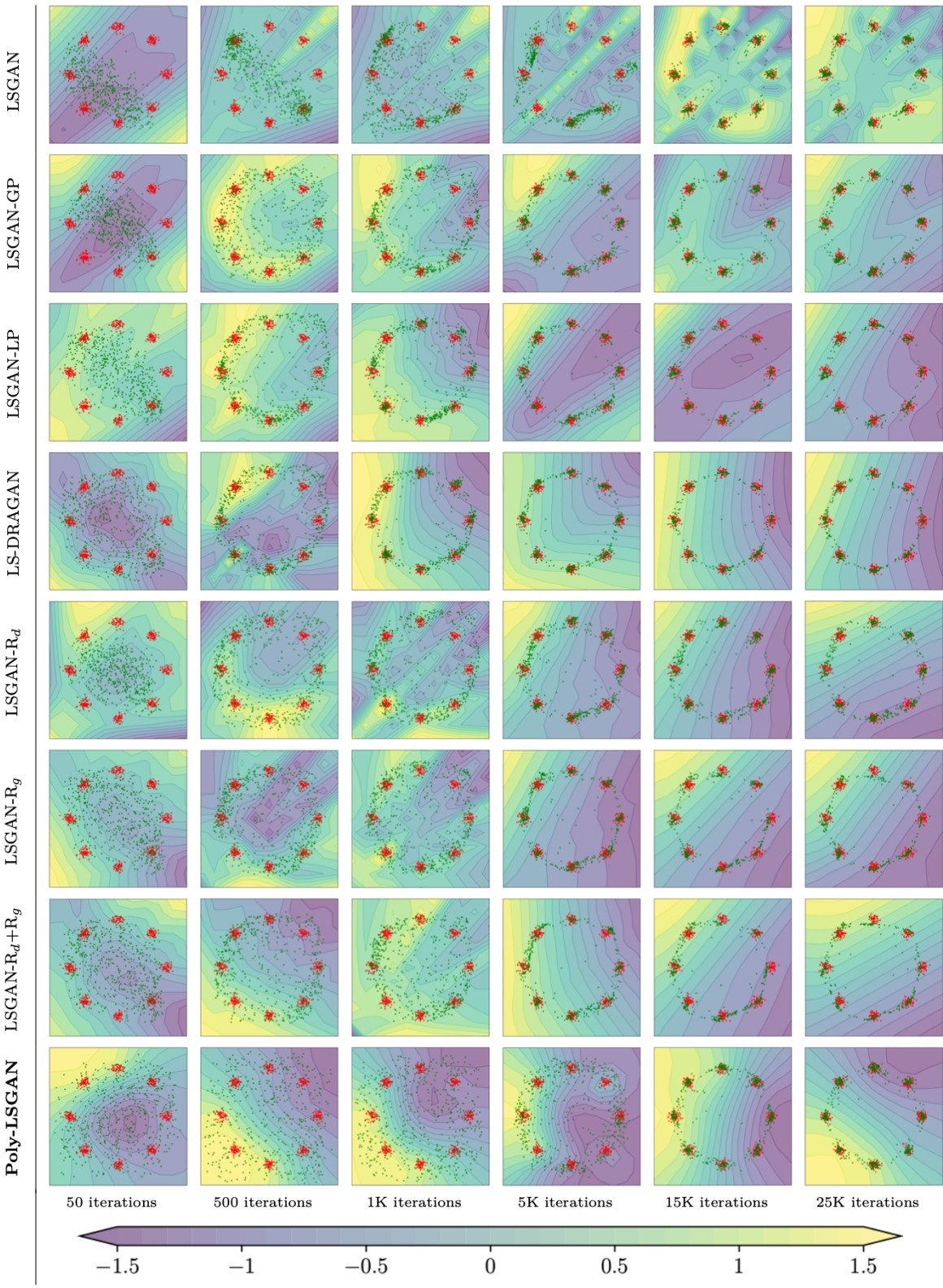

Figure 9: Convergence of generator distribution (*green*) to the target multimodal Gaussian data (*red*) on the considered LSGAN variants, superimposed on the level-sets of the discriminator. The ideal $D(\boldsymbol{x})$ assigned a value of $b = 1$ to reals and $a = -1$ to fakes. Poly-LSGAN is able to identify the modes of the GMM more accurately than the baselines.

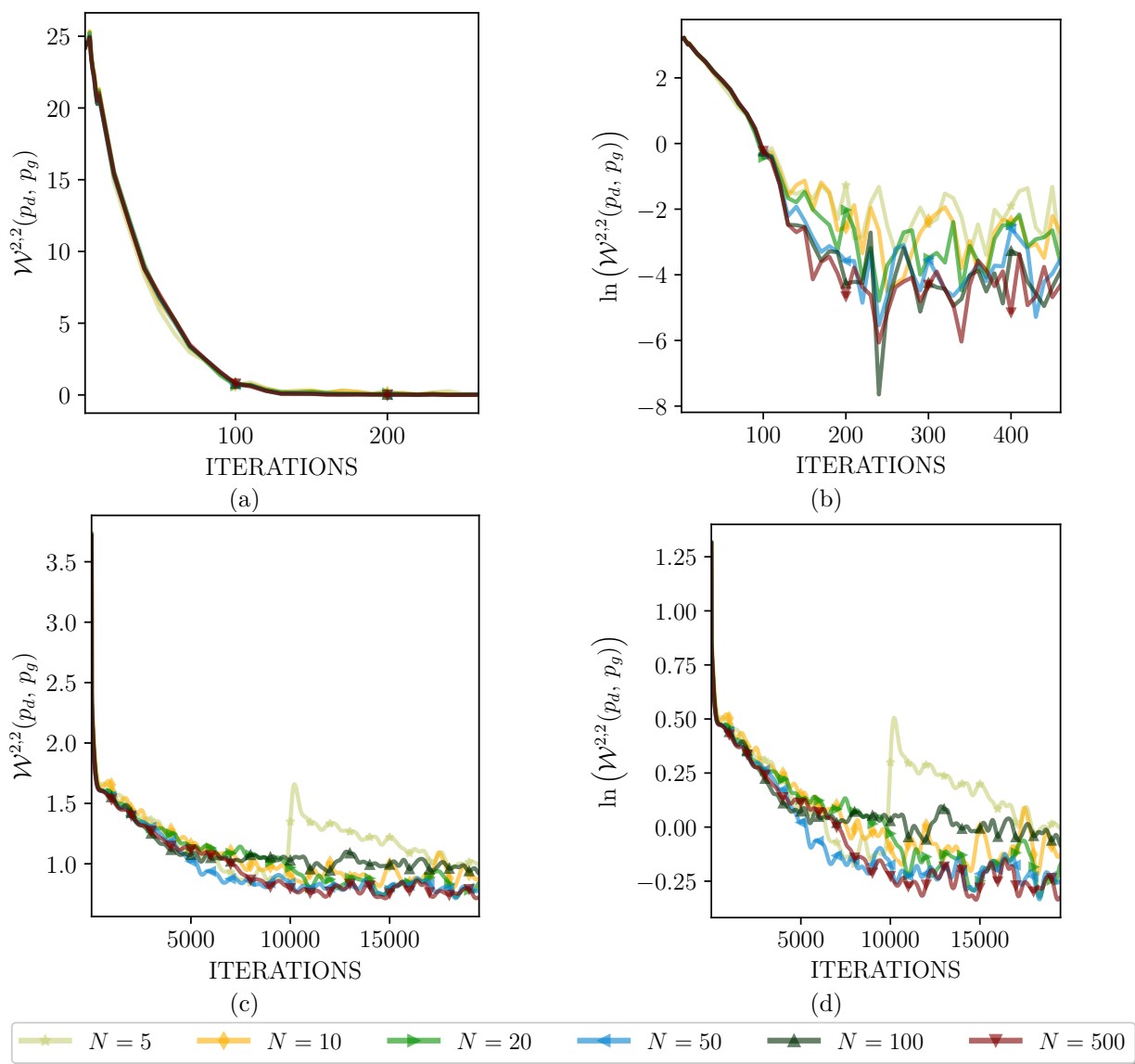

Figure 10: Training Poly-WGAN on 2-D Gaussian data: Plots comparing (a) the Wasserstein-2 distance between $p_d$ and $p_g$ ($\mathcal{W}^{2,2}(p_d, p_g)$); and (b) the natural logarithm of $\mathcal{W}^{2,2}(p_d, p_g)$ for various number of centers $N$ in the RBF network. The generator converges to a lower $\mathcal{W}^{2,2}(p_d, p_g)$ as $N$ increases. Convergence plots on training Poly-WGAN on learning 2-D Gaussian mixture data comparing (c) the Wassersting-2 distance $\mathcal{W}^{2,2}(p_d, p_g)$; and (d) the natural logarithm of $\mathcal{W}^{2,2}(p_d, p_g)$, for various choices of $N$. For small $N$, the discriminator is unable to capture all the modes in the data, resulting in training instability. Choosing large $N$ increases computational load. Setting $N = 100$ is a viable compromise.

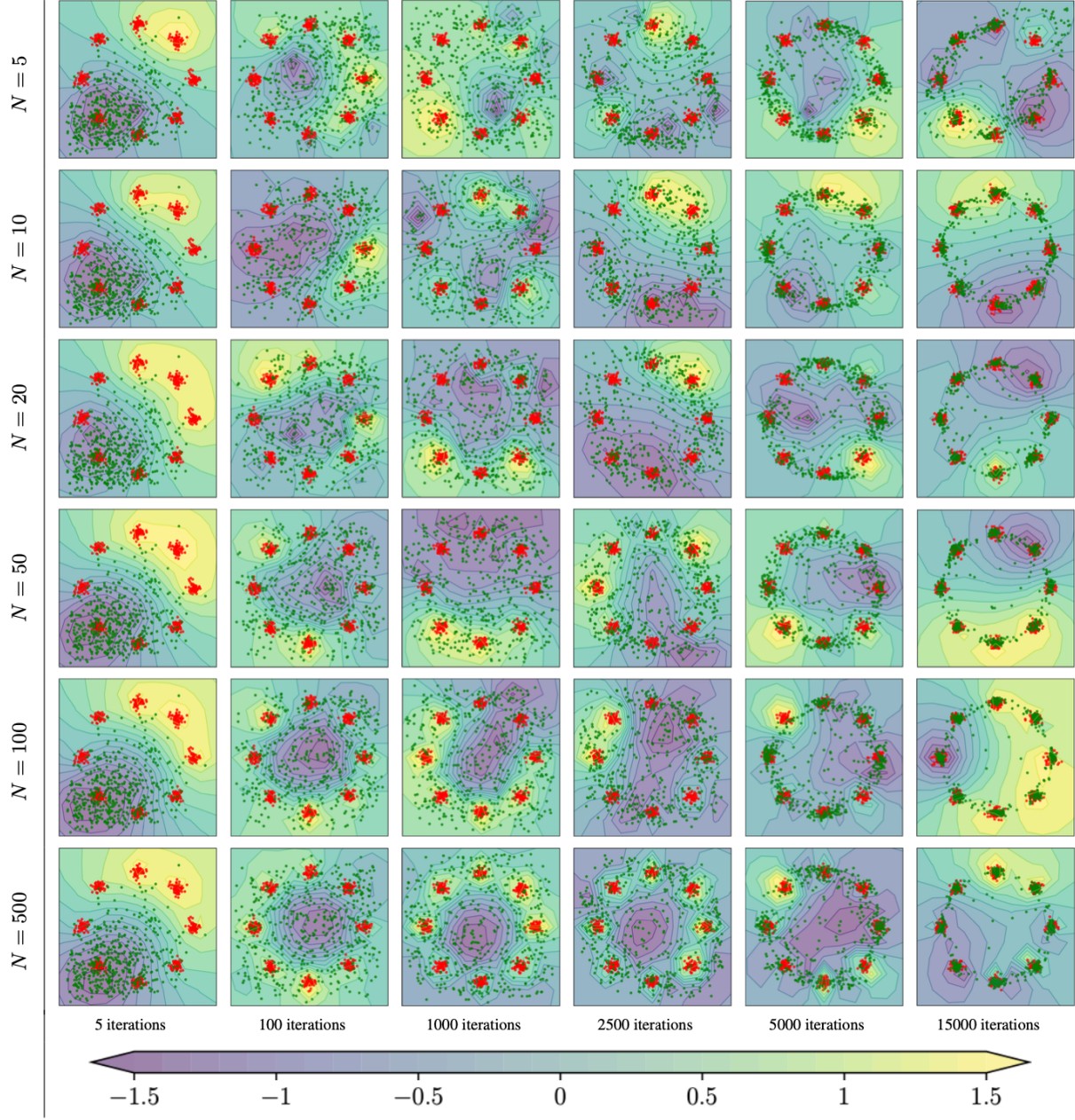

Figure 11: Illustration of convergence of the generator distribution (green dots) to the target multimodal Gaussian data (red dots) with Poly-WGAN as iterations progress, as a function of the number of RBF centers $N$. The contours are the level-sets of the discriminator. For small $N$, *mode coverage* is not adequate, whereas for large $N$, the computational overhead is large. A moderate value of $N = 100$ was found to be a workable compromise.

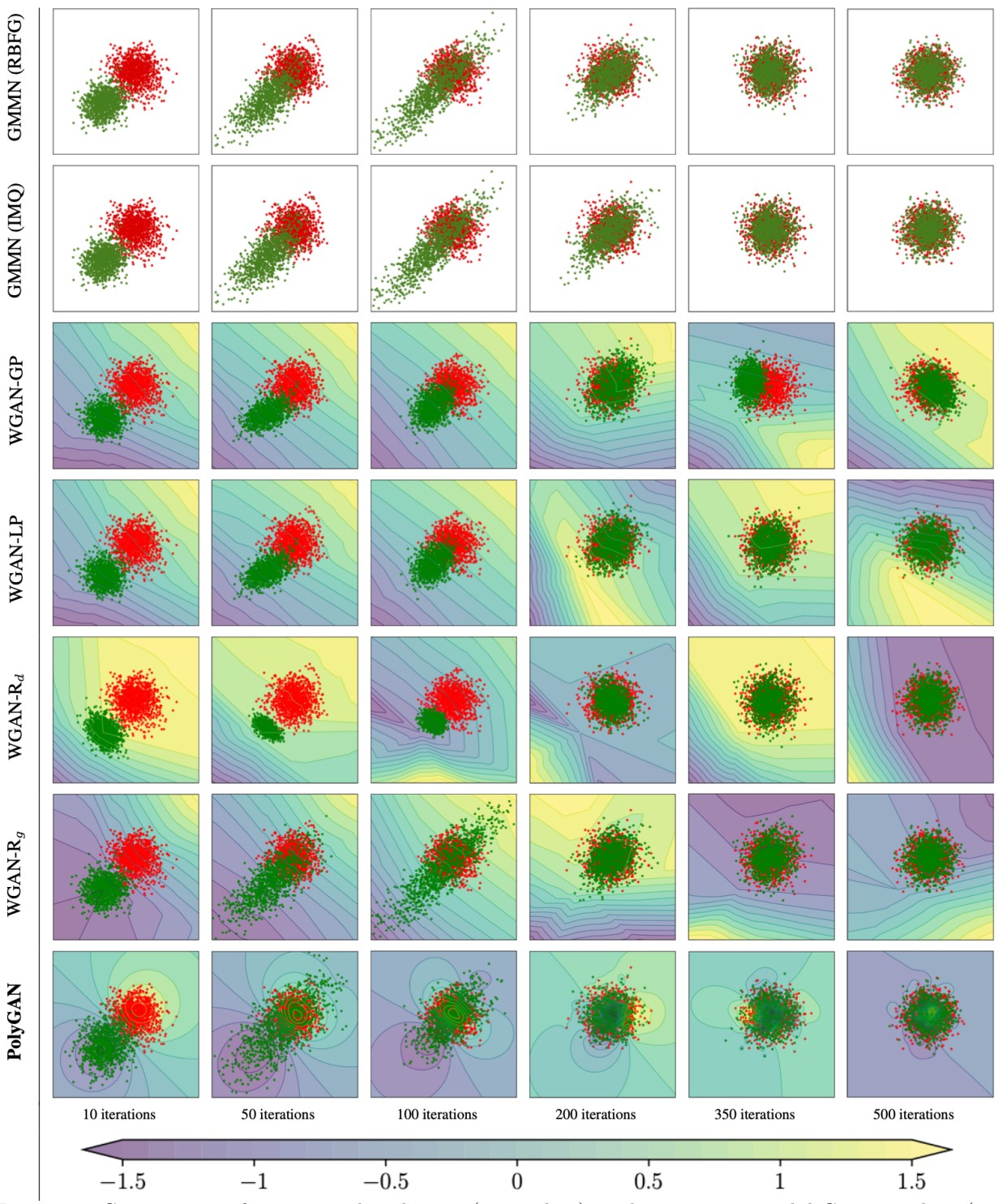

Figure 12: Convergence of generator distribution (green dots) to the target unimodal Gaussian data (red dots) on the considered WGAN and GMMN variants. The contours represent the discriminator level-sets in the GAN variants. Poly-WGAN converges significantly faster than the baseline variants.

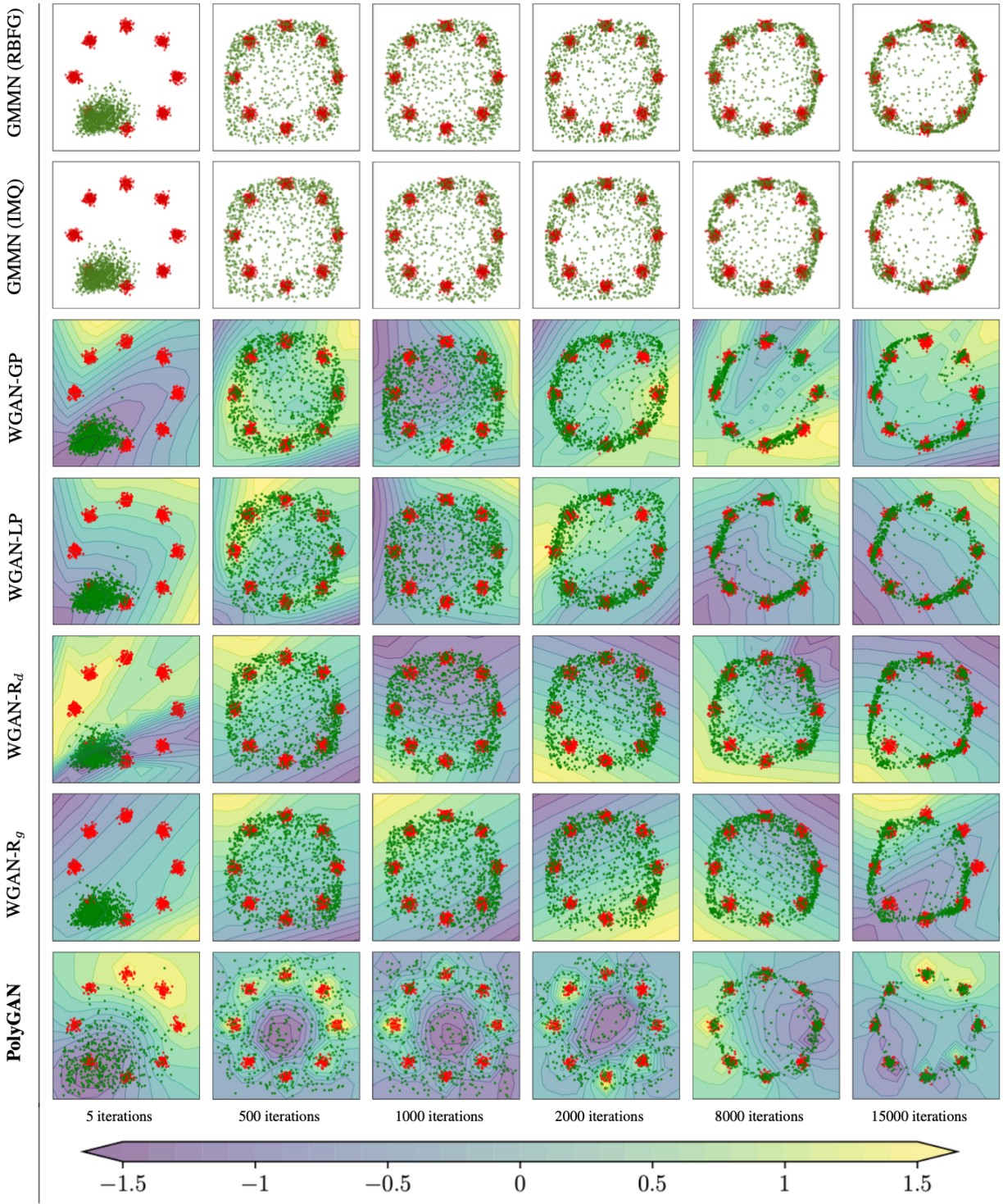

Figure 13: Convergence of generator distribution (green dots) to the target multimodal Gaussian (red dots) on the GMMN and WGAN variants. The contours represent discriminator level-sets. Poly-WGAN learns a better representation of $p_d$ leading to a faster convergence.

## F    Additional Experiments on Images

In this section, we provide additional experimental results on both image-space and latent-space matching approaches. On Poly-LSGAN, we discuss image-space generation, while of Poly-WGAN we present results for both image-space and latent-space generation tasks.

### F.1    Image-space matching with Poly-LSGAN

We train Poly-LSGAN on the MNIST, Fashion-MNIST and CelebA datasets, employing the 4-layer DC-GAN (Radford et al., 2016) generator architecture. As discussed in Section 3.1, we set $k = 2$, but restrict the solution to only include about $3^{rd}$ or $4^{th}$ order polynomials allows for training. Figure 14 depicts the images generated by Poly-LSGAN. In all scenarios, we observe that, although the generator is able to generate images resembling those from the target dataset, the visual quality of the images is sub-par compared to standard GAN approaches. Additional training of these models resulted in gradient explosion caused by the singularity of the matrix $\mathbf{B}$ as the iterations progress.

### F.2    Image-space matching with Poly-WGAN

We compare the performance of Poly-WGAN and baseline GMMN with the IMQ kernel. The generator is a 4-layer DCGAN (Radford et al., 2016). The kernel estimate as well as the polyharmonic RBF discriminator operate on the 784-dimensional data. For Poly-WGAN, we consider $m = \frac{n}{2} + 2 = 394$. The generator learning rate is set to $\eta_g = 0.01$ for both models considered. Figure 15 shows that the images generated by Poly-WGAN are comparable to those generated by GMMN-IMQ. Quantitatively, Poly-WGAN achieved an FID of 81.341, while GMMN-IMQ achieved an FID of 98.109 after 50,000 iterations.

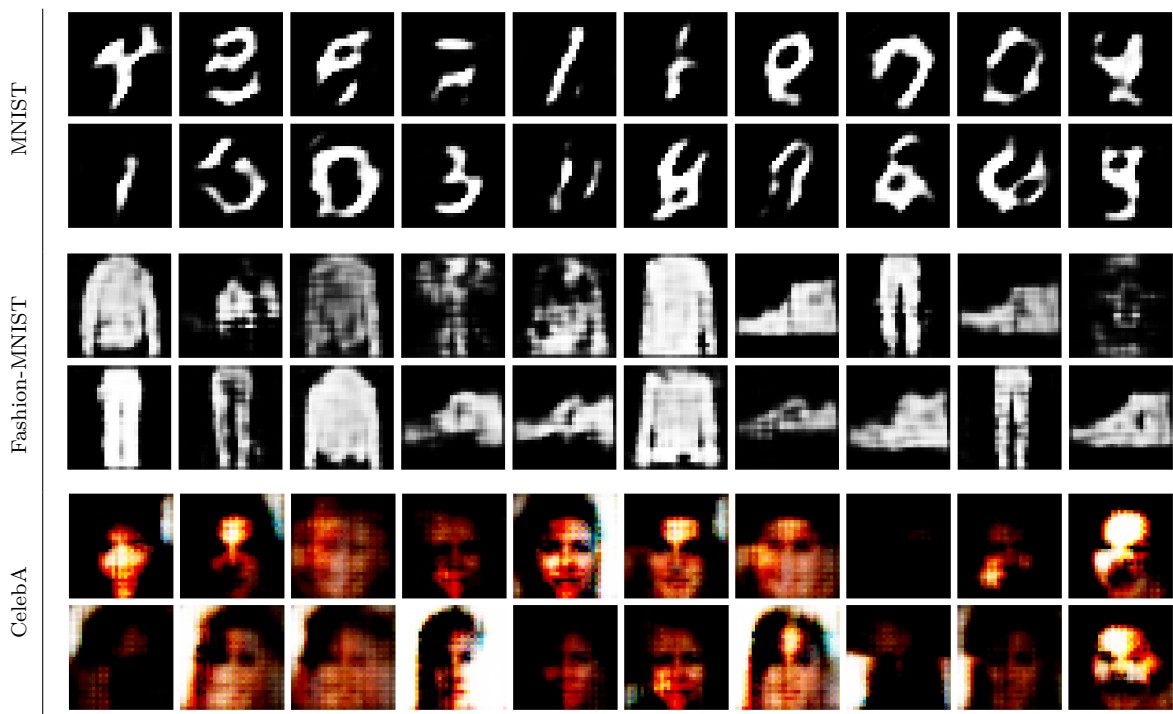

Figure 14: Images generated by training Poly-LSGAN on vectorized images drawn from (a) MNIST; (b) Fashion-MNIST; and (c) CelebA datasets. While Poly-LSGAN learns meaningful representations (although visually sub-par compared to standard GANs) on MNIST and Fashion-MNIST, the generator fails to converge in all scenarios.  *The poor performance of Poly-LSGAN can be attributed to training instability issues caused by the singularity of the matrix $\mathbf{B}$ in solving for the optimal discriminator weights.*

### F.3 Latent-space matching with Poly-WGAN

Motivated by the training paradigm in latent diffusion models (Rombach et al., 2022), as an intermediary between Poly-WGAN and PolyGAN-WAE, we consider training Poly-WGAN to learn the latent-space distribution of various datasets. We train convolutional autoencoders with 16- and 63-dimensional latent-space on MNIST and CelebA datasets, respectively. The various WGAN baselines and Poly-WGAN are trained to map a 100-dimensional noise distribution to the latent space of the target data. We also compare against a trainable version of Poly-WGAN (called Poly-WGAN(T)), which employs a single-layer RBF network with $10^3$ nodes, but whose centers and weights are learned via stochastic gradient-descent with the un-regularized WGAN loss. The gradient-penalty order is set to $m = \lceil \frac{n}{2} \rceil$ in both Poly-WGAN variants. The choice of the activation implicitly enforces the gradient penalty in Poly-WGAN(T). Motivated by the experimental results reported in Appendix 6.3, in all variants with a trainable discriminator, we update the discriminator five times per generator update. The models are evaluated in terms of the FID and *compute time*, which is the time elapsed between two generator updates. Due to the inclusion of an autoencoder in the formulation, we also compare the models in terms of their relative FID (rFID) (Rombach et al., 2022), where the reference images for FID computation are obtained by passing the dataset images through the pre-trained autoencoder.

Figure 16 presents the convergence of rFID as a function of iterations, while Table 9 presents the converged FID and rFID scores, and the compute times of Poly-WGAN and the baselines. From Figure 16, we observe that, on the 16-D learning task, the convergence behavior of Poly-WGAN is on par with the baselines, while on the 63-D data, Poly-WGAN converges significantly faster. The converged FID and rFID scores achieved by Poly-WGAN are superior to the baseline variants. On the 63-D learning task, Poly-WGAN is nearly an order of magnitude faster than the baselines in terms of compute time. Poly-WGAN(T) performs sub-optimally on both tasks, indicating that the choice of the centers and the weights indeed plays a crucial role in the performance of Poly-WGAN.

### F.4 Latent-space matching with PolyGAN-WAE

The base Poly-WGAN algorithm can be used to learn image-space distributions. Similar to GMMNs, PolyGANs also suffer from the *curse of dimensionality*. Although this can be alleviated to a certain extent by employing a generator that learns the latent-space of datasets, or PolyGAN-WAE that employ an autoencoding generator, these models are limited by the representation capability of the latent space, and the autoencoder performance. In order to explore this limitations of the WAE based approaches, we trained PolyGAN-WAE on high-resolution (192 × 192) images where encoder and decoder networks use the unconditional BigGAN (Brock et al., 2018) architecture. Figure 17 presents the images generated by PolyGAN-WAE in this scenarios. The converged model achieves an FID of 32.5. We observe that, while the images generated are not competitive in comparison to the high-resolution compute-heavy GAN variants such as StyleGAN, the generated images are superior to BigWAE-MMD and BigWAE-GAN variants with similar network complexities (having FID scores of 37 and 35, respectively) (Tolstikhin et al., 2018).

We include results of additional experiments conducted on PolyGAN-WAE and the baselines. Figures 19–22 show the samples generated by the various WAE models. The WAE-GAN and WAAE-LP models that incorporate a trainable discriminator are slower to train than the models that employ a closed for discriminator. We observe that PolyGAN-WAE generates perceptibly sharper images on MNIST. PolyGAN-WAE generates visually more diverse images than the baselines on CelebA and LSUN-Churches datasets. WAE-MMD (RBFG) and SWAE suffered from mode collapse on CIFAR-10 and LSUN-Churches, respectively.

**Latent-space continuity**: A visual assessment of latent space continuity is carried out by interpolating the latent vectors for two real images and decoding the interpolated vectors. Representative images are presented in Figures 27 to 30. Interpolated images from PolyGAN-WAE are comparable to those generated by CWAE and WAE-MMD (IMQ) on CIFAR-10 and LSUN-Churches, respectively, while they are sharper than the baselines on MNIST and CelebA datasets.

**Latent-space alignment**: As the various GAN flavors are employed in transforming the latent-space distribution of the generator to a standard normal distribution, we compare their performance in terms of their

latent-space alignment. Table 10 presents the Wasserstein-2 distance between the latent-space distribution of the encoder/generator network, and the target Gaussian $\mathcal{W}^{2,2}(p_{d_\ell}, p_z)$, while Figure 18 presents $\mathcal{W}^{2,2}(p_{d_\ell}, p_z)$ as a function of the training iterations. Across all datasets, we observe that PolyGAN-WAE attains the lowest Wasserstein-2 distance, indicating close alignment between the latent-space distributions.

**Image reconstruction**: Figures 23-26 show the images reconstructed by PolyGAN-WAE and the WAE variants. The images reconstructed by PolyGAN-WAE are sharper and closer to the ground-truth images. These are also in agreement with the qualitative results presented in Table 11. Figure 33 plots reconstruction error as a function of iterations for the various models considered. In order to have a fair comparison, we do not consider WAE-GAN and WAAE-LP in these comparisons, as the learning rates considered for the models are lower by an order. We observe that PolyGAN-WAE is on par with the baselines when trained on low-dimensional latent data (as in the case of MNIST and CIFAR-10), but outperforms the baselines, saturating to lower values in the case of CelebA and LSUN-Churches.

**Image sharpness**: Table 11 shows the image sharpness metric computed on both random and interpolated images. PolyGAN-WAE outperforms the baselines on the *random sharpness* metric, while achieving competitive scores on *interpolation sharpness*. These results indicate that, while the baseline WAE-MMDs have learnt accurate autoencoders, the latent space distribution has failed to match the prior distribution, resulting in lower scores when computing the sharpness metric on samples decoded from the prior. The closed-form optimal discriminator used in PolyGAN-WAE alleviates this issue.

**Inception Distances**: We plot FID and KID as a function of iterations in Figures 31 and 32, respectively. In both cases, we observe that PolyGAN-WAE saturates to the lowest (best) scores in comparison to the baselines. The improvements are more prominent on experiments involving higher-dimensional latent representation (for example, LSUN-Churches, using a 128-D latent space). Best case KID scores are presented in Table 11. The KID for PolyGAN-WAE is nearly 35% lower than that of the best-case baseline (CWAE) in the case of MNIST.

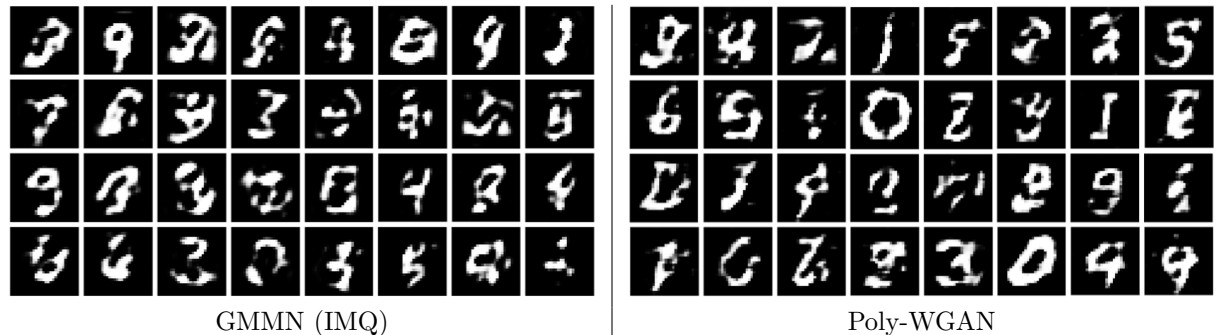

GMMN (IMQ)                     Poly-WGAN

Figure 15: Images generated by GMMN-IMQ and Poly-WGAN on the MNIST image-space matching task. While Poly-WGAN generates images marginally superior to GMMN (IMQ), both the results are inferior to the WAE and WGAN counterparts. The poor performance is a consequence of the *curse of dimensionality*, which is also the reason why we considered latent-space matching with PolyGAN-WAE.

Table 9: A comparison of WGAN flavors and Poly-WGAN when trained to learn the latent-space distribution of a pre-trained autoencoder network on MNIST and CelebA learning tasks. Poly-WGAN(T) is a trainable version of Poly-WGAN, where the weights are initialized based on Poly-WGAN, and subsequently learnt through back-propagation on the discriminator. The baseline GAN and Poly-WGAN(T) discriminators are updated five times per generator update. The performance is reported in terms of (i) The FID of the converged models; (ii) The relative FID *(rFID)* between the target samples and the output of the pre-trained autoenoder (AE); and (iii) The *Compute Time* between two generate updates. The FID of the benchmark pre-trained autoencoder is provided for reference. The rFID value is approximately the difference between the FID of the GAN samples, and that of the samples generated by the benchmark AE. Poly-WGAN achieves lower FID scores on both the MNIST and CelebA learning tasks, in a tenth of the compute time.

| WGAN flavor | MNIST (16-D) | | | CelebA (63-D) | | |
|---|---|---|---|---|---|---|
| | FID ↓ | rFID ↓ | Compute Time ↓ | FID ↓ | rFID ↓ | Compute Time ↓ |
| WGAN-GP | 19.441 | 6.363 | $0.132 \pm 0.003$ | 49.840 | 11.935 | $0.491 \pm 0.008$ |
| WGAN-LP | 17.825 | 5.657 | $0.144 \pm 0.008$ | 50.694 | 11.789 | $0.462 \pm 0.005$ |
| WGAN-$R_d$ | 17.948 | 6.780 | $0.119 \pm 0.007$ | 48.064 | 12.159 | $0.450 \pm 0.002$ |
| WGAN-$R_g$ | 18.498 | 6.330 | $0.127 \pm 0.006$ | 51.104 | 14.199 | $0.452 \pm 0.005$ |
| **Poly-WGAN(T)** | 17.445 | 5.277 | $0.150 \pm 0.003$ | 48.385 | 11.480 | $0.357 \pm 0.003$ |
| **Poly-WGAN** | **17.397** | **5.229** | $\mathbf{0.034 \pm 0.004}$ | **45.886** | **8.981** | $\mathbf{0.039 \pm 0.003}$ |
| Benchmark AE | 12.562 | 0 | – | 36.261 | 0 | – |

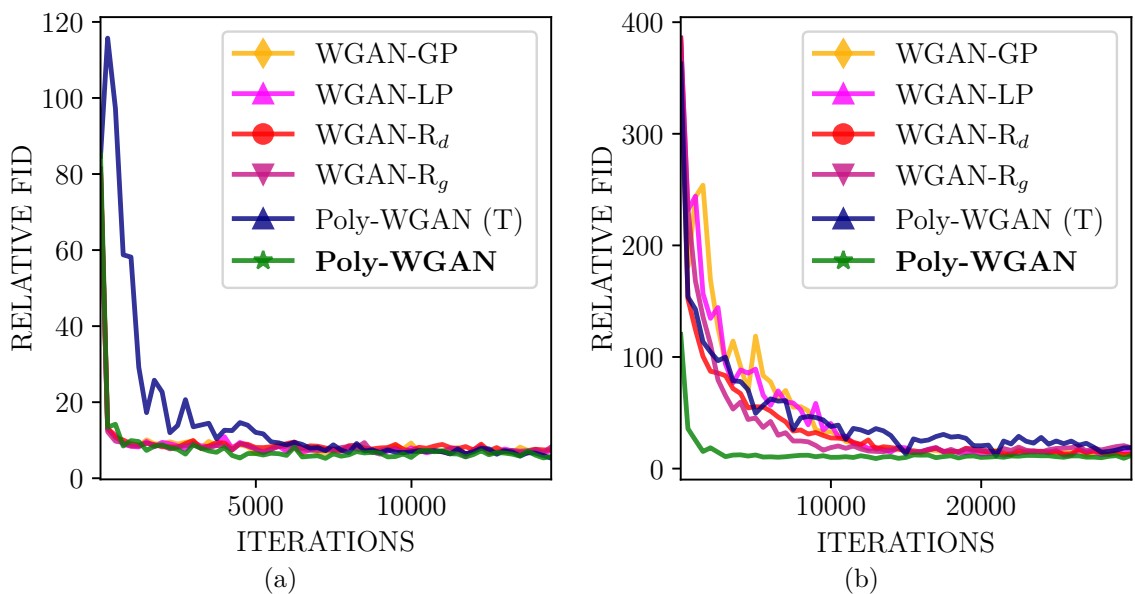

Figure 16: A comparison of the relative FID (rFID) of various WGAN and Poly-WGAN variants when trained on latent representations of (a) MNIST; and (b) CelebA datasets. The latent-space representations are drawn from a pre-trained deep convolutional autoencoder. The relative FID is computed between the *fakes* generated by decoding the generator outputs, and *reals* generated by decoding the latent representations of the dataset images. Poly-WGAN(T) is a trainable version of Poly-WGAN, where the discriminator RBF weights are learnt through back-propagation. We observe that Poly-WGAN performs on par with the baselines in learning low-dimensional latent representations, as in the case of MNIST, while converging faster (by an order) on higher-dimensional data (63-D on CelebA).

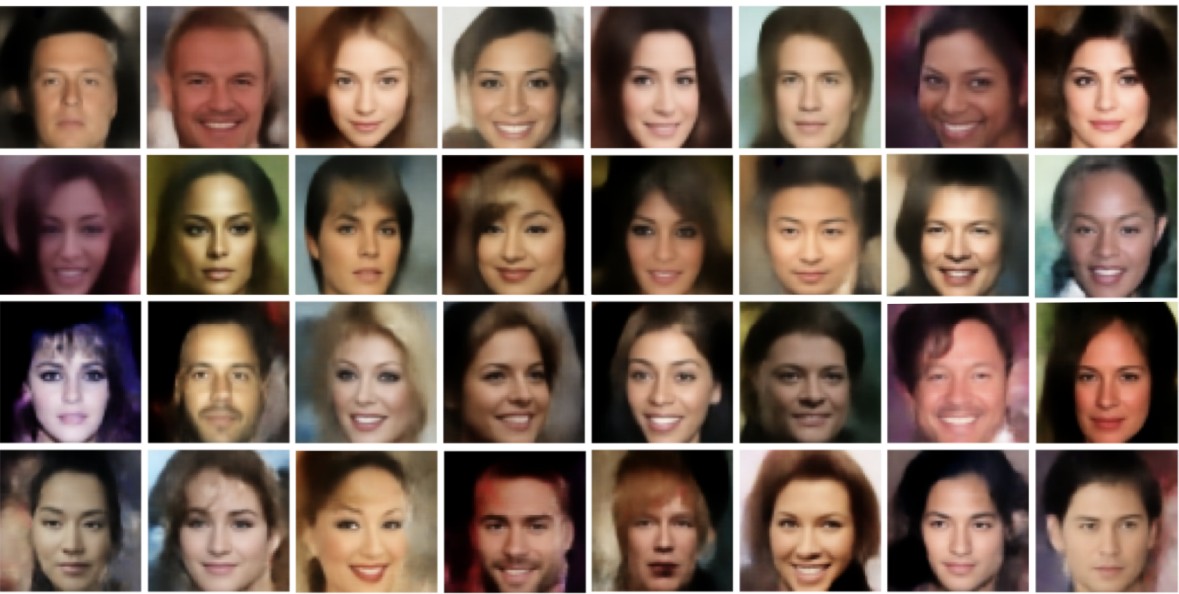

Figure 17: (🎨 Color online) High-resolution (192 × 192) CelebA images generated by PolyGAN-WAE.

Table 10: A comparison of the converged WAE models, including PolyGAN-WAE, in terms of the Wasserstein-2 distance between the latent-space distribution of the data, and the target noise distribution ($\mathcal{W}^{2,2}(p_{d_\ell}, p_z)$). WAE-GAN and WAAE-LP incorporate a trainable discriminator, while WAE-MMD variants, SWAE and CWAE compute closed-form kernel statistics between the latent-space distributions. WAEFR and PolyGAN-WAE employ a closed-form discriminator network with predetermined weights, to approximate a Fourier-series or RBF approximation, respectively. When learning relatively low-dimensional latent spaces (as in the case of MNIST), all models perform comparably, while PolyGAN-WAE is superior to the baselines in learning on higher-dimensional latent spaces. PolyGAN-WAE achieves the lowest $\mathcal{W}^{2,2}$ scores in all the four scenarios considered.

| WAE flavor | MNIST (16-D) | CIFAR-10 (64-D) | CelebA (128-D) | LSUN-Churches (128-D) |
|---|---|---|---|---|
| WAE-GAN | 1.9468 | 20.03773 | 12.6205 | 5.5128 |
| WAAE-LP | 1.8828 | 24.9344 | 23.6597 | 5.2301 |
| WAE-MMD (RBFG) | 0.8615 | 16.3907 | 27.3071 | 16.0910 |
| WAE-MMD (IMQ) | 1.1316 | 14.4645 | 5.4592 | 12.8840 |
| SWAE | 1.1441 | 18.6906 | 14.4378 | 53.4751 |
| CWAE | 0.5154 | 7.04151 | 6.2632 | 12.6781 |
| WAEFR | 0.6272 | 11.8180 | 6.0705 | 9.2847 |
| **PolyGAN-WAE** | **0.3388** | **6.1055** | **3.6195** | **5.0831** |

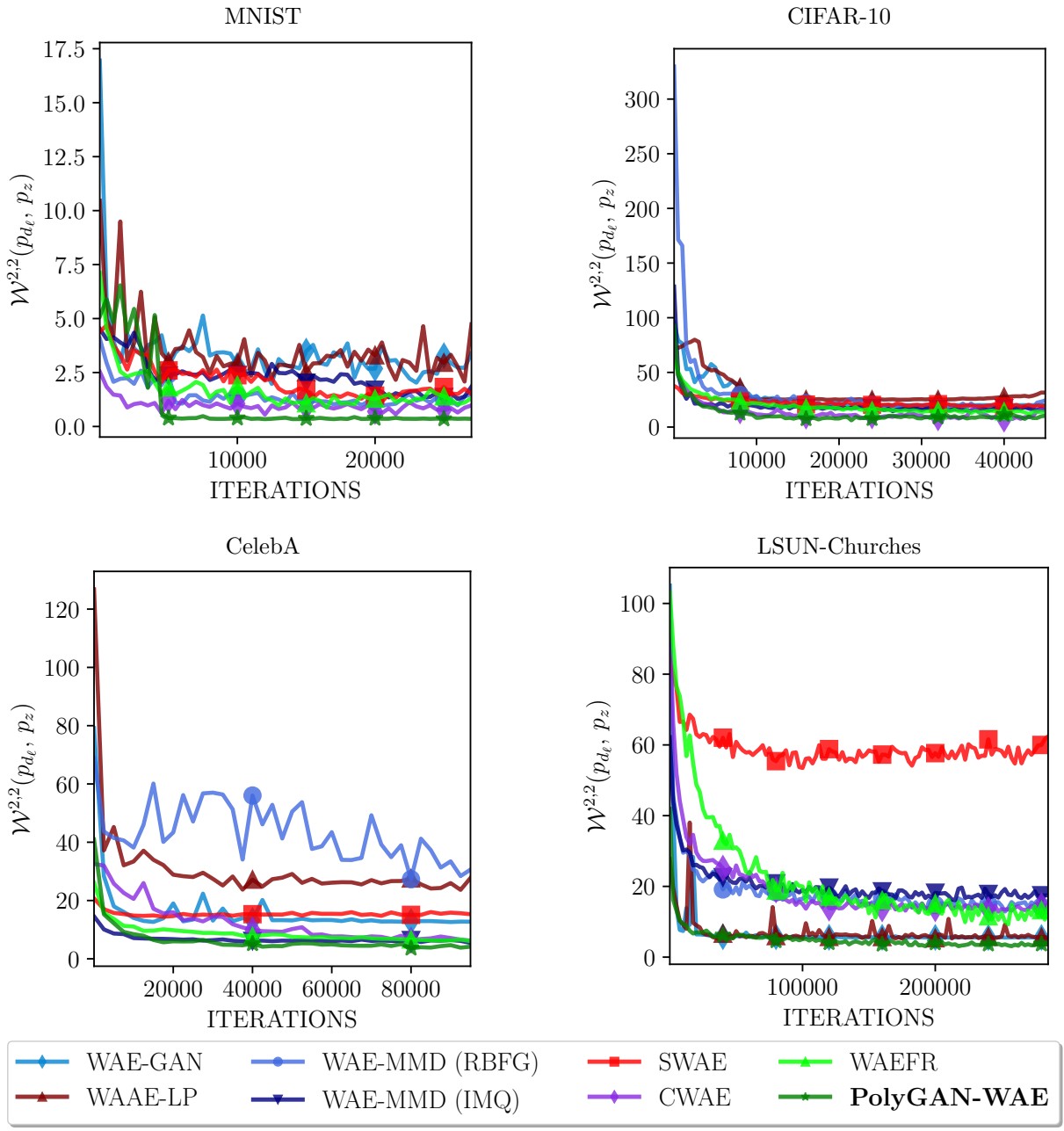

Figure 18: Wasserstein-2 distance between the latent-space distribution of the data, and the target noise distribution $(\mathcal{W}^{2,2}(p_{d_\ell}, p_z))$ versus iterations for the WAE flavors under consideration. PolyGAN-WAE converges to better (lower) $\mathcal{W}^{2,2}$ scores in all cases, indicating a superior match between the latent-space data distribution and the target Gaussian prior.

Table 11: A comparison of the WAE variants including PolyGAN-WAE in terms of kernel inception distance (KID), average reconstruction error $\langle RE \rangle$, and image sharpness. *Sharpness (Random Image)* corresponds to the sharpness computed on random samples drawn from the prior distribution, whereas *Sharpness (Interpolated Image)* is computed on the interpolated images. The benchmark sharpness is computed over images drawn from the target dataset. PolyGAN-WAE achieves the best (lowest) KID on all the datasets, while generating images with sharpness scores comparable to the baselines.

| | WAE flavor | MNIST | CIFAR-10 | CelebA | LSUN-Churches |
|---|---|---|---|---|---|
| **KID ↓** | WAE-GAN | 0.0221 | 0.1015 | 0.0423 | 0.1395 |
| | WAAE-LP | 0.0210 | 0.0832 | 0.0445 | 0.1398 |
| | WAE-MMD (RBFG) | 0.0533 | 0.1316 | 0.0623 | 0.1397 |
| | WAE-MMD (IMQ) | 0.0204 | 0.0908 | 0.0459 | 0.1379 |
| | SWAE | 0.0270 | 0.0929 | 0.0440 | 0.2129 |
| | CWAE | 0.0192 | 0.0794 | 0.0537 | 0.1858 |
| | WAEFR | 0.0206 | 0.0859 | 0.0416 | 0.1364 |
| | **PolyGAN-WAE** | **0.0120** | **0.0756** | **0.0366** | **0.1279** |
| **⟨RE⟩ ↓** | WAE-GAN | 0.0827 | 0.1250 | 0.0939 | 0.1450 |
| | WAAE-LP | 0.0747 | 0.1161 | 0.0776 | 0.1547 |
| | WAE-MMD (RBFG) | 0.1615 | 0.2246 | 0.1365 | 0.1408 |
| | WAE-MMD (IMQ) | 0.0584 | 0.1218 | 0.0920 | 0.1402 |
| | SWAE | 0.0574 | 0.1210 | 0.0885 | 0.1410 |
| | CWAE | 0.0768 | 0.1503 | 0.0982 | 0.1408 |
| | WAEFR | 0.0538 | **0.1185** | 0.0820 | 0.1387 |
| | **PolyGAN-WAE** | **0.0525** | 0.1190 | **0.0676** | **0.1365** |
| **Sharpness — Random Image** | WAE-GAN | 0.1567 | 0.0011 | 0.0015 | 0.0077 |
| | WAAE-LP | 0.1520 | 0.0029 | 0.0044 | 0.0082 |
| | WAE-MMD (RBFG) | 0.2231 | 0.0030 | 0.0034 | 0.0076 |
| | WAE-MMD (IMQ) | 0.1709 | 0.0100 | 0.0049 | 0.0091 |
| | SWAE | 0.1660 | 0.0136 | 0.0048 | 0.0087 |
| | CWAE | 0.2206 | 0.0035 | 0.0038 | 0.0068 |
| | WAEFR | 0.1717 | 0.0171 | **0.0066** | 0.0103 |
| | **PolyGAN-WAE** | **0.1776** | **0.0174** | 0.0052 | **0.0149** |
| **Sharpness — Interpolated Image** | WAE-GAN | 0.1681 | 0.0027 | 0.0032 | 0.0122 |
| | WAAE-LP | 0.1706 | 0.0041 | 0.0045 | 0.0125 |
| | WAE-MMD (RBFG) | 0.2251 | 0.0015 | 0.0044 | 0.0120 |
| | WAE-MMD (IMQ) | 0.1416 | **0.0071** | 0.0043 | 0.0124 |
| | SWAE | 0.1292 | 0.0059 | 0.0044 | 0.0130 |
| | CWAE | **0.2073** | 0.0019 | 0.0034 | 0.0107 |
| | WAEFR | 0.1396 | 0.0064 | 0.0065 | 0.0113 |
| | **PolyGAN-WAE** | 0.1496 | 0.0069 | **0.0067** | **0.0134** |
| | Benchmark | 0.1885 | 0.0358 | 0.0338 | 0.1029 |

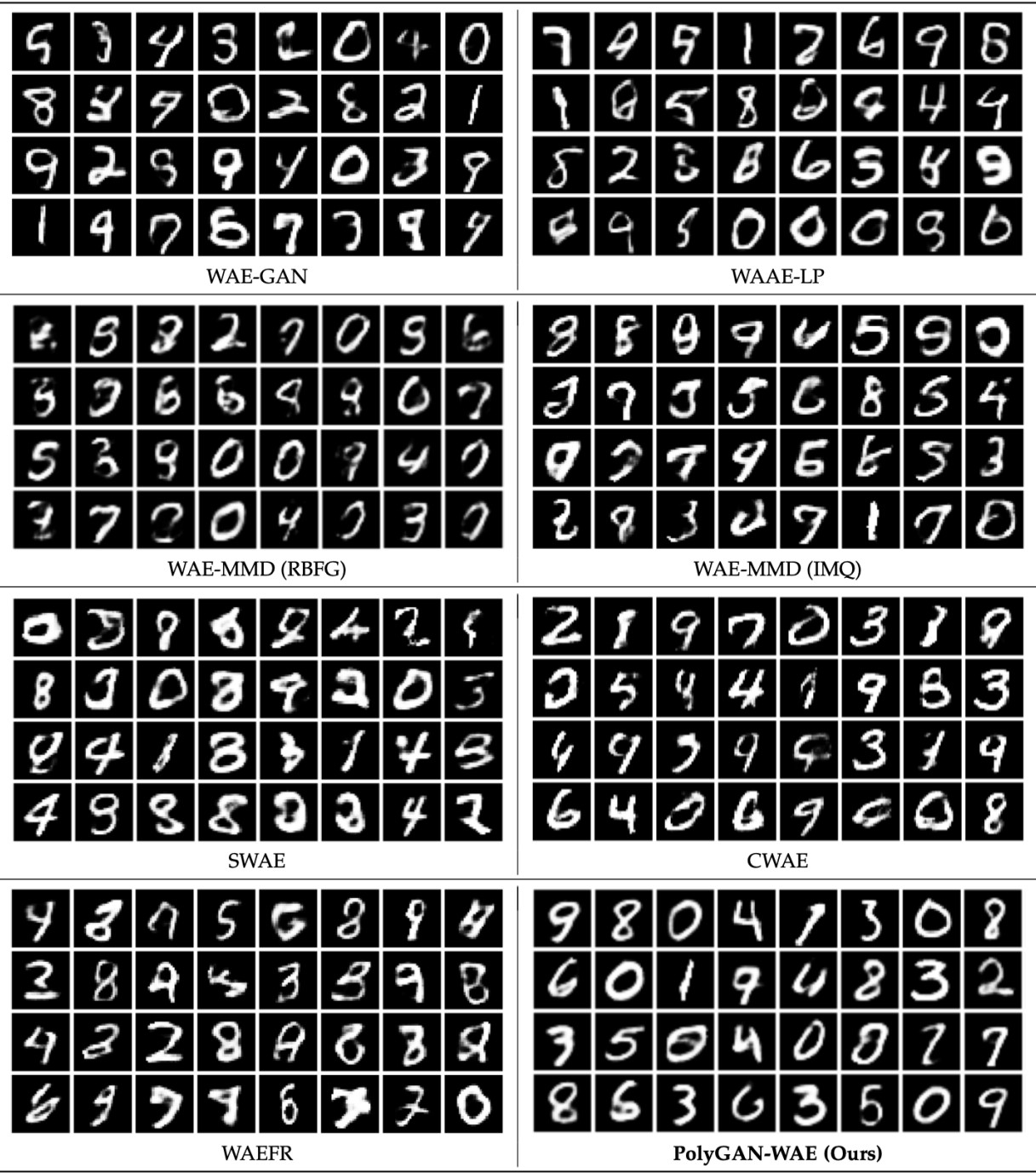

Figure 19: Images generated by decoding samples drawn from the target prior distribution on MNIST. PolyGAN-WAE generated images of superior quality than the baselines.

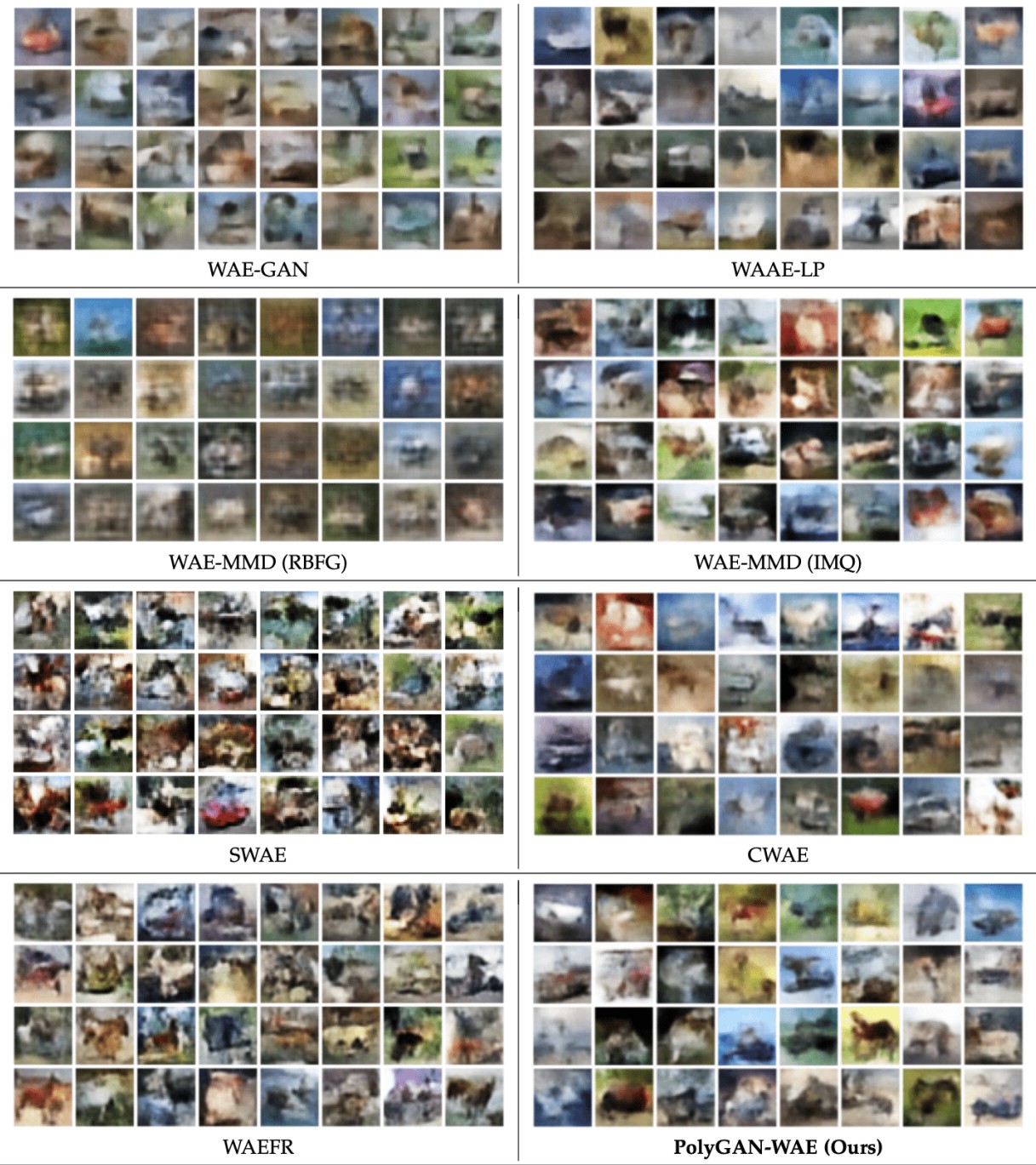

Figure 20: (🎨 Color online) Images generated by decoding samples drawn from the target prior distribution on the CIFAR-10 dataset. WAE-GAN, WAE-MMD (RBFG) and SWAE did not converge on CIFAR-10. While WAE-MMD (IMQ), CWAE and PolyGAN-WAE are comparable, the images generated have little visual similarity with those of the target dataset.

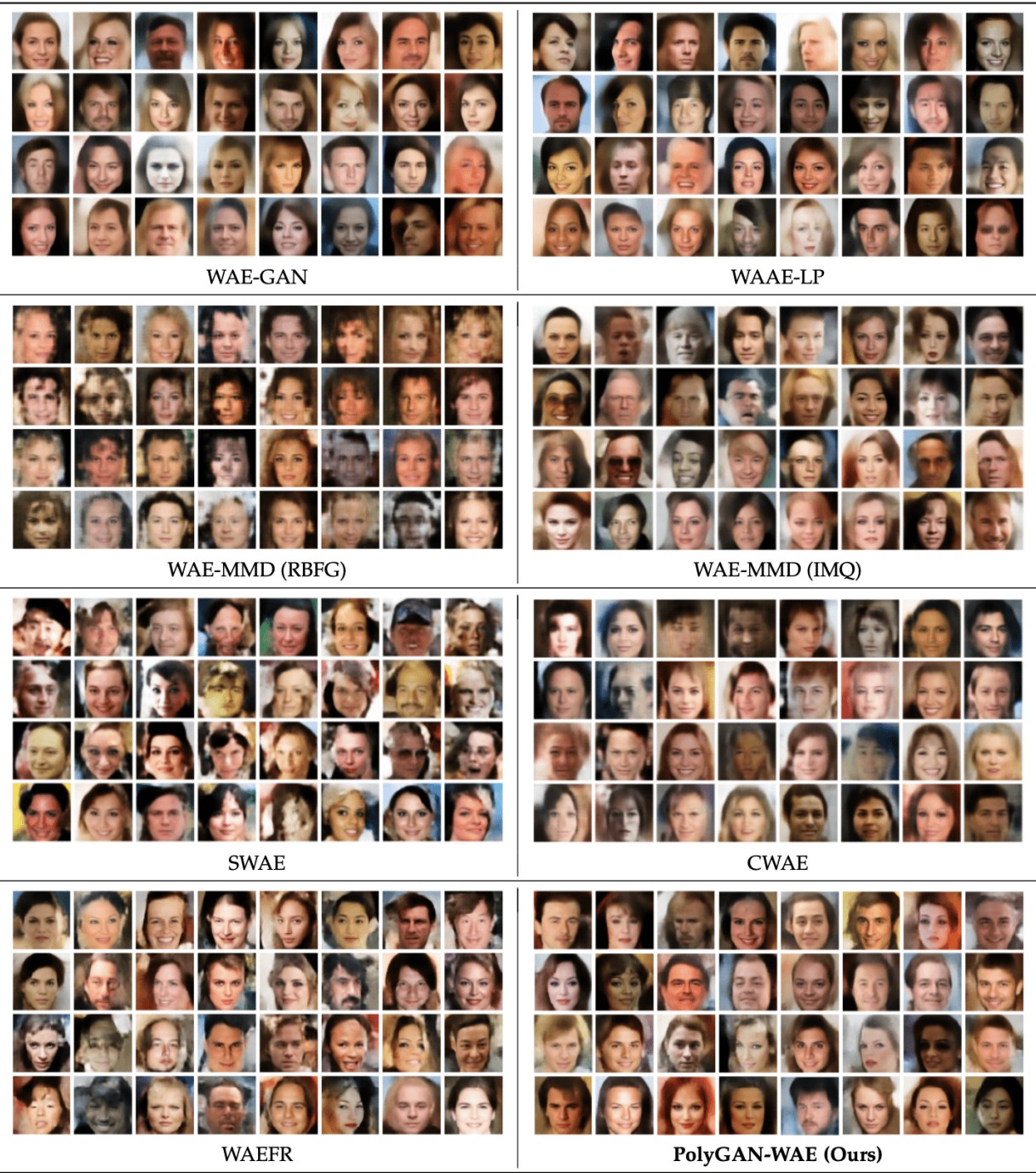

Figure 21: (🎨 Color online) Images generated by the WAE variants on decoding samples drawn from the prior distribution when trained on the CelebA dataset. Images generated by PolyGAN-WAE on CelebA are more diverse (in terms of face and background color, facial expression, etc.) compared with the baselines.

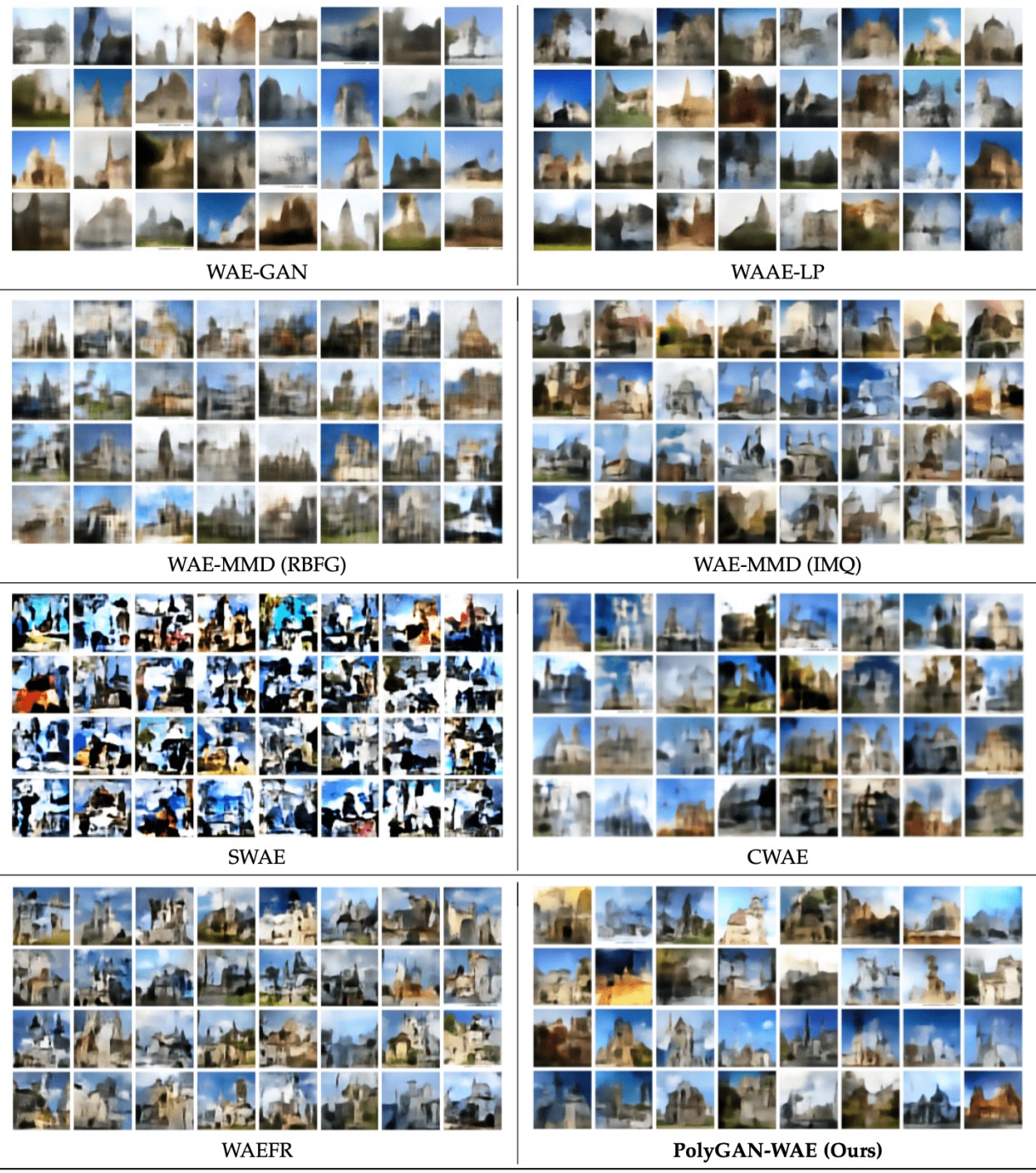

Figure 22: (🎨 Color online) Images generated by the WAE variants on decoding samples drawn from the prior distribution. PolyGAN-WAE is on par with CWAE and WAE-MMD variants on LSUN-Churches. SWAE failed to converge, while WAE-GAN and WAAE-LP resulted in smoother images, as opposed to the other WAE variants.

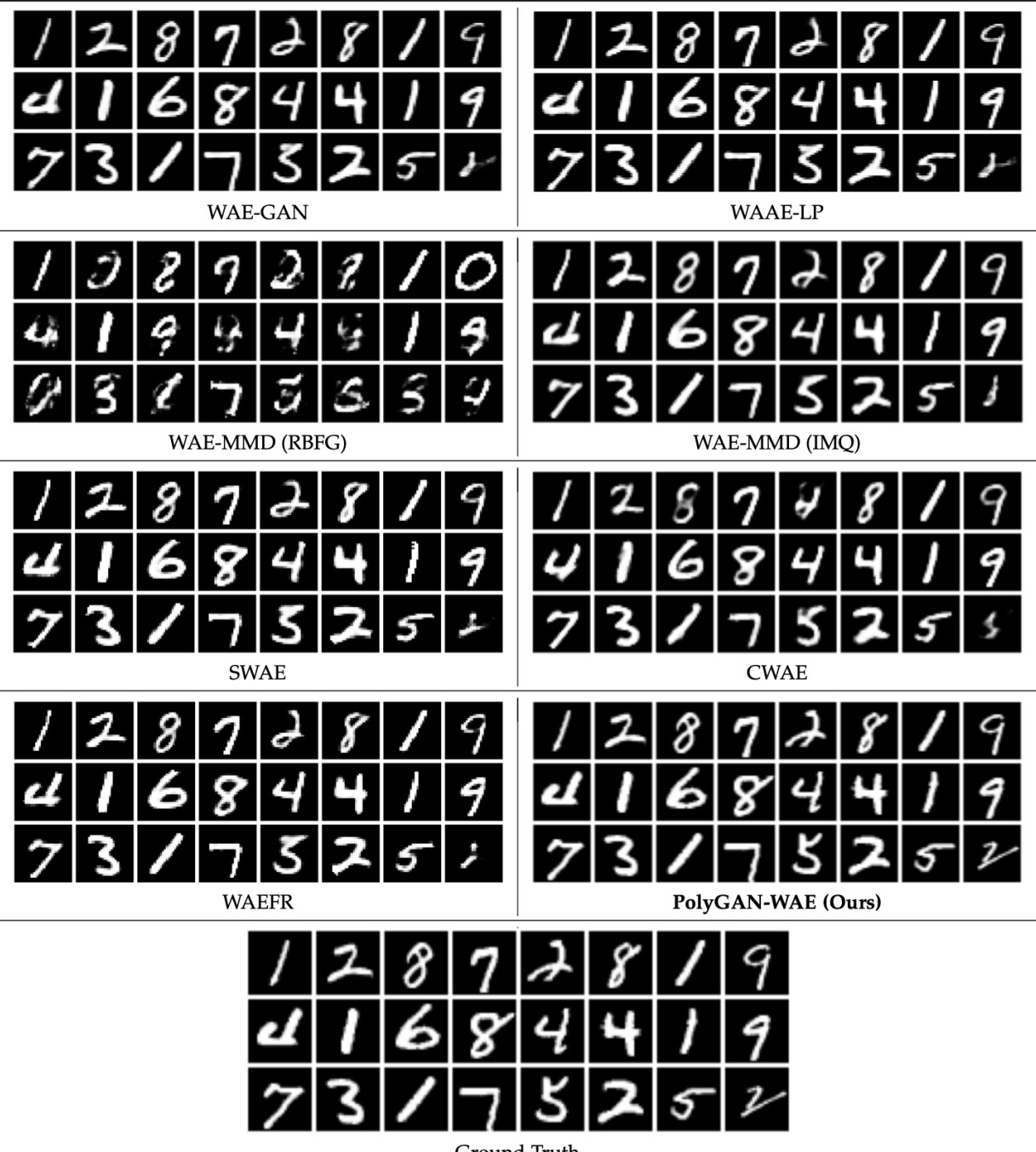

Figure 23: A comparison of the image reconstruction performance on MNIST dataset. The WAE-MMD (RBFG) baseline failed to reconstruct meaningful samples. PolyGAN-WAE generates the most accurate reconstructions on MNIST.

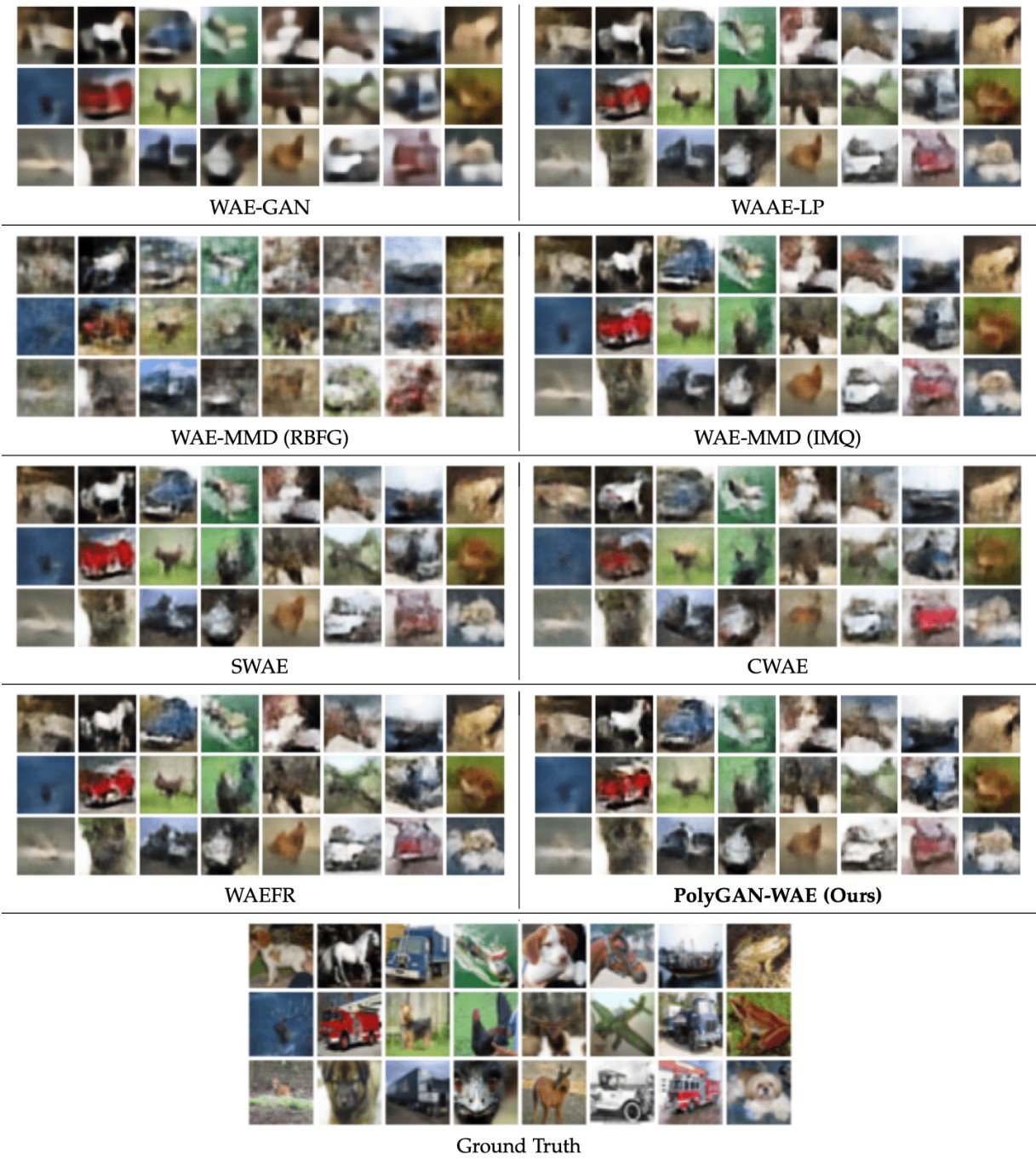

Figure 24: (🌑 Color online) Comparing the image reconstruction performance on CIFAR-10 dataset. PolyGAN-WAE reconstructions are sharper than the baselines. WAE-MMD (RBFG) does not generate good reconstructions. The adversarial nature of training in WAE variants with a trainable discriminator (WAE-GAN and WAAE-LP) results in poorer performance and blurry reconstructions.

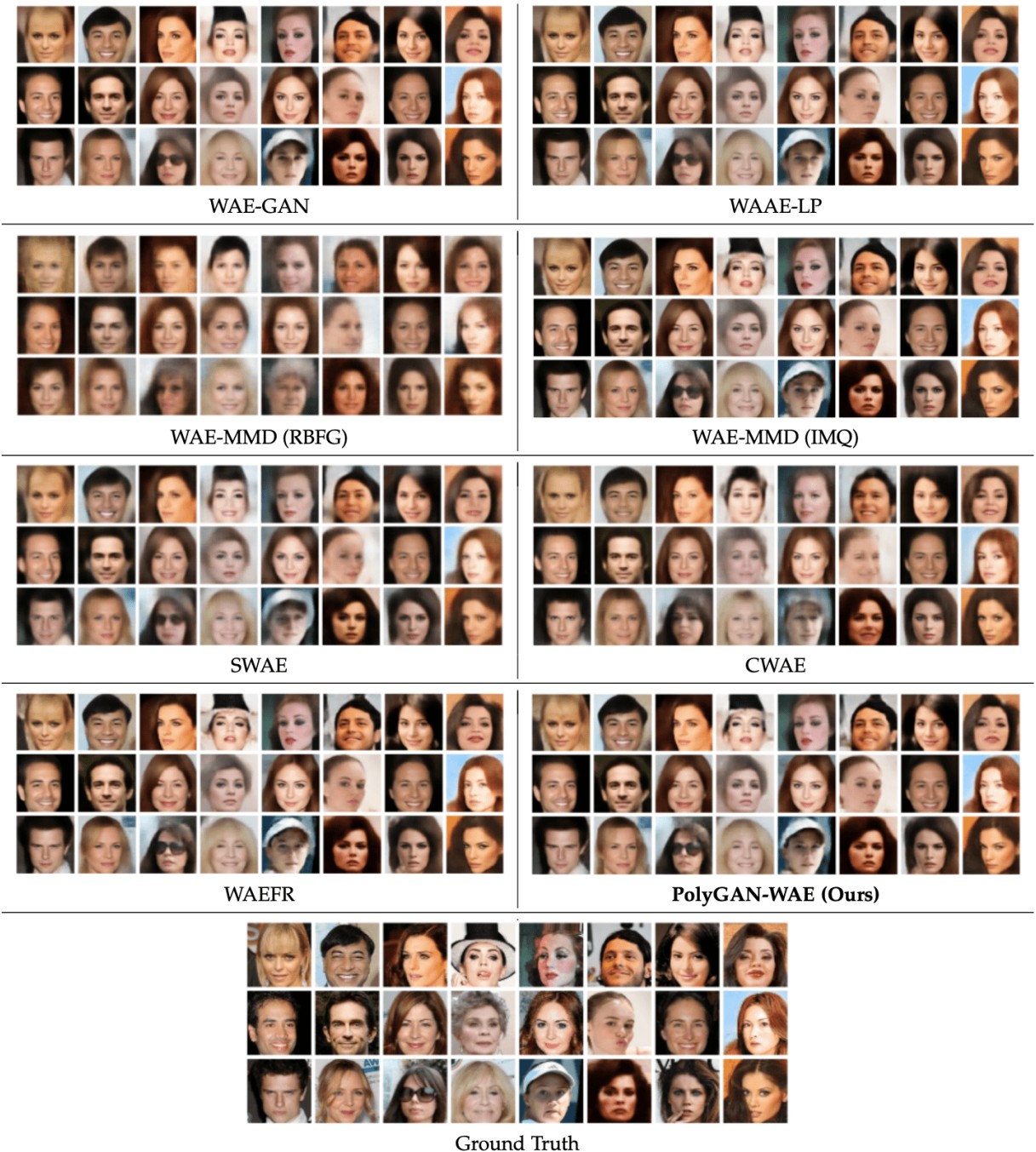

Figure 25: (🎨 Color online) Comparison of image reconstruction performance on CelebA dataset. PolyGAN-WAE and WAE-MMD (IMQ) generate reconstructions that are closest to the ground-truth images. PolyGAN-WAE is also able to recreate the colors more faithfully.

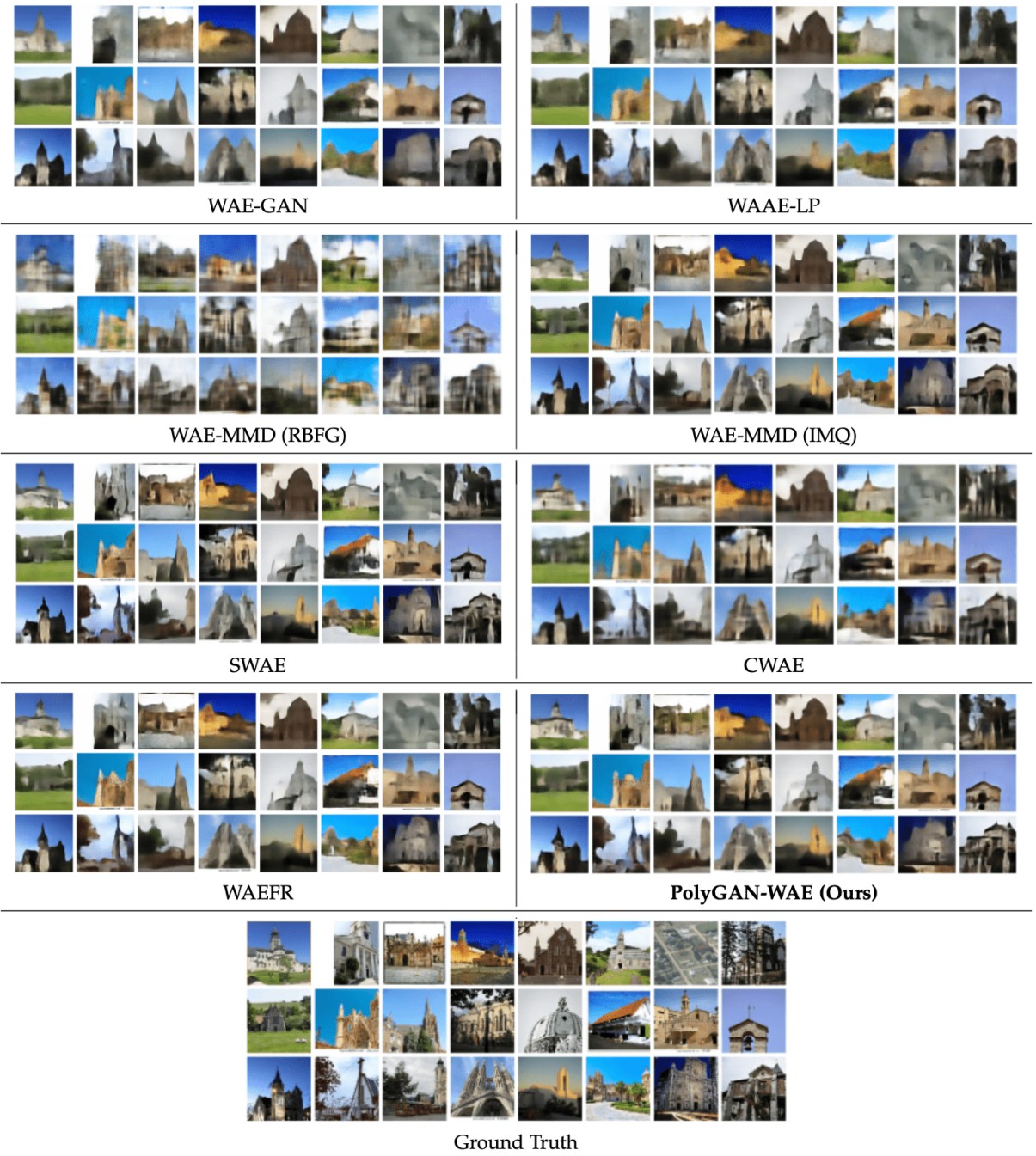

Figure 26: (🌀 Color online) Image reconstruction performance on the LSUN-Churches dataset. PolyGAN-WAE is comparable to WAE-MMD (IMQ) and CWAE on LSUN-Churches. As in the case of CIFAR-10 (cf. Figure 24), WAE variants with a trainable discriminator result in images of poorer visual quality than those with closed-form discriminators.

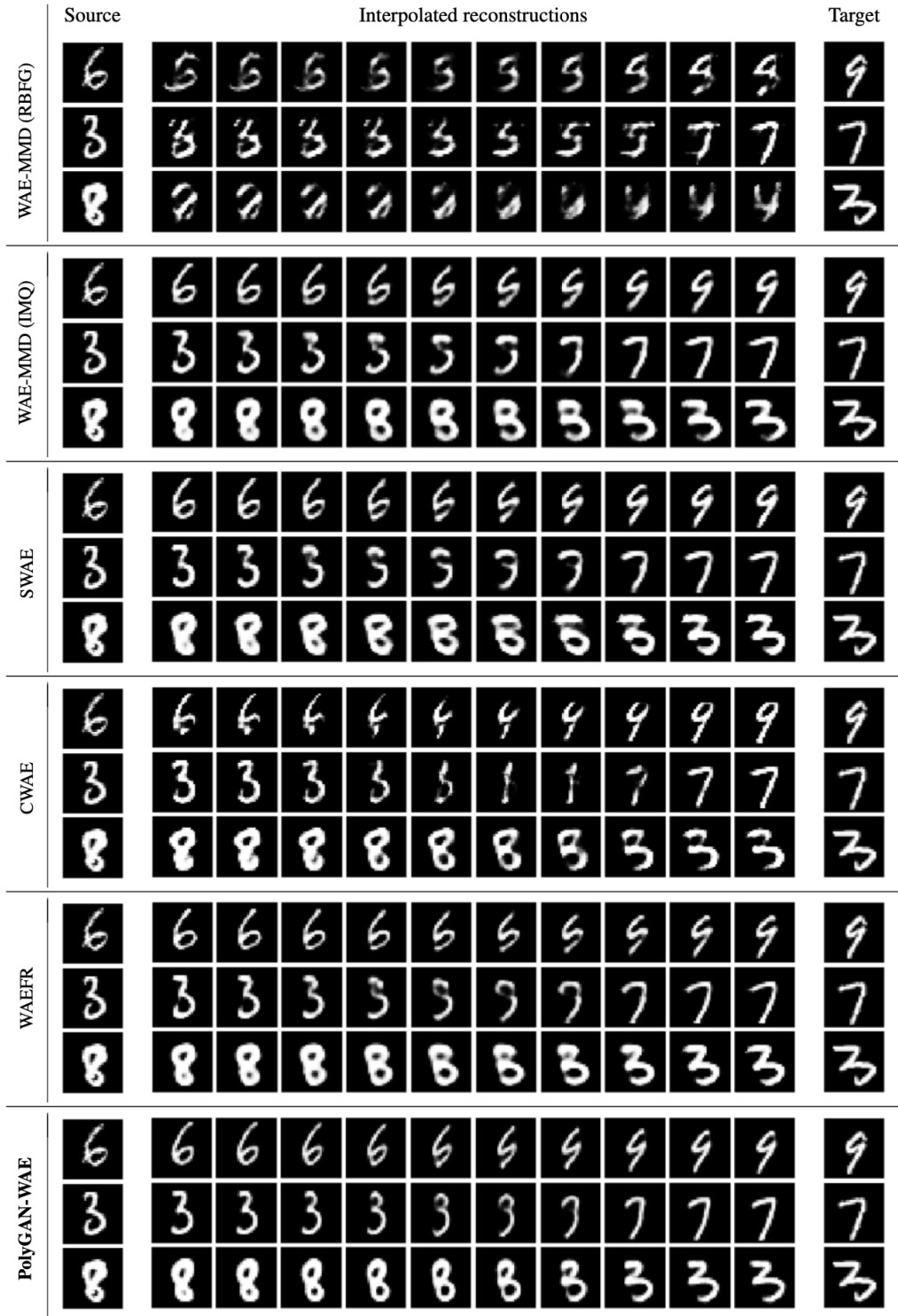

Figure 27: Images generated by decoding interpolated latent space representations of images drawn from MNIST. PolyGAN-WAE generates sharper interpolations than the baselines. CWAE and WAE-MMD (RBFG) perform an unsatisfactory job of interpolation.

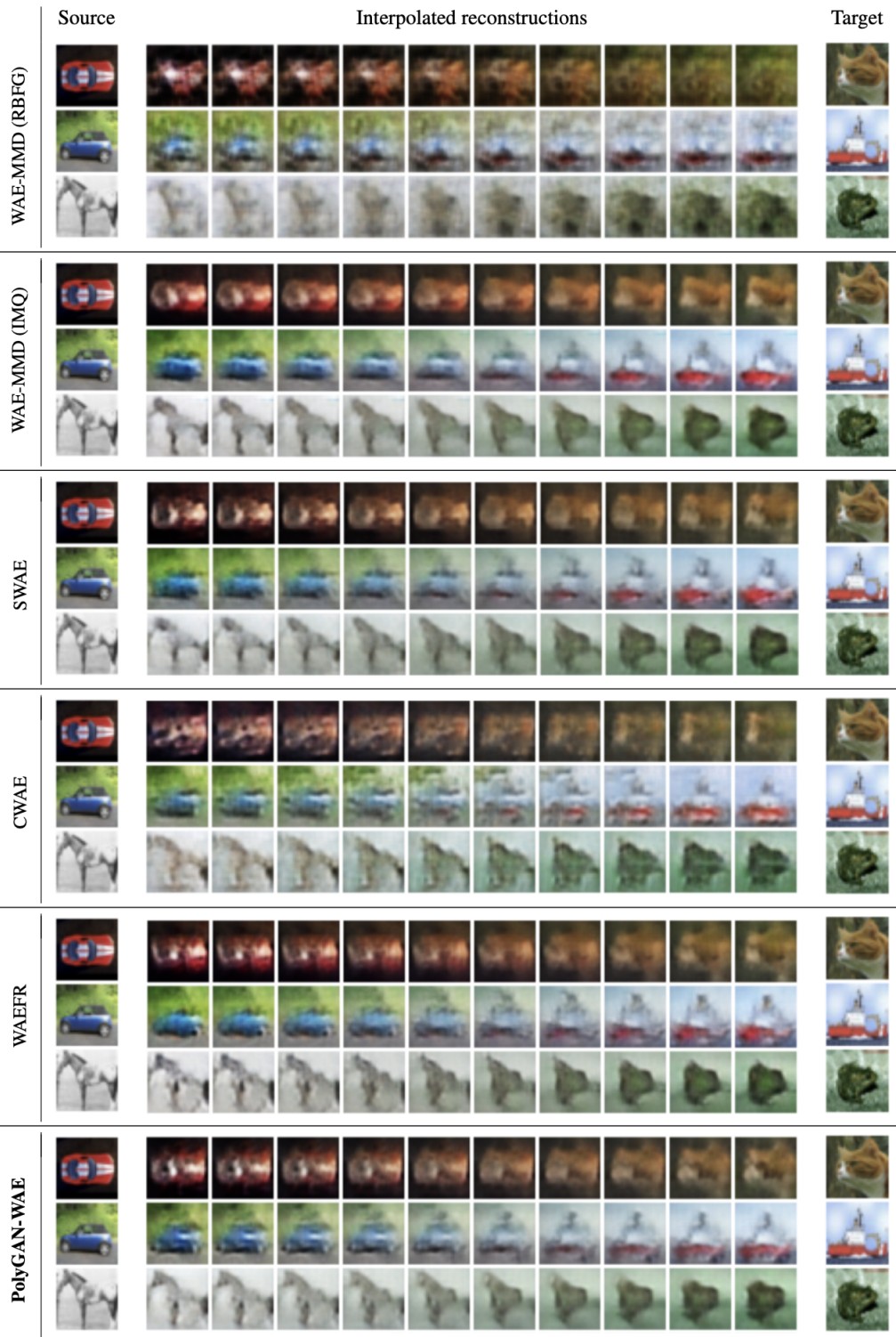

Figure 28: Images generated by decoding the interpolated latent-space vectors of the CIFAR-10 dataset. The interpolations in PolyGAN-WAE, WAEFR, and WAE-MMD (IMQ) are visually closer to the source and target images than those generated by the other models.

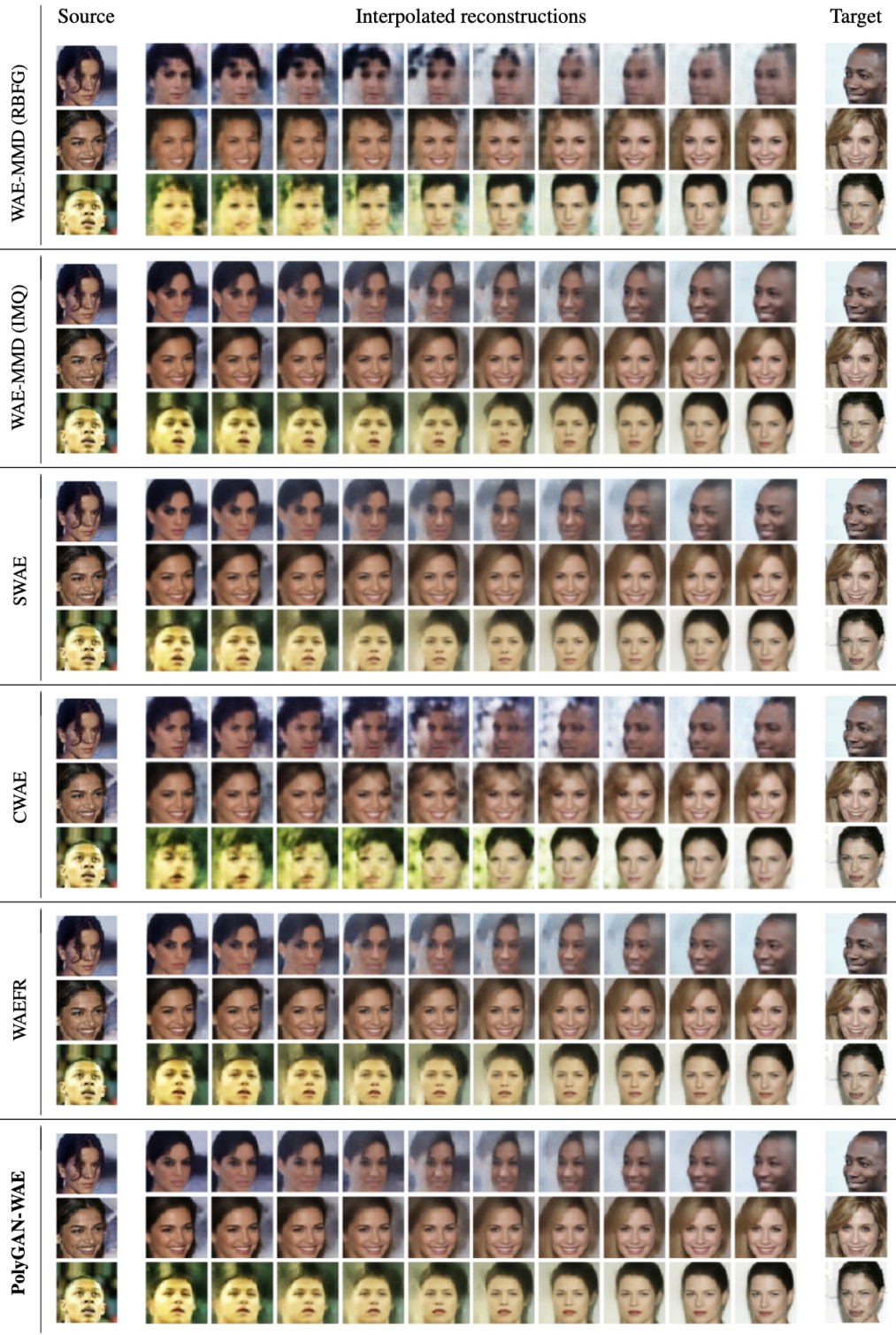

Figure 29: Images generated in the interpolation experiment on CelebA dataset. The source and target images are drawn from a held-out validation set. The images generated by PolyGAN-WAE are visually superior to the baselines. The PolyGAN-WAE generator also recreates the features of the source and target images more accurately.

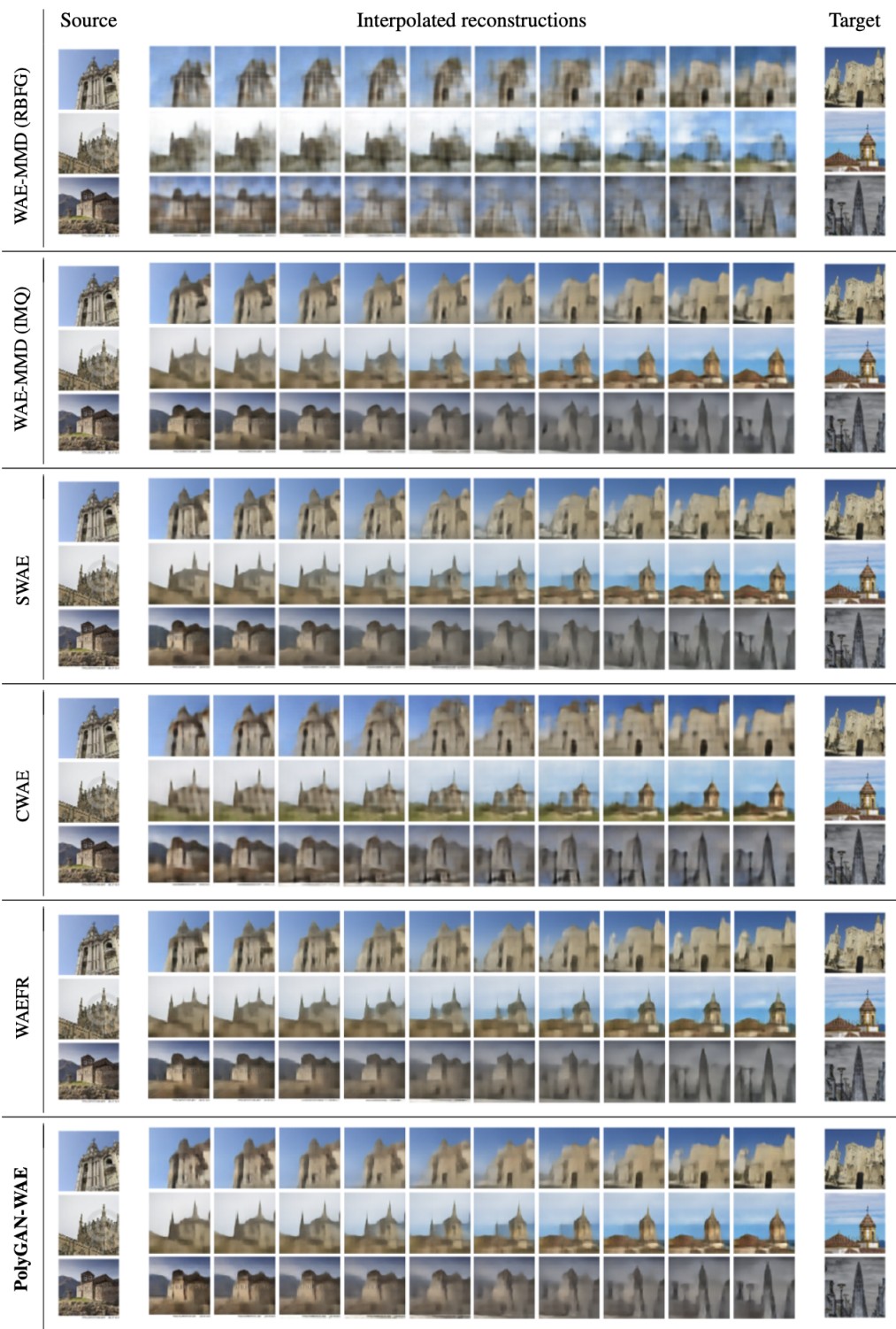

Figure 30: Images generated by decoding interpolated points between the latent space representations of pairs of images drawn from LSUN-Churches. Interpolations in SWAE are oversmooth, while those generated by PolyGAN-WAE and CWAE are sharper and comparable with each other.

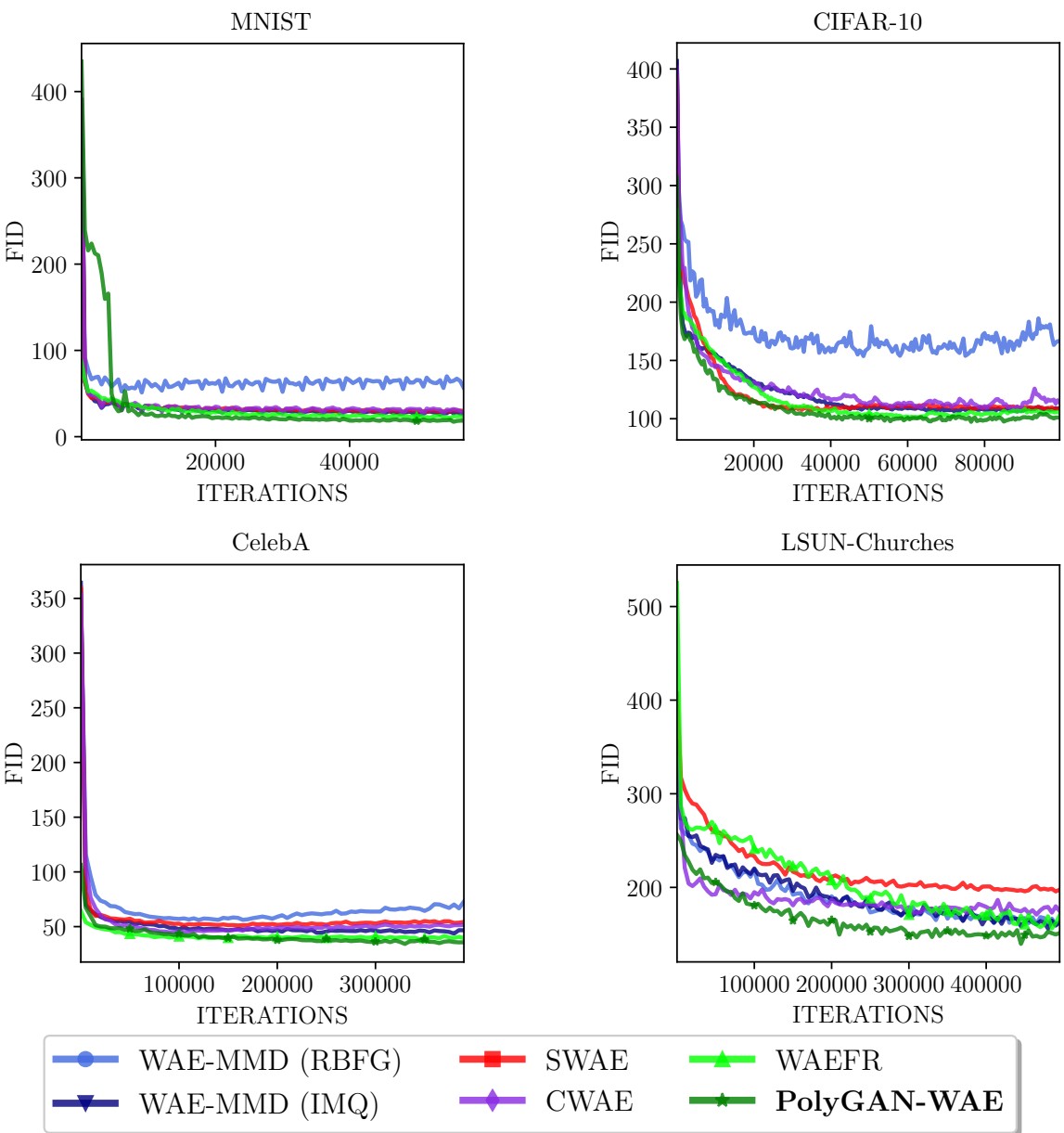

Figure 31: Comparison of FID as a function of iterations for the various WAE flavors considered. PolyGAN-WAE outperforms the baselines and saturates to better (lower) values of FID on all the datasets considered.

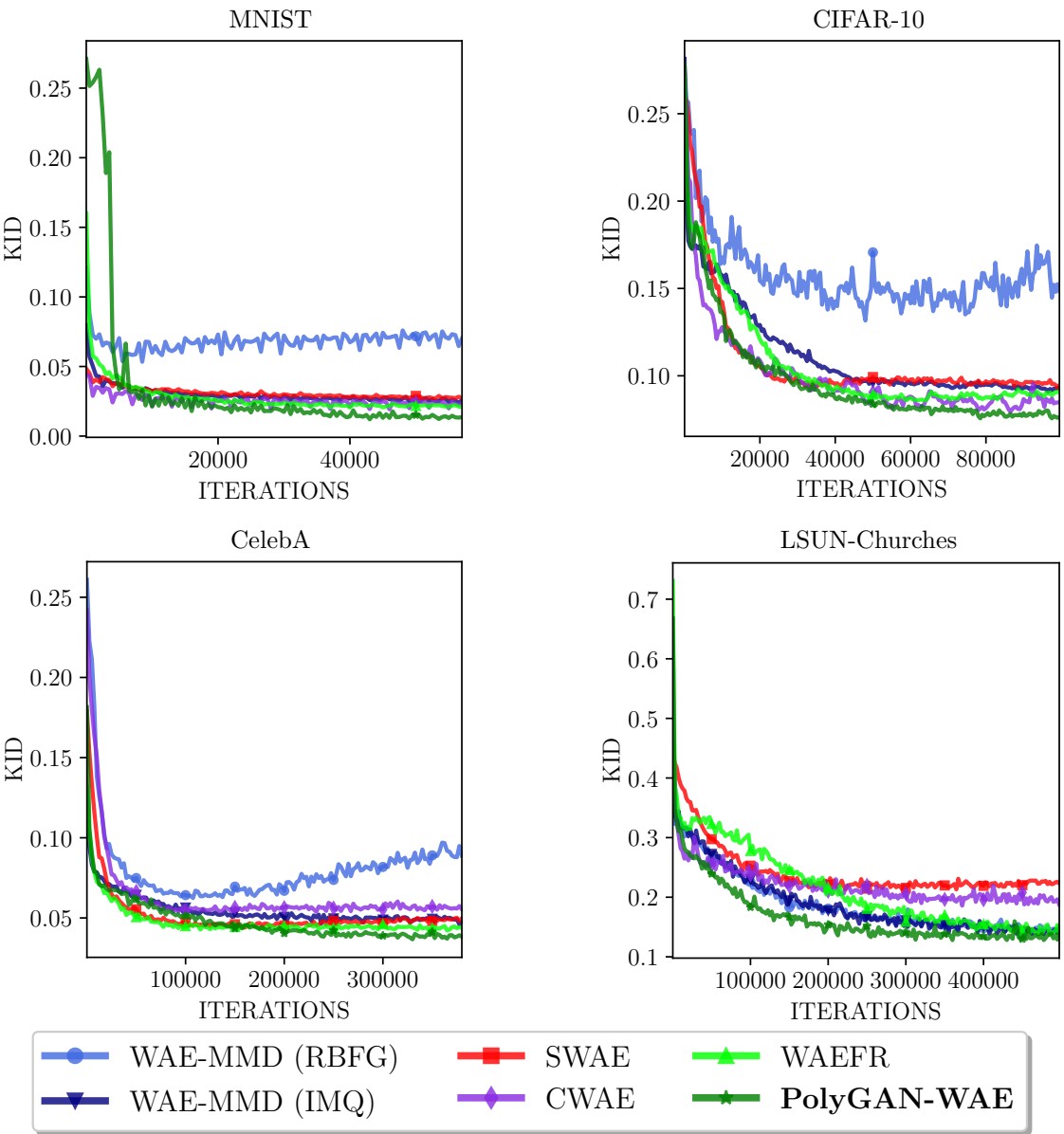

Figure 32: A comparison of the kernel inception distance (KID) as iterations progress for PolyGAN-WAE and the baselines. WAE-MMD with the Gaussian (RBFG) kernel fails to converge on most datasets. PolyGAN-WAE achieves the lowest KID in all the cases, and convergence is twice as fast as the baselines on LSUN-Churches dataset.

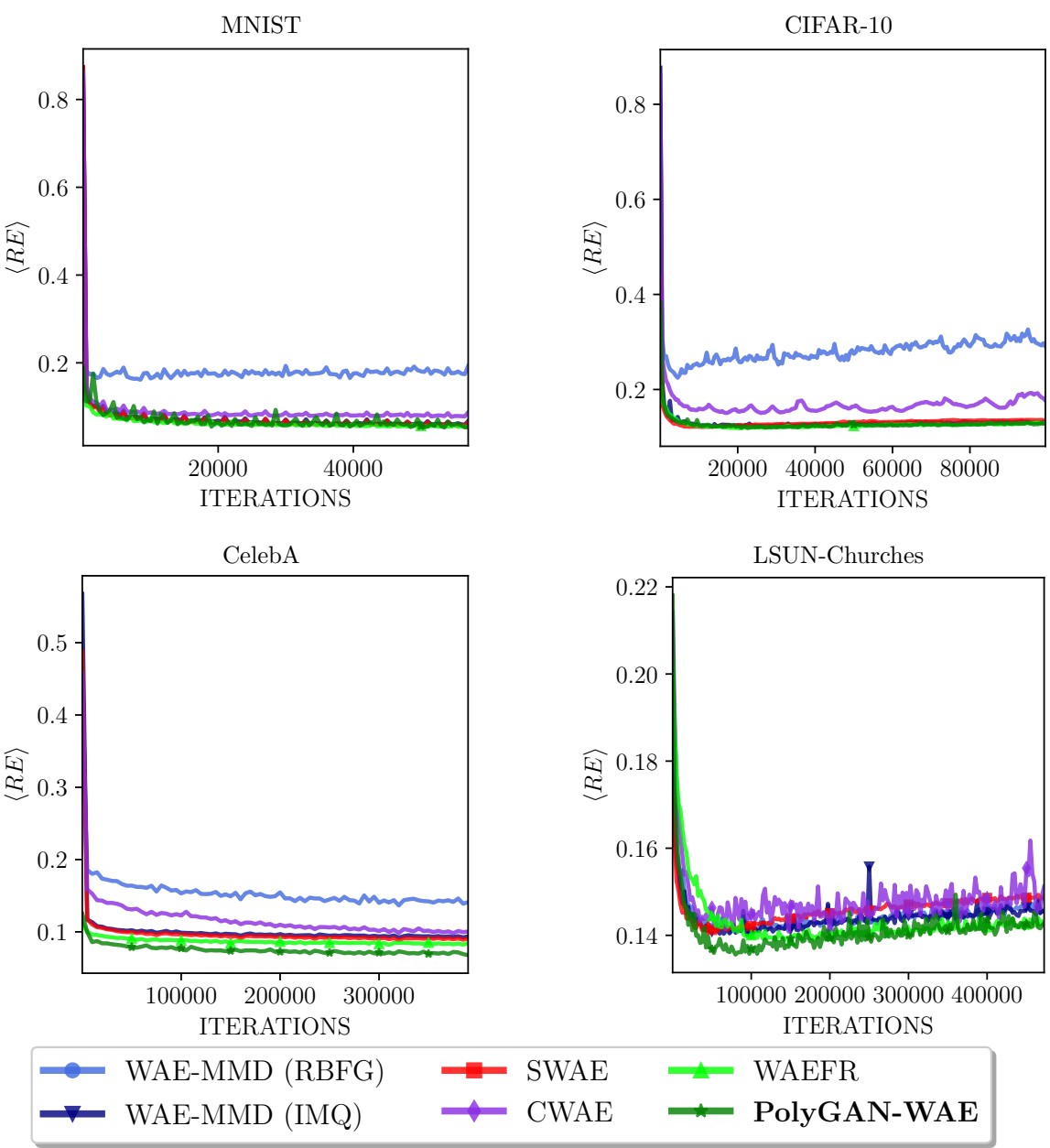

Figure 33: Average reconstruction error $\langle RE \rangle$ versus iterations for various WAE flavors considered. PolyGAN-WAE is comparable to the baseline methods on MNIST and CIFAR-10, while achieving superior convergence on datasets involving high-dimensional (128-D) latent space representations, such as CelebA and LSUN-Churches.

### F.5 Latent-space Matching with PolyGAN-D

We present comparisons on using the Poly-WGAN discriminator in MMD-GAN architectures on the MNIST and CIFAR-10 datasets. We consider the following baselines: (i) The MMD-GAN (RBFG) network with an autoencoding discriminator (Li et al., 2017a). The autoencoder is trained to minimize the $L_2$ cost, while the encoder is additionally trained to maximize the MMD kernel cost. The encoder weights are clipped to the range $[-0.01, 0.01]$. For every generator update in the first 25 updates, the discriminator is updated 100 times. Subsequently, the discriminator is updated five times per generator update. (ii) The MMD-GAN-GP (RBFG) and MMD-GAN-GP (IMQ) networks (Bińkowski et al., 2018), where the decoder network is removed, in favor of training the encoder to simultaneously minimize the WGAN-GP gradient penalty and maximize the MMD cost. As in the MMD-GAN case, the discriminator is updated five times per generator update.

In PolyGAN-D, we consider the MMD-GAN autoencoding discriminator architecture. The encoder and decoder are trained to minimize the $L_2$ reconstruction error. The latent-space of the encoder is provided as input to the polyharmonic RBF network. Unlike MMD-GANs, we do not train the PolyGAN-D encoder on $\mathcal{L}_D$. The generator network minimizes the WGAN cost in all cases. Following the approach of Li et al. (2017a), we pre-train the autoencoder for 2500 iterations. Subsequently, the autoencoder is updated once per generator update. We compare the *system times* between generator updates over batches of data in the MMD-GAN and WAE training configurations. The computation times were measured when training the models on workstations with Configuration I described in Appendix D.3 of the *Supporting Document*.

Figure 34 depicts the images output by the converged generator in PolyGAN-D and the baselines. The images generated by PolyGAN-D are visually on par with those output by MMD-GAN (RBFG) and MMD-GAN-GP (IMQ). MMD-GAN-GP (RBFG) performed poorly on CIFAR-10, which is in agreement with the results reported by Bińkowski et al. (2018). Table 12 presents the best-case FID scores computed using PolyGAN-D and the converged baselines. MMD-GAN-GP (IMQ) resulted in the lowest FID scores on both datasets. PolyGAN-D performs on par with MMD-GAN (RBFG).

From Table 13 we observe that MMD-GAN and MMD-GAN-GP have training times up to two orders of magnitude higher than PolyGAN-WAE as they update the discriminator multiple times per generator update. Among the WAE variants, WAE-GAN is slower by an order, owing to the additional training of the discriminator network. PolyGAN-WAE is on par with other kernel-based methods, while still incorporating a discriminator network, whose weights are computed *one-shot*. From Table 13 we observe that MMD-GAN and MMD-GAN-GP have training times up to two orders of magnitude higher than PolyGAN-WAE as they update the discriminator multiple times per generator update. PolyGAN-WAE scales better with dimensionality, compared to WAE-MMD. We attribute this to the increased complexity in computing the baseline RBFG and IMQ kernels in high dimensions. The Poly-WGAN discriminator complexity is only affected by the number of centers in the RBF expansion.

Table 12: A comparison of MMD-GAN flavors and PolyGAN-D in terms of FID. The performance of PolyGAN-D is comparable to MMD-GAN baseline with a trainable auto-encoder discriminator network.

| GAN Flavor $\longrightarrow$ | MMD-GAN (RBFG) | MMD-GAN-GP (RBFG) | MMD-GAN-GP (IMQ) | **PolyGAN-D (Ours)** |
|---|---|---|---|---|
| MNIST | 21.310 | 24.108 | **16.642** | 20.271 |
| CIFAR-10 | 55.452 | 64.571 | **49.255** | 53.180 |

Table 13: A comparison of **average compute time per batch (in seconds)** of samples when training various WAE and MMD-GAN models. The standard deviation was approximately $10^{-3}$ in all the cases considered. $\#D$ denotes the number of discriminator updates performed per generator update. Kernel methods are, on the average, an order of magnitude faster than GANs with a trainable discriminator network. The training time per batch is lowest for PolyGAN-WAE, on par with WAE-MMD based approaches, while implementing the optimal GAN discriminator one-shot. PolyGAN-WAE is least affected by increasing the dimensionality of the latent space of the input data.

| GAN Flavor | #D | WAE-GAN | WAE-MMD | SWAE | CWAE | WAEFR | **PolyGAN-WAE** | MMD-GAN | MMD-GAN-GP | **PolyGAN-D** |
|---|---|---|---|---|---|---|---|---|---|---|
| MNIST | 1 | 0.072 | 0.029 | 0.052 | 0.029 | 0.036 | **0.022** | 0.321 | 0.163 | 0.201 |
| (16-D) | 5 | 0.294 | - | - | - | - | - | 1.053 | 0.869 | - |
| CIFAR-10 | 1 | 0.082 | 0.036 | 0.047 | 0.036 | 0.039 | **0.023** | 0.338 | 0.243 | 0.258 |
| (64-D) | 5 | 0.328 | - | - | - | - | - | 1.110 | 0.938 | - |

Table 14: A comparison of number of trainable (T) and fixed (F) parameters present in each WAE and MMD-GAN variant considered. An ✗ denotes that the network is not present in that flavor. The inclusion of an RBF discriminator in PolyGAN-WAE does not change the training performance as the number of trainable parameters remains unaffected. MMD-GAN-GP has the fewest number of parameters, but incorporates adversarial training, unlike the WAE variants.

| GAN flavor | Adversarial Training | | Generator | Encoder | Decoder | Discriminator | Total Paramerters |
|---|---|---|---|---|---|---|---|
| WAE-MMD | ✗ | T | ✗ | $12 \times 10^6$ | $11.5 \times 10^6$ | ✗ | $23.5 \times 10^6$ |
| | | F | ✗ | $4 \times 10^3$ | $2 \times 10^3$ | ✗ | $\mathbf{6 \times 10^3}$ |
| SWAE | ✗ | T | ✗ | $12 \times 10^6$ | $11.5 \times 10^6$ | ✗ | $23.5 \times 10^6$ |
| | | F | ✗ | $4 \times 10^3$ | $2 \times 10^3$ | ✗ | $\mathbf{6 \times 10^3}$ |
| CWAE | ✗ | T | ✗ | $12 \times 10^6$ | $11.5 \times 10^6$ | ✗ | $23.5 \times 10^6$ |
| | | F | ✗ | $4 \times 10^3$ | $2 \times 10^3$ | ✗ | $\mathbf{6 \times 10^3}$ |
| WAEFR | ✗ | T | ✗ | $12 \times 10^6$ | $11.5 \times 10^6$ | 0 | $23.5 \times 10^6$ |
| | | F | ✗ | $4 \times 10^3$ | $2 \times 10^3$ | $2.5 \times 10^5$ | $2.56 \times 10^5$ |
| **PolyGAN-WAE** | ✗ | T | ✗ | $12 \times 10^6$ | $11.5 \times 10^6$ | 0 | $23.5 \times 10^6$ |
| | | F | ✗ | $4 \times 10^3$ | $2 \times 10^3$ | $4 \times 10^3$ | $9 \times 10^3$ |
| MMD-GAN | ✓ | T | $2 \times 10^6$ | $12 \times 10^6$ | $11.5 \times 10^6$ | ✗ | $25.5 \times 10^6$ |
| | | F | $2.5 \times 10^4$ | $4 \times 10^3$ | $2 \times 10^3$ | ✗ | $3.1 \times 10^4$ |
| MMD-GAN-GP | ✓ | T | $2 \times 10^6$ | $12 \times 10^6$ | ✗ | ✗ | $\mathbf{14 \times 10^6}$ |
| | | F | $2.5 \times 10^4$ | $4 \times 10^3$ | ✗ | ✗ | $2.9 \times 10^4$ |
| **PolyGAN-D** | ✓ | T | $2 \times 10^6$ | $12 \times 10^6$ | $11.5 \times 10^6$ | 0 | $25.5 \times 10^6$ |
| | | F | $2.5 \times 10^4$ | $4 \times 10^3$ | $2 \times 10^3$ | $4 \times 10^3$ | $3.5 \times 10^4$ |

Figure 34: Images output by the generator in PolyGAN-D and baseline MMD-GAN variants. The performance of PolyGAN-D is comparable to that of MMD-GAN (RBFG) with the autoencoder discriminator architecture.

