# OpenReview forum: "Optimal Discriminators for GANs with Higher-order Gradient Regularizers"
_TMLR — Rejected by TMLR_

### Review · Reviewer_NVin · 2022-12-13

**Summary Of Contributions:**

This paper analyzes the training of Integral Probability Metrics (IPM) GANs with the variational calculus. The authors find the optimal discriminator is the solution of a partial differential equation (PDE) and can be represented via a polyharmonic radial basis function (RBF) interpolation. Thus the Poly-GAN variants, including Poly-LSGAN and Poly-WGAN are proposed as new variants of the GAN family, which possess better convergence and training stability. In the low-dimensional case, the proposed method is able to match the distribution, while in the high-dimensional case, the proposed method relies on WAE as a proxy and aligns the encoded distribution with the prior distribution. Both theoretical analysis and empirical studies are conducted in the paper.

**Audience:**

Yes

**Broader Impact Concerns:**

There is no Broader Impact Statement found in the paper. Some potential ethical concerns are similar to the previous GANs studies, and I did not find special ethical concerns for this paper.

**Claims And Evidence:**

No

**Requested Changes:**

### Quality

There is a concern regarding the evaluation of image experiments in section 7. The performance is evaluated in image space through FID, KID, etc. However, all the methods are deployed to align the latent space of WAE. Thus the performance currently shown in the paper is not sufficient to support the point of view on image data. The authors might need to show a comparison on how well the latent space is aligned using a distribution distance, or directly compare PolyGAN with other GAN models in the image space (though this may be difficult).

Besides the image case, where the dimensionality is usually high, the authors may consider other data modalities, such as text data, and tabular data, to demonstrate the effectiveness of the proposed method.


### Presentation clarity

> Analysis part

The discriminator D(x) defined in $\mathcal L_D^W$ might need a different notation (for example, D^W(x)) to be distinct from a generic discriminator D(x) that is not Lipschitz-1.

$|\mathcal X|$ defined in Eq (1) needs to be explained.

> Experimental sections

Fig. 2:  it is a little bit hard to compare the case of m=1 and m=2, as only subplot (a) contains these two cases. However, the caption denotes the legend in commonly used for (a) - (d). This part is confusing to the readers.

On the choice of polynomial degree, from Fig. 2(b) it is hard to support "the performance is the best for m = floor(n/2)", as all the choices of m seem to converge to a similar point. However, we can observe the case of $m=1$ has better convergence and training stability. Do the authors want to highlight this point?


> Typos:

- P2: "backpropagation based" ->  "backpropagation-based"

- P5 (below Eq(3)): (\bf x_i, y_i) -> (\bf c_i, y_i)

- P8 (Theorem 5.2): shall we use $\mu_p(\bf x)$?

**Strengths And Weaknesses:**

### Strengths

The paper is overall well-written and organized, making it easy to follow. The proposed loss is mathematically supported and well-motivated. The authors demonstrate the connection between the gradient regularization of the discriminator and the polyharmonic radial basis function, which is interesting and provide novel insight to the community. From the empirical perspective, the authors have shown the proposed method has better convergence and training stability in both image space and latent space. The experimental details, such as implementation, experiment settings and demo code are provided, and should be sufficient for reproducibility.

### Weakness

The motivation for connecting the gradient regularization of the discriminator and the polyharmonic radial basis function is a bit unclear. Although I found in the extended workshop paper (Anonymous, 2022b) that Poly-LSGAN and (the extended) Poly-WGAN is able to overcome mode-collapse issue in GAN training, this point is not well elaborated nor well supported with experiments.

The gradient order is determined in a relatively heuristic way as $m=\frac{n}{2}$, and requires more theoretical supports. The analysis provided in appendix C.5 provides some interpretation of the choices but does not directly support this choice is optimal.

The proposed method is computationally expensive, which limits its usage in high-dimensional cases. Notably, the proposed method is even expensive to be deployed on MNIST data.

Since the image experiments are conducted with WAE, the evaluation of the latent space alignment may also be needed. (Please see the requested change.)

Considering the computation is expensive, a comprehensive comparison of the computation cost (e.g. GPU time) might be helpful to let the readers understand and trade-off between the challenge potential mode-collapse and the training cost.

---

> ### Author Response · Authors · 2023-03-18
> **Response to Reviewer NVin (Part 1)**
>
>
> Thank you for the insightful comments and suggestions. Please find below, our point-by-point response to your comments.
>
> ## Weaknesses
>
>  > __(C.1)__ The motivation for connecting the gradient regularization of the discriminator and the polyharmonic radial basis function is a bit unclear. Although I found in the extended workshop paper (Anonymous, 2022b) that Poly-LSGAN and (the extended) Poly-WGAN is able to overcome mode-collapse issue in GAN training, this point is not well elaborated nor well supported with experiments.
>
> __Response__: Thin-plate splines (Harder and Desmarias, 1972) solve the gradient-norm regularized least-squares interpolation problem in 2-D. The polyharmonic kernel interpolator solves the general higher-order gradient-regularized interpolation problem in $n$-D (Duchon, 1977). The WGAN discriminator assigns a positive value to reals and a negative value to fakes. It functions effectively as an interpolator, whose output depends on the neighborhood of the input sample. This interpolating property of the discriminator, coupled with the connection between gradient-regularization and polyharmonic kernels is established in this paper. The motivation is now a separate section (__Section 1.1__).
>
> > __(C.2)__  The gradient order is determined in a relatively heuristic way as $m=\frac{n}{2}$ and requires more theoretical supports. The analysis provided in appendix C.5 provides some interpretation of the choices but does not directly support this choice is optimal.
>
> __Response__: The choice of the gradient order, $m=\frac{n}{2}$ is made taking into account the training stability of the generator. Gradient orders $m<\frac{n}{2}$ are non-interpolating (Duchon, 1977) and can cause numerical instability owing to the negative exponent of the norm. As discussed in Appendix C.5, all orders $m\geq\frac{n}{2}$ are interpolating, and result in smooth discriminators. However, empirically, we observed that the use of polyharmonic kernels of order $m\geq\frac{n}{2}+3$ results in gradient explosion during the initial stages of training. Experiments in Section 6.1 validate this claim (cf. Figure 3(e)-(f) of the revised manuscript). We have included a new section in the revised manuscript on _"Interpreting the PolyGAN Discriminator''_ (__Section 5.1__).
>
> > __(C.3)__ The proposed method is computationally expensive, which limits its usage in high-dimensional cases. Notably, the proposed method is even expensive to be deployed on MNIST data. Since the image experiments are conducted with WAE, the evaluation of the latent space alignment may also be needed. (Please see the requested change.)
>
> __Response__: We have included experiments on latent-space alignment (cf. Response to Comment __(C.5)__ to Reviewer _NVin_; and __Appendix F.4__, __Figure 17__ and __Table 10__ of the revised manuscript). We have also included experiments on learning the latent-space distribution of pre-trained autoencoders using Poly-WGAN (cf. Response to Comment __(C.1)__ to Reviewer _toYB_, and __Appendix F.3__ of the revised manuscript).
>
> > __(C.4)__ Considering the computation is expensive, a comprehensive comparison of the computation cost (e.g. GPU time) might be helpful to let the readers understand and trade-off between the challenge potential mode-collapse and the training cost.
>
> __Response__: A comparison of the compute time between PolyGAN-WAE, PolyGAN-D and the baselines was already provided in Appendix F of the initial submission (cf. __Appendix F.5__ of the revised manuscript). We have now included comparisons on the time taken between generator updates for Poly-WGAN versus the baselines, for varying number of discriminator updates (cf. __Appendix E.3__ of the revised manuscript). We have also compared the compute times between Poly-WGAN and the baselines for learning the latent-space distribution of pre-trained autoencoders (cf. __Table 9__ of the revised manuscript).

---

> > ### Author Response · Authors · 2023-03-18
> > **Response to Reviewer NVin (Part 2)**
> >
> > ## Requested Changes
> >
> > > __(C.5) Quality:__ There is a concern regarding the evaluation of image experiments in section 7. The performance is evaluated in image space through FID, KID, etc. However, all the methods are deployed to align the latent space of WAE. Thus the performance currently shown in the paper is not sufficient to support the point of view on image data. The authors might need to show a comparison on how well the latent space is aligned using a distribution distance, or directly compare PolyGAN with other GAN models in the image space (though this may be difficult).
> > Besides the image case, where the dimensionality is usually high, the authors may consider other data modalities, such as text data, and tabular data, to demonstrate the effectiveness of the proposed method.
> >
> > __Response__: We have included experiments on latent-space alignment, and compared the performance of Poly-WAE against baselines in terms of the $\mathcal{W}^{2,2}$ distance between the latent-space distribution and the target Gaussian (cf. __Appendix F.4__, __Figure 17__ and __Table 10__ of the revised manuscript). PolyGAN-WAE achieves a lower $\mathcal{W}^{2,2}$ score compared to the baselines for all the datasets considered.
> >
> > Given our interest in image processing, our focus in the experiments is on image-learning tasks. We have little expertise in text processing, and hence we could not conduct those experiments. Extensions to other types of data is a promising direction for future research.
> >
> > > __(C.6)__ The discriminator $D(x)$ defined in   might need a different notation (for $L_D^W$ example, $D^{W}(x)$) to be distinct from a generic discriminator D(x) that is not Lipschitz-1. $|\mathcal{X}|$ defined in Eq (1) needs to be explained.
> >
> > __Response__: We use $D^{\mathrm{L}}(x)$ to denote the Lipschitz-1 discriminator in the revised manuscript. $|\mathcal{X}|$ denotes the volume of the space $\mathcal{X}$. We have added this explanation in the revised manuscript (cf. highlighted text on Page 4).
> >
> > > __(C.7)__ Fig. 2: it is a little bit hard to compare the case of m=1 and m=2, as only subplot (a) contains these two cases. However, the caption denotes the legend in commonly used for (a) - (d). This part is confusing to the readers.
> > On the choice of polynomial degree, from Fig. 2(b) it is hard to support "the performance is the best for m = floor(n/2)", as all the choices of m seem to converge to a similar point. However, we can observe the case of $m=1$ has better convergence and training stability. Do the authors want to hilight this point?
> >
> > __Response__: We have revised the caption (cf. __Figure 3__ of the revised manuscript). The $m=1$ case is the _best_ in the sense that it results in faster convergence to a lower $\mathcal{W}^{2,2}$ score (cf. Figure 3(e) of the revised manuscript). In the 2-D learning scenario, $m=1$ corresponds to the $m=\frac{n}{2}^{th}$ order. A similar argument can be made for the choice of $m = \frac{n}{2} = 3$ for the 6-D Gaussian learning task (cf. Figure 3(f) of the revised manuscript).
> >
> > > __(C.8)__ Typos:
> > P2: "backpropagation based" -> "backpropagation-based"
> > P5 (below Eq(3)): (\bf x_i, y_i) -> (\bf c_i, y_i)
> > P8 (Theorem 5.2): shall we use μp(x)?
> >
> >
> > __Response__: The typos have been fixed.

---

### Review · Reviewer_toYB · 2022-12-13

**Summary Of Contributions:**

This paper studies the optimal discriminators corresponding to the LSGAN and WGAN objectives under higher-order gradient penalties. In both cases, the authors derive their analytical expression as combinations of polyharmonic radial basis functions; in the case of WGAN, this allows them to deduce that the optimal generated distribution is indeed the target data distribution. This theoretical knowledge is then applied to generative modeling of low-dimensional Gaussians and standard image datasets (within the latent space of a WAE to avoid high-dimensional issues) where leveraging the exact expression of the discriminator with an appropriate order of gradient penalty order is shown to improve the performance of standard GAN and kernel-based models.

**Audience:**

Yes

**Claims And Evidence:**

No

**Requested Changes:**

Please refer to the above weaknesses for more details.

 - The purpose of the experiments should be clarified to properly assess them and their results should be improved for a better relevance w.r.t. the recent GAN literature.
 - The formulation and implications of Theorem 3.1 should be clarified.
 - The optimization space and optimality/uniqueness results of Theorems 3.1 and 5.1 should be made more explicit.
 - The relevance of Theorem 5.2 w.r.t. GAN practice should be discussed for a proper evaluation of the claim.
 - Missing references should be discussed and possibly considered in the comparisons.
 - The paper motivation and writing could be improved following the above recommendations.

**Strengths And Weaknesses:**

**Foreword.** Given the length of the submission, my lack of knowledge about the mathematical tools used in the appendix and the mere two weeks of reviewing time, I am not able to fully assess the correctness of the supplementary material, including the proofs of all theoretical results, in accordance with TMLR guidelines stating that such evaluation is at the discretion of the reviewers.

## Strengths

This paper tackles a **relevant problem**: determining the exact optimal discriminator expression, while taking into account higher-order gradient penalties which have not received much attention in the GAN literature. This constitutes an opportunity to assess their relevance for generative modeling.

The obtained theoretical results on the discriminator are **interesting, new and well contextualized**, especially for the WGAN version. Knowing the exact expression of the discriminator is a valuable result that may become the basis for more theoretical or practical work on GANs. The interpretation of discriminators as high-dimensional interpolators, while intuitive and already approached in previous work, is here motivated by a more **involved** theoretical study tackling higher-order gradient penalties.

Regarding the form, the paper is mostly **well-written and of high quality**. Overall, the provided explanations and results are **clearly presented in the main paper**, with the supplementary material containing **sufficient information to reproduce the results of the paper**, as far as I can tell.

## Weaknesses

I detail the weaknesses of the paper in the following, in decreasing order of importance.

### Experiments

**The purpose of the experiments is unclear and they consequently do not sufficiently support the theoretical analysis.**

If the goal is to assess the ability of the derived results to explain how GANs operate in practice, or to show that leveraging the analytical expression of the discriminator leads to a performance advantage, then the provided comparisons are not sufficient: comparisons should be made on the same model, both with a trained discriminator and with the analytical expressions derived in the paper. Moreover, experiments should include a GAN version where the discriminator is trained close to optimality instead of only a few steps per generator update.

If the goal is to propose a new GAN model based on analytical expressions of discriminators trained by higher-order gradient penalties, then the provided results are insufficient. While I would not ask for state-of-the-art performance, the results of Section 7 on real-world images are far from the standard of generative modeling even from a few years ago. Hence, it is difficult to conclude on the relevance of the model in today's literature. Furthermore, it would be beneficial to change the experiments so that:
 - the generative model is independent of the autoencoding technique, e.g. by pretraining an autoencoder and apply the generative model in the latent space afterwards (similarly to Rombach et al. (2022) to cite a recent work);
 - they focus on generative modeling in low-data regimes: this is a relevant topic (Karras et al., 2020), would better fit the computational requirements of Poly-GAN, and could allow it to outperform more up-to-date models.

Karras et al. Training Generative Adversarial Networks with Limited Data. NeurIPS 2020.\
Rombach et al. High-Resolution Image Synthesis with Latent Diffusion Models. CVPR 2022.

### Theoretical Results

The **formulation of Theorem 3.1 (optimal LSGAN discriminator) should be clarified**. From the theorem statement, it is not clear:
 - what is the optimization space for $D$ in Eq. (3) (this is partly answered, to my understanding, in the later preamble to Section 5);
 - whether $D^{\ast}$ is the unique minimizer to Eq. (3) in this optimization space;
 - whether $D^{\ast}$ even exists given the system of equations of Eq. (5);
 - whether $\mathbf{B}$ is assumed or proved to be invertible, and in any case, if it is invertible, why wouldn't $\boldsymbol{w} = 0$ given Eq. (5).

More generally, **the optimization space for both Theorems 3.1 and 5.1 (optimal LSGAN/WGAN discriminators) should be made explicit**. I believe that both result hold for functions over the convex hull $\mathcal{X}$ of the generated and data/target density, because the constraints of the loss functions outside of this domain are looser.

**Theorem 5.2 (optimal generated distribution) may not be relevant w.r.t. GAN practice**, for the following reasons.
 - Proving that $p_g$ is a valid distribution may not be useful, since in practice the generator produces a valid distribution via a pushforward of a base distribution like a standard Gaussian.
 - The optimality result only holds for the min-min (or min-max as usually considered in the literature) problem of Section 2. However, a line of previous works (cf. list below) have shown that GANs, in practice, do not solve this min-max problem because of alternating optimization, leading to validity issues in these types of result. Nonetheless, I understand that studying the effect of alternating optimization is particularly challenging.

Goodfellow. NIPS 2016 Tutorial: Generative Adversarial Networks. arXiv, 2017. Section 5.1.1.\
Metz et al. Unrolled Generative Adversarial Networks. ICLR 2017.\
Mescheder et al. Which Training Methods for GANs do actually Converge? ICML 2018.\
Hsieh et al. The Limits of Min-Max Optimization Algorithms: Convergence to Spurious Non-Critical Sets. ICML 2021.\
Franceschi et al. A Neural Tangent Kernel Perspective of GANs. ICML 2022.

### Related Work

The paper **misses references** to the recent literature on IPM GANs and kernel generative modeling (cf. possibly non-exhaustive list below). I would be interested in a discussion of their relevance and comparability in the experiments w.r.t. to the proposed approach, especially as some of them do derive and leverage in their experiments exact discriminator expressions via kernels.

Biau et al. Some Theoretical Insights into Wasserstein GANs. JMLR 22 (2021).\
Glaser et al. KALE Flow: A Relaxed KL Gradient Flow for Probabilities with Disjoint Support. NeurIPS 2021.\
Mroueh & Nguyen. On the Convergence of Gradient Descent in GANs: MMD GAN As a Gradient Flow. AISTATS 2021.\
Franceschi et al. A Neural Tangent Kernel Perspective of GANs. ICML 2022.\
Zhang et al. Single-level Adversarial Data Synthesis based on Neural Tangent Kernels. arXiv, 2022.

### Motivation

To the best of my knowledge, higher-order gradient penalties are not standard in GAN practice, unlike first-order ones. This hinders the potential applicability and the motivation of the paper to study those penalties. Strengthening the empirical study (cf. first weakness) would improve this point, as they indicate an ideal gradient penalty order in accordance with the theoretical results. Nevertheless, this is not critical as these results may be used in future works thanks to their generality.

I would also recommend the authors to nuance the following statements.
> The gradient of the discriminator would comprise a sum of dirac-delta functions, which is not compatible with backpropagation based learning. [Section 1, p. 2]

I understand that this is a widely accepted statement in the GAN literature, but it is questionable when the discriminator is implemented as a neural network as it is in practice.

>  Given an unseen sample $\boldsymbol{x}$, the output of a smooth discriminator should ideally depend on the values assigned to the points in the neighborhood of $\boldsymbol{x}$, which is precisely what kernel based interpolation achieves. [Section 1, p. 3]

This property is not exclusive to kernel-based methods: neural networks as used in practice also share it and could also be seen as high-dimensional interpolators, both in practice and in theory; cf. Jacot et al. (2018).

Jacot et al. Neural Tangent Kernel: Convergence and Generalization in Neural Networks. NeurIPS 2018.

### Writing

The paper would benefit from a slight reorganization; some suggestions:
 - details on polyharmonic radial basis functions, Beppo-Levi and Sobolev spaces could be gathered in the same section instead of being developed in both Section 3 and Section 5, to properly contextualize Theorem 3.1 and for better exposition;
 - Section 4 could be included in the empirical study of Sections 6 and 7 to avoid cutting the theoretical analysis, or at least be integrated in Section 3 to then motivate the WGAN analysis of Section 5;
 - the choice of paper title may not be ideal as it is quite general and does not highlight the main contributions of the paper -- deriving the expressions of optimal discriminators for GANs under higher-order gradient penalties.

### Typos, proofreading and quality

 - The "(Color online)" mention in each caption should be removed.
 - Figure 1 should be a vector graphics figure.
 - All equations in the paper should be numbered for better referencing.
 - "gradienft-regulazied" (Section 1.1, p. 3) should be "gradient-regularized".
 - $p_d + \mathcal{N}(\boldsymbol{0}, \mathbb{I})$ (Section 2, p. 4) should be written as a convolution: $p_d \ast \mathcal{N}(\boldsymbol{0}, \mathbb{I})$.
 - Section 3, proof of Theorem 3.1: "constitutes polyharmonic sum" -> constitutes a polyharmonic sum, "the polynomial component which represent" -> the polynomial component which represents, "The detail provided in Appendix B." -> The details are provided in Appendix B.
 - There is a missing space in "generative models such as GMMNs.The following" (Section 5.2, p. 8).
 - "Therefore, we set [...]" (Section 5.2, p. 9) is an incomplete sentence.

---

> ### Author Response · Authors · 2023-03-18
> **Response to Reviewer toYB (Part 1)**
>
> Thank you for the insightful comments and suggestions. Please find below, our point-by-point response to your comments.
>
> ## Weaknesses
>
>  > __(C.1)__ The purpose of the experiments is unclear and they consequently do not sufficiently support the theoretical analysis...
>
> __Response__: The goal of the experiments in Poly-WGAN is to gain a deeper understanding of how GANs work. To validate the hypothesis that trainable neural-network discriminators are approximating the form that Poly-WGAN implements, we considered experiments on training the baseline discriminator for $D_{\mathrm{iters}}$ iterations per generator update, and showed that, as $D_{\mathrm{iters}}$ increases, the performance of the baselines approaches that of Poly-WGAN (cf. __Appendix E.3__ , __Figure 8__ and __Table 8__ of the revised manuscript).
> To isolate the performance improvements obtained by using the Poly-WGAN discriminator to learn the latent space, we have included the suggested experiment on learning the latent-space distribution of a pre-trained autoencoder. Poly-WGAN outperforms the baselines on higher-dimensional latent-space models (CelebA with 63-D latent space). We also compare the performance against a trainable version of Poly-WGAN, where the kernel is fixed, but the centers and weights are learnt via gradient descent. The experiments and comparisons are provided in __Appendix F.3__, __Figure 13__ and __Table 9__ of the revised manuscript.
> Karras _et al.,_ (2020) train GANs under high-resolution low-data regimes (small cardinality of the dataset), which is different from low-dimensional data. The computational complexity in Poly-WGANs arises from the latter, and not the cardinality of the dataset. As a consequence, the results of Poly-WGAN are not readily applicable to the problem considered by Karras _et al.,_ (2020).
>
> > __(C.2)__ The formulation of Theorem 3.1 (optimal LSGAN discriminator) should be clarified...
>
>
> __Response__: The optimization in both Poly-LSGAN Poly-WGAN corresponds to drawing functions from the Beppo-Levi spaces of order $m$. We have included a discussion in __Section 3.1__ of the revised manuscript on the uniqueness of the solution. The necessary conditions for uniqueness (of weights and polynomial coefficients) are a distinct set of centers, and full column-rank for $\mathbf{\mathrm{B}}$ (Duchon, 1977). Please also see our Response to Comment __(C.2)__ to Reviewer _NfVB_.   However, the matrix $\mathbf{\mathrm{B}}$ is full-rank only if the centers do not lie on a low-dimensional subspace in $\mathbb{R}^n$, which is violated by ambient-dimension image vectors (_Manifold Hypothesis_). We have clarified this aspect in __Section 3__ of the revised manuscript.
> The solution $\mathbf{w}=0$ is trivial, as it corresponds to a polynomial interpolator with zeros at all the centers, and is of no practical use in GANs.
>
>
> > __(C.3)__ the optimization space for both Theorems 3.1 and 5.1 (optimal LSGAN/WGAN discriminators)...
>
>
> __Response__: The constraint space for both Poly-WGAN and Poly-LSGAN are Beppo-Levi spaces of order $m$. The solution space is defined over the domain $\mathcal{X}$, which is the convex hull of the supports of $p_d$ and $p_g$. Outside of this domain, the loss is identically zero, which makes the functional optimization irrelevant. We have included this discussion in __Section 3__ of the revised manuscript.
>
> > __(C.4)__ Theorem 5.2 (optimal generated distribution) may not be relevant w.r.t. GAN practice...
>
>
> __Response__: As the PolyGAN optimization problem is neither one of divergence minimization, nor one of IPM minimization over an RKHS, the result is nonobvious. Theorem 5.2 (__Theorem 4.2__ in the revised manuscript) shows that the optimal generator distribution indeed matches with the desired data distribution. While in practice, by virtue of how GANs are implemented, a valid push-forward distribution is assured, in theory, this is nonobvious and therefore, explicit constraints are needed. This will also take care of certain corner cases, such as those observed by _Anonymous (2023)._
>
>
> > __(C.5)__ On missing references...
>
> __Response__: Thank you for the references. We have included them in __Section 5.2__ on _"Related Work"_ in the revised manuscript to discuss the relation between PolyGAN and kernel-based and flow-based GAN approaches.
>
> > __(C.6)__ On the motivation, and on the ideal discriminator characteristics...
>
> __Response__: We have rewritten the Motivation section to place in better context the higher-order gradient regularizers. We have included discussion on relating kernel-based interpolations of PolyGAN and neural tangent kernels (cf. __Section 1.1__ of the revised manuscript). Please also see our Response to Comment __(C.1)__ from Reviewer _NVin_.

---

> > ### Author Response · Authors · 2023-03-18
> > **Response to Reviewer toYB (Part 2)**
> >
> >
> >
> > > __(C.7)__ Writing: The paper would benefit from a slight reorganization...
> >
> >
> > __Response__: Done. We now introduce the Beppo-Levi constraint space in __Section 3__ of the revised manuscript. The key results from the experimental validation on Poly-LSGAN, those crucial for motivating Poly-WGAN, are presented in  __Section 3.1__ of the revised manuscript. Detailed experiments have been provided in __Appendix E.1__ of the revised manuscript. The title has been revised to accurately reflect the claims of the paper.
> >
> > > __(C.8)__ Typos, proofreading and quality...
> >
> > __Response__: Thank you. The bugs have been fixed in the revised submission. Figure 1 is now a vector-graphics figure.
> >
> >
> >
> > -------------------
> > -------------------
> > ## Requested Changes
> >
> > > __(C.9)__ The purpose of the experiments should be clarified to properly assess them and their results should be improved for a better relevance w.r.t. the recent GAN literature.
> >
> > __Response__: Additional experiments have been included to improve the insights given by Poly-WGANs into the optimality of gradient-regularized GAN variants (cf. __Appendix E.3, F.3 and F.4__ ).
> >
> > > __(C.10)__ The formulation and implications of Theorem 3.1 should be clarified.
> >
> > __Response__: Done. Please see __Section 3__.
> >
> > > __(C.11)__ The optimization space and optimality/uniqueness results of Theorems 3.1 and 5.1 should be made more explicit.
> >
> > __Response__: The constraint space is introduced in __Section 3__ and also discussed in __Section 4__. The domain of the optimal discriminator is clarified in __Theorems 3.1 and 4.1__ of the revised manuscript.
> >
> >
> > > __(C.12)__ The relevance of Theorem 5.2 w.r.t. GAN practice should be discussed for a proper evaluation of the claim.
> >
> > __Response__: A discussion on the need for deriving the optimal generator distribution is now included in __Section 4__ of the revised manuscript.
> >
> > > __(C.13)__ Missing references should be discussed and possibly considered in the comparisons
> >
> > __Response__: A new subsection ( __Section 5.2__) has been included to discuss the related works on kernel-based and flow-based GAN approaches.
> >
> >
> > > __(C.14)__ The paper motivation and writing could be improved following the above recommendations
> >
> > __Response__: Done. A new subsection ( __Section 1.1__) has been included to clearly present the motivation for PolyGANs, and __Section 5.1__  presents the intuition behind the optimal discriminator learnt in PolyGANs.

---

> ### Comment · Reviewer_toYB · 2023-03-22
> **Remaining Concerns**
>
> I would like to thank the authors for their detailed answers, which I read in details along with the other reviews. Before providing a definitive recommendation to the AE, let me take this opportunity to follow up and express my remaining concerns.
>
> I would like to stress that the authors tackled and addressed many of my concerns in their response and revision. In particular:
> - the motivation to study higher-order gradient penalties is now clearer and highlighted at the beginning of the paper;
> - the suggested related work was properly discussed in the submission;
> - most of my questions on the theoretical results have been answered either in our discussion or in the paper;
> - some writing issues were addressed by a slight reorganization of the paper.
>
> However, I still think that main weaknesses either still hold or were incompletely addressed, in particular the following ones, listed below in decreasing order of importance. Points 1. and 2. are particularly crucial as they affect key claims of the paper and are also shared by the other reviewers.
> 1. The purpose of the experiments unfortunately remains unclear. While the authors state that "the goal of the experiments in Poly-WGAN is to gain a deeper understanding of how GANs work", the experimental section remains focused on experimental performance. This induces two problems which should be solved by clarifying the aims and conclusions of the experimental section.
>     - As stated in my initial review, the current performance achieved in the experiments is outdated, limiting their informative value beyond the fact that knowing the analytical discriminator expression is beneficial for generator training.
>     - The experiments, focusing on performance, thus provide little further information on GANs beyond the already developed theoretical results. They lack, for instance, a comparison between PolyWGAN and its strict equivalent with a trained discriminator, including the same gradient penalties, that would allow to further assess the relevance of the theoretical study.
> 2. While the provided clarifications on the validity conditions of the theoretical results as well as on the uniqueness and optimization space of the discriminator are appreciable, their integration in the revised manuscript should be improved. In the current state of the paper, these discussions are scattered around the theoretical results instead of being included in their statement, which obfuscates their interpretation. This should be solved by partially rewriting the theoretical parts to make assumptions clearer.
> 3. Even though the authors provided many clarifications and implemented some of my suggestions, the introduction of Beppo-Levi spaces remains scattered throughout the paper. Section 5 should probably be renamed as it mostly consists in discussions of the previous theoretical results, instead of being framed as "practical considerations".
> 4. Less importantly, some of my suggestions to nuance some claims were not completely taken into account.
>     - Section 1.1 still includes without discussion the claim that discriminators gradients can be sums of Dirac deltas, which may not be true when discriminators are implemented by neural networks in practice.
>     - The wording of Theorem 4.2 (ex-5.2) improved its presentation and connection with pushforward generated distributions. Nonetheless, I still think that its relevance is limited because it relies on a min-max interpretation of GANs which has been questioned in the literature (cf. my initial review). I would not recommend it to remove it from the paper as it can be valuable, but rather to discuss its relevance.

---

> > ### Author Response · Authors · 2023-03-28
> > **Response to remaining concerns**
> >
> > > I would like to thank the authors for their detailed answers, which I read in details along with the other reviews. Before providing a definitive recommendation to the AE, let me take this opportunity to follow up and express my remaining concerns.
> > I would like to stress that the authors tackled and addressed many of my concerns in their response and revision.
> > However, I still think that main weaknesses either still hold or were incompletely addressed, in particular the following ones, listed below in decreasing order of importance. Points 1. and 2. are particularly crucial as they affect key claims of the paper and are also shared by the other reviewers.
> >
> > __Response__: Thank you for your assessment. Please find our point-by-point responses to the additional concerns raised. The corresponding changes in the revised documents are highlighted in __magenta__.
> >
> > > __(C.1)__: The purpose of the experiments unfortunately remains unclear. While the authors state that "the goal of the experiments in Poly-WGAN is to gain a deeper understanding of how GANs work", the experimental section remains focused on experimental performance. This induces two problems which should be solved by clarifying the aims and conclusions of the experimental section.
> > > - As stated in my initial review, the current performance achieved in the experiments is outdated, limiting their informative value beyond the fact that knowing the analytical discriminator expression is beneficial for generator training.
> > > - The experiments, focusing on performance, thus provide little further information on GANs beyond the already developed theoretical results. They lack, for instance, a comparison between PolyWGAN and its strict equivalent with a trained discriminator, including the same gradient penalties, that would allow to further assess the relevance of the theoretical study.
> >
> > __Response__: To better understand the discriminator used in Poly-WGAN, we have included additional ablation experiments comparing against trainable discriminators (now moved to the main manuscript, __Section 6.3__). We now include comparisons against two baseline discriminator Scenarios -- (i) A neural-network discriminator,  trained using the regularized Poly-WGAN loss. The higher-order gradients are computed by means of nested _automatic differentiation_ loops. (ii) A trainable version of the Poly-WGAN discriminator, wherein the centers and weights are initialized as in Poly-WGAN, but are subsequently updated by means of an un-regularized WGAN loss.
> >
> > > __(C.2)__: While the provided clarifications on the validity conditions of the theoretical results as well as on the uniqueness and optimization space of the discriminator are appreciable, their integration in the revised manuscript should be improved. In the current state of the paper, these discussions are scattered around the theoretical results instead of being included in their statement, which obfuscates their interpretation. This should be solved by partially rewriting the theoretical parts to make assumptions clearer.
> >
> > __Response__: The theorems have been revised as suggested. (cf. revised versions of Theorem 3.1 and Theorem 4.1).
> >
> > > __(C.3)__: Even though the authors provided many clarifications and implemented some of my suggestions, the introduction of Beppo-Levi spaces remains scattered throughout the paper. Section 5 should probably be renamed as it mostly consists in discussions of the previous theoretical results, instead of being framed as "practical considerations".
> >
> > __Response__: The discussion on the _”Constraint Space of the Discriminator”_ is composed  into a new section (cf. __Section 4.1__). The discussion on _“Practical Considerations”_ has been moved to __Section 4.3__. The revised version of __Section 5__ deals with the interpretation of the optimal discriminator and related works.
> >
> > > __(C.4)__: Less importantly, some of my suggestions to nuance some claims were not completely taken into account.
> > > - Section 1.1 still includes without discussion the claim that discriminators gradients can be sums of Dirac deltas, which may not be true when discriminators are implemented by neural networks in practice.
> > > - The wording of Theorem 4.2 (ex-5.2) improved its presentation and connection with pushforward generated distributions. Nonetheless, I still think that its relevance is limited because it relies on a min-max interpretation of GANs which has been questioned in the literature (cf. my initial review). I would not recommend it to remove it from the paper as it can be valuable, but rather to discuss its relevance.
> >
> > __Response__: The discussion in __Section 1.1__ has been reworded suitably. Additional discussions have been provided in __Section 4.2__ to improve clarity on the relevance of Theorem 4.2.

---

> > > ### Comment · Reviewer_toYB · 2023-03-29
> > > **Acknowledgement**
> > >
> > > I would like to thank you for your additional response and revision, which I will consider when providing my recommendation in the next few days.

---

### Review · Reviewer_YSPr · 2023-03-06

**Summary Of Contributions:**

* The paper studies the optimal GAN discriminators subject to high-order gradient-norm regularization, and casts the discriminator optimization as high-dimensional interpolation problem.
* The paper shows (theorem 3.1) that in the case of Regularized LSGAN, the optimal discriminator is a radial basis function network. This RBF's centers and weights are computed on batches of generated samples, and the (fixed) discriminator is used to train the generator with SGD. * The paper studies high-order gradient-norm regularization in IPM-GANs, and specifies the optimal discriminator and generator (theorem 5.1) and the optimal generator density (theorem 5.2). The paper proposes Poly-WGAN using RBF network as a more practical approximation to the optimal discriminator. Experimental validation shows that Poly-WGAN is
* Experimental results of the paper shows that Poly-LSGAN and Poly-WGAN works well on 2D Gaussian problems, but fail to scale to image generation problems and the Ploy-WAE is competitive with other adversarial autoencoders.

**Audience:**

Yes

**Claims And Evidence:**

No

**Requested Changes:**

* The paper claims that their method is "two orders faster" (abstract) and "two times faster" (section 6.1). I'd like the authors to be more specific and explain if speedup is a computational or statistical.
* In general, to explain why studying higher-order gradient regularization is an interesting path to pursue for generative models, and help educate the community on further directions for improving performance on more practical settings.

**Strengths And Weaknesses:**

+ The paper studies high-order gradient norm in GANs and derives the optimal a closed-form for the optimal discriminator in both LSGAN and WGAN.
+ The paper provides promising results on 2D data showing better generation and faster convergence than comparable methods.
- The paper claims that their analysis "help[s] in understanding the optimal structure and behavior of gradient-regularized GAN discriminators". However, I find that the paper uses the analysis to propose an algorithm (which only works in low dimensional data) but doesn't really provide a deep insight into their optimization as claimed in the paper.
- While it is interesting to see promising results even in two dimensional cases, I believe that more analysis needs to be done to understand why the proposed algorithms are better than other method, especially since the current method doesn't scale to high dimensional data and more work is needed to pursue this line of research.

---

> ### Author Response · Authors · 2023-03-18
> **Response to Reviewer YSPr**
>
>
> Thank you for the insightful comments and suggestions. Please find below, our point-by-point response to your comments.
>
> ## Weaknesses
>
>  > __(C.1)__ The paper claims that their analysis "help[s] in understanding the optimal structure and behavior of gradient-regularized GAN discriminators". However, I find that the paper uses the analysis to propose an algorithm (which only works in low dimensional data) but doesn't really provide a deep insight into their optimization as claimed in the paper
>
> __Response__: The proposed higher-order gradient penalty provides a general framework for gaining insights into the role of the discriminator in GANs. The interpolation property established in this paper can potentially inspire newer architectures and implementations of the discriminator. We derive, through the _Calculus of Variations_, the optimal discriminator in closed-form and show that, in gradient-regularized LSGAN and WGAN, its form is that of a radial basis function (RBF) interpolator. A visualization of this discriminator, for various choices of $m$ in 1-D is provided in __Section 5.1__ of the revised manuscript. We have also improved the clarity of the claim made in __Section 1__ of the revised manuscript.
>
>  > __(C.2)__ While it is interesting to see promising results even in two dimensional cases, I believe that more analysis needs to be done to understand why the proposed algorithms are better than other method, especially since the current method doesn't scale to high dimensional data and more work is needed to pursue this line of research
>
>  __Response__: The proposed RBF discriminator results in superior performance compared to GANs with a trainable discriminator. We hypothesize that the ideal neural-network discriminator in gradient-regularized GANs must be approximating the closed-form interpolator. To validate this hypothesis, we train the baseline discriminator for $D_{\mathrm{iters}}$ iterations per generator update, and show that, as $D_{\mathrm{iters}}$ increases, the performance of the baselines approaches that of Poly-WGAN (cf. __Appendix E.3__ of the revised manuscript).
>  As with other kernel-based methods, Poly-WGAN is limited by the number of centers that can be practically used in the RBF expansion in $n$-D, while not causing out-of-memory issues in training the generator. Exploring alternative generative modeling schemes that could leverage the optimal form of $D_p^*(x)$ is a promising direction for further research.
>
>
> -------------------
> -------------------
> ## Requested Changes
>
>
> > __(C.3)__ The paper claims that their method is "two orders faster" (abstract) and "two times faster" (section 6.1). I'd like the authors to be more specific and explain if speedup is a computational or statistical.
>
> __Response__: The speedup is in computation time and also reflects as reduced training time. A plot of FID vs. iterations in PolyGAN-WAE demonstrates the speedup over the baselines (cf. Section 6.1). The computational speedup was demonstrated using PolyGAN-WAE and PolyGAN-D in Appendix F of the initial submission (cf. __Table 13__ of the revised manuscript). We have also included additional comparisons on the compute time per generator update (_i.e.,_ the time between successive generator updates) in Poly-WGAN and the baselines in __Appendix E.3__ and __Appendix F.3__ of the revised manuscript (cf. Response to Comment __(C.4)__ to Reviewer _NVin_ and Response to Comment __(C.1)__ to Reviewer _toYB_, respectively).
>
> > __(C.4)__ In general, to explain why studying higher-order gradient regularization is an interesting path to pursue for generative models, and help educate the community on further directions for improving performance on more practical settings
>
> __Response__: Polyharmonic interpolation was shown to solve the general gradient-regularized interpolation task in $n$-D (Duchon, 1977). We showed that the WGAN discriminator effectively functions as an interpolator, whose output depends on the neighborhood of the input sample. This interpolating property of the discriminator, and the connection between gradient-regularization and polyharmonic kernels is the link that is established in this paper. We have updated __Section 1.1__ of the revised manuscript to motivate the problem better.
> Access to the optimal discriminator allows for analyzing/exploring efficient implementations of the GAN discriminator. The closed-form discriminator can also be leveraged to explore flow-based implementations of Poly-WGAN. We discuss this aspect in __Section 5.2__ of the revised manuscript.

---

### Review · Reviewer_NfBV · 2023-03-15

**Summary Of Contributions:**


The authors consider the high-order gradient penalty for both LSGAN and WGAN.

Theoretically, they showed that the optimal LSGAN discriminator, under high-order gradient penalty, is a radial basis function network, where the weights can be obtained by solving the linear system of equations (Thm 3.1). Due to computational efficiency, they proposed the high-order gradient penalty WGAN and showed the optimal discriminator is also a radial basis function network based on the fundamental solution to the polyharmonic equation (Thm 5.1). Moreover, they specified the optimal generator density in this scenario (Thm 5.2).


**Audience:**

Yes

**Broader Impact Concerns:**

None.

**Claims And Evidence:**

Yes

**Requested Changes:**

Major changes:

(Appendix A) Need to include the condition under which the extremum is the minimum.

(Appendix B) Need to show that this extremum is indeed the minimum.

Typos and minor changes:

(Thm 3.1) The "monomials" is kinda confusing here. I am not sure about the right word though.
(Thm 3.1) Matrix A is required to be full-rank instead of B.
(Equation 7) K|X| should be outside the integral.
(Lemma 5.3) It is awkward to call this a lemma.
(Page 21) The second last equation has two (-1)^m.
(Equation 19) Bv is a vector but the first term is a scalar.
(Equation 20) (-1)^m was missing.
(Equation 21) Maybe mention A is invertible here.


**Strengths And Weaknesses:**

Strengths:

High-order gradient penalty is quite novel in GAN literature.

The idea of obtaining the optimal discriminator based on the PDE solution is interesting and new, which improves understanding of the structure of discriminators.

Weaknesses:

(Thm 3.1) Such optimal D* is only shown to be an extremum instead of a minimum. The proof lacks the procedure of checking the sign of second-order variation.

(Thm 3.1) The linear system may not be consistent, which means such D* may not
even exist.

(Thm 3.1 and 5.1) Uniqueness was not examined.

---

> ### Author Response · Authors · 2023-03-18
> **Response to Reviewer NfBV**
>
>
> Thank you for the insightful comments and suggestions. Please find below, our point-by-point response to your comments.
>
> ## Weaknesses
>
>  > __(C.1)__ (Thm 3.1) Such optimal $D^*$ is only shown to be an extremum instead of a minimum. The proof lacks the procedure of checking the sign of second- order variation
>
> __Response__:The second-order variation check was already included in the initial submission (cf. __Appendix C.2__, particularly the _"The sign of $\lambda_d^*$"_). The condition on the sign of the second-variation translates to the sign of the optimal Lagrange multiplier $\lambda_d^*$ (now, Page 26 of the revised manuscript).
>
>
>  > __(C.2)__ (Thm 3.1) The linear system may not be consistent, which means such $D^*$ may not even exist
>
>  __Response__: We agree. However, the consistency issue does not arise if the set of real/fake centers are unique, and the kernel order $2m-n\geq 0$. We have clarified this aspect in __Appendix A__ of the revised manuscript.
>
>   > __(C.3)__ (Thm 3.1 and 5.1) Uniqueness was not examined.
>
>  __Response__: The linear system in Theorem 3.1 has a unique solution when the matrix $\mathbf{\mathrm{B}}$ is full column-rank. This property may be violated if the centers lie in a constrained low-dimensional manifold in $\mathbb{R}^n$. The solution to Theorem 5.1 (Theorem 4.1 in the revised manuscript) is unique up to the homogeneous component $P(x)$ (Aronszajn _et al.,_ 1983). However, the convergence of the generator to the desired target distribution is independent of the homogeneous component (cf. __Theorem 4.2__ of the revised manuscript).
>
>
>
> -------------------
> -------------------
> ## Requested Changes
>
>
> > __(C.3)__ Need to show that the extremum is indeed the minimum.
>
> __Response__: These results were already part of the original submission ( __Appendix C.2__ in particular  _"The sign of $\lambda_d^*$"_, now Page 26 of the revised manuscript).
>
> > __(C.4)__ (Thm 3.1) The "monomials" is kinda confusing here. I am not sure about the right word though. (Thm 3.1) Matrix A is required to be full-rank instead of B. (Equation 7) K|X| should be outside the integral. (Lemma 5.3) It is awkward to call this a lemma. (Page 21) The second last equation has two (-1)^m. (Equation 19) Bv is a vector but the first term is a scalar. (Equation 20) (-1)^m was missing. (Equation 21) Maybe mention A is invertible here.
>
>
> __Response__: _Monomials_ refers to the individual terms of the $(m-1)^{th}$-order polynomial, whose coefficients constitute the vector $\mathbf{v}$. Since this usage seems to have led to a confusion, we have dropped it altogether. Both matrices $\mathbf{\mathrm{A}}$ and $\mathbf{\mathrm{B}}$ must be full rank for the solution in Eqn. (7) to exist. $\mathbf{\mathrm{B}}\mathbf{v}$ has been rectified to correctly indicate $[\mathbf{\mathrm{B}}\mathbf{v}]_i$.
> The other suggestions have been incorporated.

---

### Author Response · Authors · 2023-03-18
**Summary of Changes**


Dear Action Editor and Reviewers:

Thank you for the insightful review comments. The following is a summary of changes incorporated in the revised manuscript:

- The __title of the paper__ has been revised to better reflect the contributions of the paper.
-  We have rewritten the Motivation section to place in better context the higher-order gradient regularizers (now __Section 1.1__).
- __Section 3__ includes details on the domain and constraint space of the optimal discriminator in Theorem 3.1. We also discuss the existence and uniqueness of the solution.
- Experiments on Poly-LSGAN are summarized in __Section 3.1__.
- Improved clarity on the uniqueness of the solution in __Theorem 4.1__.
- Included a new __Section 5.1__ on interpreting the optimal discriminator in PolyGANs with illustrations in 1-D.
- Included a new __Section 5.2__ discussing more related works on GANs, gradient flows and neural tangent kernels.
- Additional ablation experiments on Gaussian learning have been included (cf.__Appendix E.3__).
- Additional experiments on learning the latent-space distribution of a pre-trained autoencoder using Poly-WGAN (cf. __Appendix F.3__).
- Results on comparing the latent-space alignment between PolyGAN-WAE and the baselines (cf. __Appendix F.4__).
- Bug-fixes, corrections to typographical errors and figure captions as suggested.
- Anonymous (2023) is now accepted for publication in JMLR with minor revisions. The citation has been updated accordingly.

The portions in the manuscript where the changes have been made are highlighted in _blue_.

Thank you.

Authors

---

> ### Author Response · Authors · 2023-03-28
> **Summary of Changes (Revision No.2)**
>
> Dear Reviewers:
>
> Please find below a summary of changes incorporated in the second revision of the manuscript (in the order in which they appear, highlighted in __magenta__ in the revised manuscript):
>
> - __Section 1.1__ has been revised as suggested by Reviewer _toYB_.
> -  Theorem 3.1 has been updated to make the assumptions explicit.
> - The discussions on the _”Constraint Space of the Discriminator”_ are composed into a new section (cf. __Section 4.1__).
> - Improved the clarity on the uniqueness of the solution to __Theorem 4.1__.
> - The discussions on _“Practical Considerations”_ has been moved to __Section 4.3__.
> - The revised __Section 5__ deals with the interpretation of the optimal discriminator and related works.
> - Included additional ablation experiments comparing against trainable discriminators, and moved existing ablation experiments (Appendix E.3 of Revision 1) to the main manuscript (cf. __Section 6.3__).
>
> Thank you.
>
> Authors

---

### Decision · Action_Editors · 2023-05-09

**Recommendation:** Reject

**Comment:**

The experiments carried out in the paper do not provide clear evidence of any significant experimental advantages compared to standard GANs, mainly due to their limited scope. Additionally, the experiments do not demonstrate any new insights enabled by the proposed framework. To address this issue, the paper could benefit from shifting the scope of the experiments and revising the corresponding sections.

Sadly I don't think this is only a minor revision as the current version of the paper would benefit from a complete rewriting to include better the comments made by reviewers to have a more precise and clear version of the current work. Yet some progresses have been made during the reviewing process; the current version lacks a clear message into the presentation of the results according to the reviewers.

**Audience:**

Some individuals in TMLR's audience would likely be interested in knowing this paper's findings, particularly those involved in research on Generative Adversarial Networks (GANs) and their optimization. The paper proposes compelling theoretical results on the optimal discriminator in GANs involving higher-order gradient penalties, which could interest researchers working on improving the training of GANs. Additionally, while the practical value of the paper is limited at the moment, the results could eventually lead to later empirical advances, which could be of interest to researchers working on developing more effective GAN models. However, the lack of clear experimental evidence demonstrating a relevant observed advantage concerning standard GANs may limit the interest of some individuals in the audience who are more focused on the practical applications of GANs.
Studying the effect of high-order gradient regularization in generative modeling has been pointed out as an interesting question by all reviewers.

**Claims And Evidence:**

It is diffult to answer this questions since reviewers suggests that the theoretical results proposed in the submission are compelling and well-motivated in the literature, but it also indicates that the practical value of the paper is limited at the moment.

Furthermore, at the moment the experiments do not demonstrate clear evidence of a relevant experimental advantage with respect to standard GANs, which suggests that the claims made in the submission may not be supported by accurate, convincing and clear evidence. However, it is possible that the issues with the experiments could be addressed in a revised version of the paper, so it is not possible to definitively say whether the claims made in the submission are or are not supported by accurate, convincing and clear evidence without further information. toYB provided several possibilities to have better evidence of the claims but some of them are not matched in the current version of the work.